# Technical Note:
# General Formulation For the Distribution Problem:
# Prognostic Assumed PDF Approach Based on
# The Maximum–Entropy Principle and
# The Liouville Equation

Jun-Ichi Yano[1], Vincent E. Larson[2,3], and Vaughan T. J. Phillips[4]

[1]CNRM UMR3589, CNRS and Météo-France
[2]Department of Mathematical Sciences, University of Wisconsin — Milwaukee
[3]Pacific Northwest National Laboratory
[4]Department of Physical Geography and Ecosystem Science, Lund University, Lund, Sweden

**Correspondence:** Jun-Ichi Yano (jun-ichi.yano@cnrs.fr)

**Abstract.** A general formulation for the distribution problem is presented, which is applicable to frequency distributions of subgrid–scale variables, hydrometeor size distributions, as well as to probability distributions characterizing data uncertainties. The general formulation is presented based upon two well–known basic principles: the maximum–entropy principle and Liouville equation. The maximum–entropy principle defines the most likely general distribution form, if necessary constraints are specified. This paper proposes to specify these constraints as the output variables to be used in a host model. Once a general distribution form is defined, the problem of temporal evolution of the distribution reduces to that of predicting a small number of parameters characterizing it. This paper derives prognostic equations for these parameters from the Liouville equation. The developed formulation, which is applicable to a wide range of atmospheric modelling problems, is specifically applied to condensation growth of cloud droplets as a demonstration.

## 1 Introduction

The present paper considers the distribution problem in a general manner. As the problems of distributions, three examples are, at least, identified in atmospheric sciences.

The first and perhaps most obvious example is the problem of determining the distribution of a variable over a domain as a distribution density function (DDF). Typically, the domain corresponds to a grid box in a numerical simulation, and the obtained distribution is used for calculating subgrid-scale characteristics that are required by a host model. This problem may be called the *subgrid-scale distribution problem*. A specific application of this problem is the determination of the fractional area occupied by clouds within a grid box (Sommeria and Deadorff 1977, Mellor 1977, Bougeault 1981, LeTreut and Li 1991, Bechtold *et al.* 1992, 1995, Richard and Royer 1993, Bony and Emanuel 2001, Golaz *et al.* 2002, Tompkins 2002). Arguably all subgrid-scale processes may be represented under this subgrid-scale distribution framework (*cf.*, Yano 2016).

The second example is the size distribution of hydrometeor particles (condensed water, ice species, etc). Information on particle size distributions (PSDs) is crucial for predicting various conversion processes from one hydrometeor type to another as well as for evaluating fall–out rates of those hydrometeors (*cf.*, Khain *et al.* 2015, Khain and Pinsky 2018). The rates of all these processes depend sensitively on the hydrometeor particle size.

     The third and perhaps most abstract example of the distribution is the probability. The notion of probability appears in many
places of the atmospheric sciences (*e.g.*, Yano and Manzato 2022). Here, we especially keep in mind applications to data assimilation, in which data uncertainties are measured by probability distributions (*cf.*, Wikle and Berliner 2007).

     The present paper considers all three of these problems under a single framework. It is possible to consider these three qualitatively different problems together because time evolution of all these distributions can, in principle, be predicted by similar equations. The time evolution of both the subgrid-scale distribution (DDF) and the probably density function (PDF) is
30 predicted by the Liouville equation (*cf.*, Sec. 3.5). Here, it is hard to overemphasize the clear difference between them: simply put it, distribution and probability are distinctively different concepts. Unfortunately, in the literature, both are often referred to as PDFs. We follow this custom to some extent, but we will add "DDF" in parentheses whenever it is not cumbersome. Thus, though these two problems deal with different types of distributions, both are governed by the same equation. Time evolution of PSDs is, fundamentally, described by the stochastic–collection equation by adding additional tendency terms to
35 it (*cf.*, Sec. 3.6). Though its form is not identical to the Liouville equation, it can still be considered in an analogous manner. Thus, it becomes possible to deal with these three different problems under a single framework.

     All three of these problems also suffer from the same difficulty: direct use of these fundamental equations (Liouville and stochastic–collection) turns out to be very expensive computationally. In data assimilation, an ensemble–forecast method is adopted as an alternative, but the difficulty remains of generating a statistically large enough ensemble. Thus a numerically
more efficient method must be sought in order to make them practically useful. This difficulty has been, so far, addressed separately in these three problems. One originality of the present work is to simply point out that all three of these computational problems can be considered under a common framework.

     In the subgrid-scale distribution problem, a main strategy is to assume a simple distribution form that is specified by a small number of parameters, sometimes termed "PDF parameters". The PDF parameter values, and hence the distribution
itself, evolve with time as conditions change from, say, overcast cloud to cumulus cloud. Hence the time evolution of the PDF parameters must be predicted. Typically, this is done by first prognosing a set of moments of the distribution and then inverting the set to solve for the PDF parameters. This general approach can be called an *assumed PDF* method. One of the earliest applications of the idea to turbulence is by Lockwood and Naguib (1975). See also an early review by Pope (1979).

     A main strategy in cloud microphysics is to integrate the full information over all the possible particle sizes. Thus, the
50 distribution problem becomes implicit. This approach is called *bulk*. However, for performing integrals over particle size, we need to assume a certain distribution form, which is typically exponential or the gamma distribution. In this manner, we see a clear link of the "bulk" microphysical approaches to the "assumed PDF" approach.

     In data assimilation, typically, a decision is made to focus only on the mean, variance, and some covariances of a probability distribution. As a result, a full probability is not explicitly considered. Here, typically, an "assumed PDF" is Gaussian.

As overviews about how these three problems are constructed using the distributions, we refer to Machulskaya (2015) for the subgrid-scale distribution problem, Seifert and Beheng (2001, 2006), Khain *et al.* (2014), and Khain and Pinsky (2018) for microphysics, and Bannister (2017), and Carrassi *et al.* (2018) for the data assimilation. Among those three problems, the role of probability may be least obvious in data assimilation, especially for those who only consult the final formulation of the standard variational principle. Here, we specifically refer to Sec. 2 of Carrassi *et al.* (2018), in which a more formal formulation in terms of the probability is presented.

The purpose of this study is to present a more coherent, and self–contained formulation for the distribution problems under the framework of those assumed–PDF approaches in a general sense. The assumed–PDF approaches solve only about a half of the whole problem. There are still two major issues to be resolved: i) a choice of an assumed PDF form and ii) methodologies for evaluating the time evolution of the introduced PDF parameters efficiently. The present study proposes the solutions to those two major issues. Currently, there is no clear guiding principle for the first issue (*cf.*, Sec. 3.2.2). The second issue is currently dealt with by relating the PDF parameters and moments to each other, but the conversion from moments to parameters is not guaranteed to be simple or analytic (*cf.*, Sec. 3.2.1). By addressing these two major issues in the distribution problem, the present study generalizes the formulation for the existing assumed–PDF approaches developed for the subgrid–scale distribution problem into more general distribution problems. By doing so, the present study links the subgrid–scale distribution problem to the distribution problem, more general, as found in *e.g.*, cloud microphysics and data assimilation. Conversely, the generalized formulation of the present study reduces to the existing assumed–PDF approaches by introducing additional approximations and assumptions (*cf.*, Sec. 5.4).

Here, these two questions are addressed separately in this study in Secs. 4 and 5, respectively. Thus, those two methodologies can also be adopted independently, if desired. Yet, the present study suggests that the most coherent formulation for the assumed–PDF approaches can be developed by adopting both methodologies together.

In this manner, the present work also constitutes an effort to develop a unified and coherent formulation for subgrid–scale representations (Yano 2016, Yano *et al.* 2014, 2018). Readers are encouraged to refer to them for the authors' general perspectives on the subgrid-scale representations, but see further Yano (2015a, b). Refer especially to Yano (2016) for general discussions of the subgrid–scale distribution problem. Keep in mind that the subgrid-scale representation problem is considered exclusively from a perspective of DDF in the present study, but without excluding the possibilities of alternative approaches as discussed in Yano (2016). More generally, the fundamental research, as pursued in the present study, is extremely crucial for the improvements of subgrid–scale parameterizations (Yano *et al.* 2014). Even the current operational numerical forecasts, which are based on a premise of high–resolution cloud modeling, may break down without such fundamental research to support them (Yano *et al.* 2018).

The paper begins in the next section by introducing a basic governing equation system, that is adopted throughout the paper to construct the general formulation for the distribution problems. As seen therein, the adopted governing equation is general enough that the formulation of the present study can be applied to more or less any problems that can be expected in atmospheric sciences. Sec. 3 reviews our basic knowledge about the distribution problem. The moment concept is first introduced in Sec. 3.1, because it is so central in the current approaches. The basic idea of the assumed PDF approach is

outlined in Sec. 3.2, where its basic problems are also pointed out. The maximum–entropy principle, which defines the form for the most likely general distribution under given constraints, is introduced in Sec. 3.3, and its applications are further discussed in Sec. 3.4. The Liouville equation is introduced in Sec. 3.5, and the stochastic–collection equation is separately introduced in Sec. 3.6.

Both the maximum–entropy principle and the Liouville equation play key roles in the present study, but in different manners, for resolving the aforementioned two major issues. The first issue is addressed by taking the maximum entropy as a guiding principle. A key open question here is the precise conditions to be posed under this principle in order to define a PDF (DDF). A standard procedure is to take what we already know as a specified condition of a system (*e.g.*, total kinetic energy of an ideal–gas system). An original argument of the present paper is to propose to constrain the form of the PDF (DDF), rather, by the quantities that are required, for example, for purpose of predictions (*e.g.*, cloud fractions, precipitation rate) in modelling. This argument is developed in Sec. 4 by discussing the contrasts between the standard statistical problems and those typically addressed in atmospheric sciences. The second issue is addressed in Sec. 5 by deriving from the Liouville equation a general form of prognostic equations for predicting the time evolution of the PDF parameters introduced under an assumed PDF. The derived general formulation is applied to the illustrative example of condensation growth of cloud droplets in Sec. 6. The paper is concluded in Sec. 7.

It is emphasized that this work presents a new formulation by addressing the aforementioned two major problems in the assumed-PDF approaches, rather than solving any specific problems. For this reason, the basic style is to present basic principles first in a straight and concise manner. We choose this style for ease of referring to formulations, especially based on the fact that the presented formulation can be applied to almost any distribution problems in atmospheric sciences. Specific examples are gradually introduced so that a more concrete picture of the methodology gradually emerges. Readers who would like to see a concrete example first are suggested to go to Sec. 6 directly, and read backwards. Also for this reason, the governing equation system to be considered in the present study is introduced in a stand–alone manner separately in the next section so that the range of the applicability of the present study to various atmospheric problems should become immediately clear. Along the line of generality intended in the present study, all the equations are presented in nondimensional forms, setting the various physical constants to unity for convenience, throughout the paper.

## 2  Governing Equation System

Many physical variables, say, $\phi$, of the atmospheric flows are advective, thus they are governed by an equation of the form:

$$\frac{\partial \phi}{\partial t} = -\mathbf{v} \cdot \nabla \phi + F. \tag{2.1}$$

Here, $\mathbf{v}$ is the velocity, and $F$ designates all the tendencies (source) contributing to the variable, $\phi$, apart from the advection as represented as the first term on the right–hand side. The source, $F$, generally depends on the variable, $\phi$, and also possibly on time $t$ and space. For instance, in a cloud macrophysical application, $\phi$ is the liquid water content or the number density of liquid droplets with a particular radius. Eqs. (2.9), (2.10), (2.11) of Machulskaya (2015) are examples of the equations

of the form (2.1) considered in the context of the DDF problem. Yano *et al.* (2005) and Yano (2016) show that the basic formulations for the subgrid–scale parameterizations can be reproduced by simply examining this general form (2.1). More specifically, Yano (2014) shows that all the essential, basic standard formulas for the mass–flux convection parameterizaiton can be reproduced from a general governing equation (2.1).

With an ultimate application to the systems described by governing equations of the form (2.1) in mind, for ease of the deductions in the following, the present study focuses on a case with no spatial dependence in the above:

$$\frac{d\phi}{dt} = F. \tag{2.2}$$

Yet, in spite of this restriction, and without arguing for any general physical relevance, it may also be worthwhile to emphasize that the source term, $F$, in Eq. (2.2) includes any type of physical processes that are locally defined. Most of the microphysical processes, for example, fall to this category. More importantly, this restriction does not have any seriouis consequences, because the final general formulation for predicting the assumed–PDF parameters in Sec. 5.1 is very easily generalized to the cases with spatial dependence (*cf.*, Sec. 5.3)

To maintain the generality of the formulation, the term, $F$, is left unspecified in considering the time evolution of various types of distributions in Sec. 5. As a result, the formulations presented in the following, especially our key result given by Eqs. (5.8a, b) is applicable to any types of physics. All we have to do is to specify the form, $F$, as required in applications. As a specific example, the source term is set $F = 1/r$ in Sec. 6 (*cf.*, Eq. 6.1a) in considering the condensation growth of a droplet with a radius, $r$. We also examine the behavior of systems with mathematically simple forms for $F$ in Sec. 5.5 as well as in the Appendix C. Keep in mind that in Eq. (2.1), the source term, $F$, can be space dependent, say, involving spatial derivatives. However, in considering Eq. (2.2) in the following, this possibility will be excluded for ease of the analysis. Furthermore, in the present study, the source term, $F$, is assumed to be deterministic, but except for the case of the Brownian motion considered in Sec. 4.4.

Throughout the study, only the cases with a single variable, $\phi$, are considered explicitly for the economy of presentation. However, when multiple variables are involved in a problem, as typically the case in any realistic applications, the only modification required is to replace the scalar, $\phi$, by a vector. Examples with systems with multiple dimensions are presented in Yano (2024). Probably, a more serious restriction in the following development of the formulation is in considering only the cases with no spatial dependence. However, as it turns out, the generalization of the final formulation to the spatially–dependent cases with Eq. (2.1) is fairly straightforward, as going to be discussed in Sec. 5.3.

In the present work, we proceed with the hypothesis that the physics of a system is already completely known in the form (2.2) with a forcing term, $F$, completely specified. In practical applications, this hypothesis is satisfied by specifying all the terms in a system in a closed form, as parameterizations, if required. Especially, in applying the general formulation of the present study to the subgrid–scale distribution problems, this hypothesis means that we know the governing equation system of the small–scale to be parameterized fully: *cf.*, Sec. 4.3 for further discussions. Under this spirit, for example, Yano (2014) proceeds with a hypothesis that we know perfectly the equations for the cloud–resolving modeling, and re–constructs the standard mass–flux based convection parameterization based on this hypothesis. Same wise, the present method solves a

subgrid–scale distribution problem assuming that we know the full equations for all the scales of a system. As going to be seen, for this reason, we do not need any *turbulence closures* (*cf.*, Meller 1973, Mellor and Yamada 1974) in the present formulation.

# 3 Basic Principles

The purpose of this section is to summarize well–known basic principles for describing PDFs (DDFs).

## 3.1 Distributions and Moments

Let us designate a PDF (DDF) for a variable, $\phi$, by $p(\phi)$. Moments, $\langle \phi^n \rangle$ $(n = 1, 2, \ldots)$[1] can be constructed from a given distribution, $p$, by

$$\langle \phi^n \rangle = \int \phi^n p d\phi. \tag{3.1}$$

Here, an unspecified integral range may be taken from $-\infty$ to $+\infty$ with many of the physical variables, but some physical variables are semi–positive definite (*e.g.*, temperature, mixing ratios). In the latter case, the integral range above must be from 0 to $+\infty$.

The series of moments may be interpreted analogously to the Taylor series, in the sense that it constrains a function. However, unlike the latter, there is no closed analytical formula for reconstructing the original distribution from a given series of moments: although a series of moments can be derived from a given distribution in a straightforward manner, the reverse is hardly the case. This is in spite of the extensive literature on the subject (*e.g.*, Daniels 1954, Butler 2007, Dang and Xu 2019). On the other hand, the usefulness of the moments for describing the turbulent flows can be hardly overemphasized (*e.g.*, Tatsumi 1980, Stull 1988, Garratt 1992), either.

## 3.2 Assumed PDF(DDF)

### 3.2.1 General Formulation

The basic idea of assumed PDF (DDF) approach is to introduce a generic form of PDF (DDF) characterized by a few free parameters, say, $\lambda_i$ $(i = 0, \cdots, N$, where $N$ is kept as small as possible) but in such manner that the distribution of a variable of concern can be represented:

$$p = p(\phi, \lambda_0, \lambda_1, \cdots, \lambda_N). \tag{3.2a}$$

Here, $\lambda_0$ will be used to designate a constant factor for a normalization of a distribution throughout the paper, whenever the assumed–PDF formulation is discussed in a general manner. On the other hand, $p_0$ will be adopted for the normalization factor,

---

[1]In literature, higher moments $(n \geq 2)$ are often defined in terms of a deviation from mean (*i.e.*, the first moment). Here, the definition is presented without discrimnating between the first and the higher moments for mathematical lucidity

whenever a specific PDF form is discussed: the latter choice is consistent with the fact that different notations than $\lambda_i$ $(i \neq 0)$ are also often adopted for the other assumed–PDF parameters.

Importantly, the distribution, $p$, is related to the cumulative probability, $P$, by

$$p = dP/d\phi, \tag{3.2b}$$

where $P$ is more precisely defined as the probability that the variable $\phi'$ is less than a specified value $\phi$, *i.e.*, $P = P(\phi' < \phi)$.

Examples of distributions taking the form (3.2a) are discussed in subsequent subsections (*e.g.*, Eqs. 3.22, 3.23 setting $p_0 = \lambda_0$ therein). Once the functional form of a distribution is constrained by Eq. (3.2a), the problem of determining the distribution $p(\phi)$, which must be defined for every value of $\phi$, reduces to that of determining a given finite set of parameters $\{\lambda_i\}$, which evolve with time by following that of the distribution. However, keep in mind that the parameters $\{\lambda_i\}$ should not depend on $\phi$ for obvious reasons. Note that $\{*\}$ indicates a set of parameters throughout the paper.

Here, it is important to remember that an assumed PDF form is only an approximation: to state this fact more emphatically, it may be better to state it as

$$p = p(\phi, \lambda_0, \lambda_1, \cdots, \lambda_N) + \varepsilon \tag{3.2c}$$

with $\varepsilon$ designating the possible error under this approximation. Yet, in the following deductions, this error term is mostly neglected, except for couple of exceptions of adding it as a reminder.

A major exception to the above rule is when the assumed PDF is an exact solution of the original equation, and when the initial PDF follows the assumed form, there is no error. We may further expect that the error remains small even if the initial PDF does not follow the assumed form. Otherwise, there is no way that an assumed PDF form predict the evolution of a distribution in any accurate manner.

The parameters $\{\lambda_i\}$ defining the distribution (3.1) may be determined, for example, from a known set of moments, $\langle \phi^n \rangle$, *i.e.*,

$$\lambda_i = \lambda_i(\{\langle \phi^n \rangle\}). \tag{3.3}$$

The prognostic equations for these moments, or diagnostic approximations of these equations, are, in turn, known from *e.g.*, the turbulence theories for the system (2.1) in the context of the subgrid–scale distribution problem; thus the problem is closed in this manner. That is the current basic strategy of the assumed PDF (DDF) approach (Larson 2022).

However, there are problems with this strategy. First, the functional form of a PDF (DDF), $p(\phi, \{\lambda_i\})$, must somehow be prescribed. However, no clear principle has been identified. A main thread of this paper is to use the maximum–entropy principle (*cf.*, Sec. 3.3) for this purpose. The second is the difficulty of deriving a closed expression (3.3) for defining the PDF (DDF) parameters from a given set of moments. Here, it is straightforward to compute the moments from a given PDF (DDF), thus we can readily write this down as

$$\langle \phi^n \rangle = \langle \phi^n \rangle(\{\lambda_i\}). \tag{3.4}$$

However, inverting Eq. (3.4) into a form of Eq. (3.3) is often not at all trivial due to the nonlinearity in the former, and PDF parameters are defined only in implicit manner from a set of moments. See Eq. (6) of Milbrandt and Yau (2015), for example. Often extra assumptions and approximations are required to make this inversion possible (*cf.*, Machulskaya 2015). Alternatively, an iterative procedure can be adopted in order to invert a given set of moments and deduce the PDF parameter values (*e.g.*, Lewellen and Yoh 1993). The difficulty of the inversion is exacerbated by prognosing more (higher-order) moments and also by moving to multivariate PDFs. In Sec. 5, we will show that by more explicitly invoking the Liouville equation, as introduced in Sec. 3.5, how a prognostic set of equations for $\{\lambda_i\}$ can be written down explicitly. These equations are closed in the sense that no further inversion is required.

### 3.2.2 Choice of Assumed Distribution Forms

In current assumed–PDF (DDF) approaches in the context of the subgrid–scale distribution problem, distribution forms to be adopted are rather chosen in a subjective manner, mostly based on a computational convenience. For this reason, one of popular choices is double Gaussian, *i.e.*, a sum of two Gaussians, for a purpose of representing a skewness (*e.g.*, Larson *et al.* 2002, Fitch 2019, Naumann *et al.* 2013). Although those studies show some fits to distributions obtained either from observation or large–eddy simulations as a support, we should not consider that double Gaussian distributions are verified by data: no objective comparisons with alternative possible distributions have been made.

On the other hand, observations suggest that the hydrometeor PSD follows an exponential distribution in the size (Marshall and Palmer 1948). However, Yano *et al.* (2016) point out the difficulty of identifying the best fit for the PSDs observationally from various exponential distribution forms, that are derived from the maximum–entropy principle (*cf.*, Sec. 3.3): it is indeed *not* possible to identify in any convincing manner that any of those fits better the observations than the others, although it is possible to discuss about different values of errors by those fittings.

### 3.3 Maximum–Entropy Principle: Derivation

### 3.3.1 Derivation

To address the first issue of the choice of PDF (DDF) form, we take the maximum–entropy as a *guiding principle*. It must be emphasized that this is merely a mathematical principle. Here, the *guiding principle* suggests that it is *not* any physical principle, but it is merely a principle that guides a choice of an assumed PDF form. Note especially that the Boltzmann's entropy, which takes a mathematically identical form, can be derived by a physical reasoning. However, it should not be confused with the information entropy in general. From a physical point of view, although the principle is plausible, there is no guarantee that it actually works. For this reason, we adopt this principle merely as a guide for identifying a necessary assumed form of a PDF (DDF). Such a guiding principle is useful when there is no other principle for choosing an assumed distribution. This principle should not be interpreted as a hypothesis, either, because it suggests that the guiding principle may be disproved by experiments. Here, a success of a guiding principle may vary from the case to case. However, so long as this principle is used with cautions, we expect that it remains useful for choosing a PDF form.

The maximum–entropy principle asks a question of the "most likely" distribution of a variable under a given set of "constraints" (*cf.*, Eq. 3.10 below). It simply argues that the "most likely" distribution is a distribution that actually realizes in a given system. The argument of this principle is simple and appealing enough to gain extensive applications (*cf.*, Kapur 1989), notably for statistical description of the geophysical flows (*e.g.*, Robert and Sommeria 1991, Verkley and Lynch 2009, Verkley 2011, Verkley *et al.* 2016). For this reason, the present study also invokes this principle. See Yano (2019) for further implications of this principle, as well as further references for applications in atmospheric sciences as well as many other disciplines.

Here, the "most likely" is defined in terms of the number of possible combinations for a given state of a variable (*cf.*, Eq. 3.6 below). We develop the idea for a discrete system first for ease of explanation. Thus, we assume a variable, $\phi$, takes $m$ values, say, designated by $\phi_i$ ($i = 1, \ldots, m$). For instance, in a cloud macrophysics application, $\phi$ might represent the liquid water content, whose values might be binned into $m$ categories (*i.e.*, 0 to 1 g/kg, 1 to 2 g/kg, etc.). Let us assume that a total number of data (*e.g.*, measurements, model outputs) is $n$, and among them, $n_i$ takes a value $\phi_i$ ($i = 1, \ldots, m$). For instance, we might sample a cloud $n$ times, each time drawing a value of liquid water content, and we might denote the number of samples that fall into the $i$th bin as $n_i$. Thus, the frequency distribution of the variable, $\phi$, is given by

$$p_i = n_i/n \tag{3.5}$$

with $i = 1, \ldots, m$.

The total number of possible combinations for realizing this distribution is

$$W = \frac{n!}{n_1! \cdots n_m!}. \tag{3.6}$$

By taking a logarithm to the above, and also applying Stirling's formula

$$\log n! = n \log n + O(n), \tag{3.7}$$

which is valid in asymptotic limit of $n \to \infty$, to every integer involved in definition of $W$, we can approximate

$$\frac{1}{n} \log W \simeq -\sum_{i=1}^{m} p_i \log p_i. \tag{3.8}$$

The right hand side of Eq. (3.8) is the information entropy (Shannon 1948), which we shall refer to as "entropy" in short.[2] Thus, the problem of maximizing the number of possible combinations reduces to that of maximizing the entropy, and it leads to the notion of the maximum–entropy principle.

The most extreme case of this distribution is when $\phi$ takes only a particular value, say, $\phi_j$, always, thus $p_i = \delta_{ij}$, using the Kronecker's delta. In this case, there is no possibility of reshuffle the data, thus $W = 1$ and the entropy is zero. Qualitatively, as a variable is more widely distributed, the entropy becomes larger.

A continuous version of the entropy is:

$$-\int p \log p \, d\phi, \tag{3.9}$$

---

[2]However, this entropy should not be confused with the thermodynamic entropy.

where $p = p(\phi)$. However, some subtleties will be remarked about later in Sec. 3.4.1.

In applying the maximum–entropy principle, here, we suppose that the distribution is constrained by $L$ conditions given by

$$\int G_l(p,\phi)d\phi = C_l \tag{3.10}$$

for $l = 1, \cdots, L$. Here, $G_l(p, \phi) = p\sigma_l(\phi)$, and $\sigma_l(\phi)$ are functions of $\phi$; they define the constraints by Eq. (3.10); $C_l$ are known constants. See Sec. 3.4.2 for specific examples (*cf.*, Eq. 3.20), and Yano *et al.* (2016) for physical considerations on choices. Also keep in mind that a distribution is normalized by

$$\int p d\phi = 1. \tag{3.11}$$

The normalization can be considered as a special case of the constraints (3.10) with $G_0 = p$ and $C_0 = 1$ by extending the above series to $l = 0$. Note that exceptionally, when the PSDs are considered, $C_0$ must be equal to the total particle number density.

Thus, the most likely distribution is obtained by maximizing (3.9) under the constraints (3.10) with $l = 0, \cdots, L$. This goal is accomplished by applying a variational principle, as defined by following a standard notation (*cf.*, Ch. 2, Goldstein *et al.* 2002):

$$\delta \left[ -\int p\log p d\phi - \sum_{l=0}^{m} \lambda'_l \int G_l(p,\phi)d\phi \right] = 0 \tag{3.12}$$

with Lagrange multipliers, $\lambda'_l$. The above variation reduces to

$$\frac{\delta}{\delta p} \left[ \int p\log p d\phi + \sum_{l=0}^{L} \lambda'_l \int G_l(p,\phi)d\phi \right] \delta p = \int \left[ \log p + \sum_{l=0}^{L} \lambda_l \frac{\partial G_l}{\partial p} \right] d\phi \, \delta p = 0, \tag{3.13}$$

where the multipliers are re–set to

$$\lambda_l = \begin{cases} 1 + \lambda'_0 & l = 0, \\ \lambda'_l & l \neq 0. \end{cases} \tag{3.14}$$

Noting that $\partial G_0/\partial p = 1$ and $\partial G_l/\partial p = \sigma_l(\phi)$ $(l = 1, \ldots, L)$, and further re–setting to $p_0 = e^{-\lambda_0}$, the most likely distribution under these constraints is

$$p = p_0 \exp[-\sum_{l=1}^{L} \lambda_l \sigma_l(\phi)]. \tag{3.15}$$

Here the constants, $p_0$ and $\lambda_l$ are determined from the constraints (3.10) and (3.11) by directly substituting the distribution form (3.15) into them. This is the basic premise of the maximum–entropy principle: a distribution of a variable, $\phi$, is completely
determined from only $L$ constraints, if they are chosen properly. These $L$ constraints determine $L$–parameters, $\{\lambda_j\}$, that characterize the distribution. Recall that whenever the general assumed–PDF formulation is discussed, we further re–set to $\lambda_0 = p_0$ (*cf.*, Eq. 5.2a).

### 3.3.2 Technical Remarks

A rather ostensible limitation of the general result (3.15) from the maximum–entropy principle is that it does not include the possibility of a distribution zero at the zero value, as is the case with many semi–positive definite, atmospheric variables, in any obvious manner. However, it simply stems from the fact that results from the maximum entropy are not exact: it is based on an approximate logarithmic expression of the number, $W$, of possible combinations under an asymptotic limit of $n \to \infty$ (*cf.*, Eq. 3.7). In this respect, this principle may be considered a special case of the large deviation principle (*e.g.*, Touchette 2009): it can elucidate only a predominant exponential dependence as seen in Eq. (3.15). A possible additional subdominant, algebraic dependence is kept implicit, because such a weak dependency drops out in the given asymptotic approximation. Thus, if required, an algebraic dependence of, say, $\phi^\mu$ can be multiplied on this distribution without contradicting with the given result (3.15). Here, $\mu$ is an unspecified free positive parameter. This slight generalization ensures the condition $p(0) = 0$ as required for many atmospheric variables. See further discussions in Sec. 3.4.3, and further mathematical backgrounds in Guiasu (1977).

## 3.4 Maximum–Entropy Principle: Examples

In order to understand the general distribution given by Eq. (3.15) above better, this sub–section considers some special cases. Implications from the maximum entropy principle are also remarked.

### 3.4.1 Homogeneous Distribution

The simplest case for consideration is one without any constraints (*i.e.*, $L = 0$). Then Eq. (3.15) simply reduces to a homogeneous distribution:

$$p = p_0. \tag{3.16}$$

This means that a variable, $\phi$, has an equal chance for having every possible value, when there is nothing to constrain $\phi$ .

However, there are a few difficulties in applying this conclusion to arbitrary physical variables. First, a distribution of a variable must be bounded both from below and above in order to apply this distribution. Second, the conclusion depends on a choice of a physical variable. This is realized by noting that any physical variable, $\phi$, can be transformed into another, $\varphi$, by assuming a relation, for example,

$$\phi = \varphi^\alpha \tag{3.17}$$

with a constant $\alpha$, and then the distribution is transformed by a relation

$$p(\phi)d\phi = \alpha\varphi^{\alpha-1}p(\varphi^\alpha)d\varphi. \tag{3.18}$$

Here, recall the definition of the distribution, $p$, given by Eq. (3.2b).

Thus, although the system may represent a homogeneous distribution in terms of a particular variable, $\phi$, it is no longer homogeneously distributed in terms of another related variable, $\varphi$. This is a contradiction, because a constant distribution is

obtained for a transformed variable, $\varphi$, when the maximum entropy principle is directly applied to the latter. In this case, the original variable, $\phi$, no longer follows a constant probability by the relation (3.18).

The source of this ambiguity, *i.e.*, dependence of the result from the maximum–entropy principle depending choice on the distribution variable ($\phi$ or $\varphi$), stems from the fact that in translating a discrete expression for entropy (3.8) into a continuous version (3.9), it is assumed that a variable, $\phi$, takes discrete values defined by a constant increment, $\Delta\phi = (\phi_m - \phi_1)/(m-1)$, over an interval, $[\phi_1, \phi_m]$:

$$\phi_i = \phi_1 + (i-1)\Delta\phi \tag{3.19}$$

with $i = 1, \ldots, m$. Then Eq. (3.9) is obtained from the right–hand side of Eq. (3.8) by multiplying $\Delta\phi$ on the latter, and taking a limit of $m \to \infty$. Conversely, Eq. (3.9) can be approximated into the right–hand side of Eq. (3.8) multiplied by $\Delta\phi$ with the discretization (3.19). Note that ambiguity with an arbitrary algebraic factor is also consistent with the nature of the maximum entropy principle, that is valid only in an *asymptotic sense* as already suggested in Sec. 3.3.2.

### 3.4.2 Constraints by Moments

When a variable is constrained by the first $L$ moments, general constraints (3.10) reduce to

$$\int \phi^l p d\phi = C_l \tag{3.20}$$

with $C_l$ a value of the $l$–th moment with $\sigma_l = \phi^l$. The general distribution (3.15) reduces to

$$p = p_0 \exp[-\sum_{l=1}^{L} \lambda_l \phi^l]. \tag{3.21}$$

In particular, when a system is constrained only by a mean (*i.e.*, $L = 1$), the distribution reduces to an exponential distribution

$$p = p_0 \exp[-\lambda_1 \phi], \tag{3.22}$$

*i.e.*, the probability for the first occurrence of an event under a Poisson process, and when a system is also constrained by a variance (*i.e.*, $L = 2$), it reduces to a Gaussian distribution

$$p = p_0 \exp[-\lambda_2(\phi - \langle\phi\rangle)^2] \tag{3.23}$$

with a slight reconfiguration of the general form (3.21). Here, the mean is given by $\langle\phi\rangle$ with $\lambda_1 = 2\lambda_2\langle\phi\rangle$. These results are consistent with our common usage of these distributions: when only a mean (*e.g.*, waiting time) is of concern, an exponential distribution can be adopted. When a variance is further also of interest, a Gaussian distribution would be the most convenient.

### 3.4.3 Gamma Distribution

Note that general distribution forms obtained from the maximum entropy principle, as seen by Eqs. (3.15) and (3.21), always takes an exponential form without any algebraic factor. However, in many atmospheric applications, a distribution with an

algebraic dependence is observed. The best example would be the gamma distribution, which is commonly adopted for representing PSDs in cloud microphysics (*e.g.*, Khain *et al.* 2014). Furthermore, the gamma distribution is a favorable choice for representing various semi–positive definite variables (*e.g.*, water vapour, mixing ratios of various microphysical water species) as argued by Bishop (2016).

The issue may be commented from three perspectives. First, it is important to keep in mind the asymptotic nature of the maximum entropy principle, which is derived under an asymptotic limit of $n \to \infty$. As remarked in Sec. 3.3.2, for this reason, the maximum entropy principle is best understood as a special application of the large-deviation principle, which is designed to express only the dominant exponential dependence, and a remaining subdominant, algebraic dependence is left implicit. From this perspective, the gamma distribution can be interpreted to be a straight generalization of an exponential distribution,

obtained by multiplying an arbitrary, subdominant algebraic factor.

     A way of deriving the gamma distribution, more explicitly, is as a consequence of a transformation of a distribution variable, as discussed in Sec. 3.4.1. By setting a new variable to be $\varphi$, a transformed distribution can contain an algebraic factor as shown by Eq. (3.18). Lastly, it is in fact possible to obtain an algebraic dependence from the maximum entropy principle simply by setting one of the constraints to be $\sigma_l = \log \phi$. The physical meaning of such a constraint is not immediately clear, but it is

370 a question that may be worthy of further investigation. When the constraints are chosen to be $\sigma_1 = \log \phi$ and $\sigma_2 = \phi$, then a gamma distribution is obtained.

## 3.5   Liouville Equation

When a system is governed by an equation of the form (2.2), as introduced in Sec. 2, the Liouville equation

$$\frac{\partial p(\phi)}{\partial t} = -\frac{\partial F p(\phi)}{\partial \phi} \tag{3.24}$$

describes the time evolution of a distribution density, $p(\phi)$, of a given physical variable, $\phi$. Note that so long as the original full physics is exactly described by Eq. (2.2) in a deterministic manner, with $F$ as a continuous function of $\phi$, the associated evolution of the probability distribution density is also exactly described by Eq. (3.24). See Yano and Ouchtar (2017) for a very concise derivation. Generalization for the multiple–variable case is accomplished straightforwardly by replacing $\phi$ and $F$ by vectors. See, *e.g.*, Risken (1984) for systems with stochasticity. More general formulations for the partial–differential

equation (PDE) systems are presented, *e.g.*, as Eq. (15) in Larson (2004), with a full derivation given, *e.g.*, by Pope (1985), and Klimenko and Bilger (1999).

     In spite of its advantage of directly evaluating the time evolution of a given distribution, the Liouville equation is rarely adopted in the studies of atmospheric sciences (*e.g.*, Ehrendorfer 1994a, b, 2006, Yano and Ouchtar 2017, Garret 2019, Hermoso *et al.* 2020), unfortunately, due to its prohibitive computation cost. An efficient computation methodology, which may

make much wider application possible, will be presented in Sec. 5. The result can easily be generalized to a PDE system, as outlined in Sec. 5.3

### 3.6 PSD Equation

A prognostic equation for a PSD, $n(r)$, of hydrometeors can be considered in an analogous manner as the Liouville equation, but it differs in details (*cf.*, Khain *et al.* 2014). A PSD, being considered at a single macroscopic point, is advected, and also a source term, $S$, does not generally take a flux divergence form:

$$\frac{\partial n(r)}{\partial t} + \nabla_h \cdot n(r)\mathbf{u} + \frac{\partial (w - w_t(r))n(r)}{\partial z} = S(r). \tag{3.25}$$

Here, $r$ is a particle size, $\mathbf{u}$ the horizontal velocity, $w$ the vertical velocity, $w_t(r)$ the terminal velocity of the particle with the size $r$. The source term may furthermore be separated into two distinctive processes, collision processes, $S_{col}$, and non–collision processes, $S_{loc}$, namely the growth and the reduction (*e.g.*, evaporation) processes of the individual particles:

$$S(r) = S_{col}(r) + S_{loc}(r). \tag{3.26a}$$

The collision term may take a form

$$S_{col}(m) = \frac{1}{2}\int_0^m n(m')n(m-m')K(m', m-m')dm' - \int_0^\infty n(m)n(m')K(m, m')dm', \tag{3.26b}$$

setting the particle masses to be $m = m(r)$ and $m' = m(r')$. Here, $K(m, m')$ is the collision kernel between the particles of the masses, $m$ and $m'$. Also note $S_{col}(r)dr = S_{col}(m)dm$. The first and the second terms on the right hand side above, respectively, represent gain and loss for a given particle size. The collision process prevents Eq. (3.25) from reducing it to the Liouville equation, because this process makes the source, $F$, discontinuous as a function of the particle size. Nevertheless, Eq. (3.25) can be treated in an analogous manner as the Liouville equation (3.24) by replacing the right hand side of Eq. (3.24), $-(\partial p(\phi)F/\partial\phi)$, by the tendency of PSD as given in Eq. (3.25). Furthermore, the PSD equation (3.25) reduces to the Liouville equation, when the advection and the collision effects can be neglected, as seen in Sec. 6 later.

## 4 Applications to Atmospheric Processes

In applying the statistical principles discussed in the last section to atmospheric problems, some additional considerations are required due to differences from typical statistical problems. This section discusses those differences. Our discussions may be rather abstract and philosophical. However, we believe that they provide insights to critical issues of atmospheric modelling, which are often overlooked. Our discussions lead to a principle for choosing distribution constraints in atmospheric problems, as required for the maximum–entropy principle, which we call the *output–constrained distribution principle*.

There are, namely, three important differences in the atmospheric applications from standard statistical applications. Those are discussed in the following three subsections.

### 4.1 Static or Non–Static, Diagnostic or Prognostic

First, statistics, or more precisely, mathematical statistics is fundamentally *static* and *diagnostic*: methodologies of statistics and probability (*e.g.*, hypothesis testing, probabilities with a binomial system) as found in standard textbooks (*e.g.*, Feller

1968, Wonnacott and Wonnacott 1969, Jaynes 2003, Gregory 2005) do not involve any time–dependent problems. Extensive time–dependent statistical models in the literature belong rather to the statistical mechanics and stochastic modeling than to the mathematical statistics. This fundamentally static nature of the statistics is also reflected upon more modern statistic theories: for example a standard textbook on *deep learning* (Goodfellow *et al.* 2016) does not address any time dependent problems.

Very symbolically, the notion of "updating" a prior distribution in Bayesian probability theories (*e.g.*, Bernardo and Smith 1997), contrary to its connotation, does not involve a concept of time, but just an update of *our knowledge* under a fixed time.

This is a rather stark contrast to the atmospheric system, which continuously evolves with time: we are inherently interested in forecasts. Allegorically speaking, there is no time to *update* the *priors* with the atmospheric system, because as soon as information is updated, the original prior is already obsolete, because the system itself has changed. Still allegorically speaking,

the best we can do is to *update* (in meteorological sense, but *not* in statistical sense) the priors themselves with time. In other words, in describing the atmospheric processes, the key issue is to predict the time evolution of the probability and the statistics: atmospheric problems are fundamentally *non–static* and *prognostic*.

The data assimilation problem falls into a middle of the two. As in any other atmospheric problems, the prediction of the evolution of the data uncertainty is a key aspect of data assimilation. At the same time, the *statistical update* of data

by incorporating observational information is another key aspect of the data assimilation. In the present study, the focus is exclusively on the first aspect.

## 4.2    Output–Constrained Distribution Principle

Second, in many statistical applications as well as in standard equilibrium statistical mechanics, as summarized by Jayens (1978), a final aim is to know a distribution of a given variable. For this goal, the integrated quantities are inputs to a problem

that constrains a distribution. Under these constraints, we define the most–likely distribution from the maximum–entropy principle.

However, in atmospheric modelling, knowing a distribution itself, though it may be of theoretical interest, is not an ultimate aim. It is merely a means of obtaining certain integrated quantities (*e.g.*, microphysical tendencies, grid–box averaged quantities, standard–deviation error measures) for a modelling purpose. For this goal, a precise form of a distribution is not

of interest, but it must be just accurate enough for providing these required final outputs. This is a very different problem compared to those in standard equilibrium statistical mechanics. Here, we must clearly recognize that these are two different problems: although a more accurate distribution may help to evaluate the required statistical quantities more accurately, there should be a way of making the latter more accurate without making the former more accurate than necessary.

This observation leads to an interesting possibility for constructing a prescribed PDF (DDF) form in atmospheric–science

applications: take the *necessary outputs* as constraints rather than the available inputs. Thus, for example, if a purpose is to know a mean value (*e.g.*, a waiting time), take an exponential distribution (Eq. 3.22). If a purpose is to know a variance (*e.g.*, a standard deviation error in temperature measurements), take a Gaussian distribution (Eq. 3.23). We propose to call them *output–constrained distributions*. Note that an assumed PDF obtained under this principle may provide a poor fit to the actual distribution. Our basic argument here is that, nevertheless, an assumed PDF will work reasonably well for the purpose

of estimating the required output values (*i.e.*, "constraints"), because the given distribution is obtained from the maximum–entropy principle by taking those required outputs as the constraints.

The proposed re–interpretation is consistent with a basic requirement for an assumed PDF form: if we need to fit a PDF to $L$–statistical variables, $L$–parameters must be introduced. This is not the case with a popular approach of introducing an assumed double Gaussian distribution for the sake of representing a skewness of an actual distribution (*e.g.*, Larson *et al.* 2002,
Fitch 2019, Naumann *et al.* 2013): a number of parameters of an assumed PDF becomes greater than that of required outputs. For example, with a single variable, a double Gaussian distribution introduces five parameters when only three outputs are required (mean, variance, skewness). In contrast, the proposed principle suggests how to choose a distribution that contains the minimum number of parameters compatible with the required number of outputs. Here, we invoke the maximum–entropy principle for this purpose.

A current standard approach of updating the PDF parameters is from some moments (*e.g.*, mean, variance) of variables. The output-constrained distribution principle dictates to update those parameters by using the actual output variables that are required for a host model or quantities that crucial for predicting the evolution of the system in concern. This rather philosophical statement poses an important practical question of, for example, whether it is optimal to choose the radar reflectivity as a third constraint in bulk microphysics.

Here, the notion of "output variables that are required for a host model" is more specifically relevant for the subgrid–scale distribution for a parameterzation. Recall that the goal of a parameterization is *not* providing every detail of subgrid–scale processes, *but only* the so–called apparent sources, $Q_1$ and $Q_2$, *i.e.*, tendencies of the temperature and moisture due to the subgrid–scale processes (*cf.*, Yaani *et al.* 1973), and only as grid–scale averages. All the other details are only for a purpose of a consistent calculation of the subgrid–scale processes. In case of the clouds microphysics with explicit cloud modeling (thus
the cloud processes themselves are not "parameterized"), certain variables must be passed over to different components of the model, that plays a role of "host model" in this context. For example, the mixing ratios, $q_c$ and $q_r$, of clouds and rain must be counted for an accurate definition of the buoyancy in the momentum equation. Some radiation schemes require inputs of mean radius, $r_c$ and $r_p$, of the cloud and rain droplets, although those are typically *not* prognostic variables of the cloud microphysics. Those variables are considered to be the "necessary variables (outputs) for the host model". Thus, especially in the context of
the cloud microphysics, the "necessary variables (outputs)" should be clearly distinguished from the prognostic variables in a cloud microphysics. The case of data assimilation is more subtle, because there is neither a host model nor another model components to which information must be passed around. Yet, for the operational purposes, we are not interested to know a full shape of a probability distribution of a variable in order to quantify the uncertainty. In traditional assimilation formulations, we merely asks for the standard–deviation errors/uncertainties of variables: those are considered the "necessary outputs" for the
data assimilation.

### 4.3 Availability of Input Data

A third major difference of atmospheric problems from standard statistical problems is the availability of input data (*e.g.*, initial constraints). In the latter, we assume a situation that available input data (information) is rather limited. For example, the

Maxwell-Boltzmann distribution is derived by assuming that only the total energy is known. It is rarely asked how to obtain more information so that, for example, a higher-order correction to the Maxwell-Boltzmann distribution can be obtained. A limited amount of information is the given starting point.

On the other hand, in atmospheric problems, available information is rather unlimited, or at least, we believe that we can obtain more data either by modelling or observation if necessary. In other words, the input data is rather unconstrained. For the subgrid–scale distributions, more explicit models such as cloud–resolving models (CRMs) or large–eddy simulations (LESs) can be used at will for any detailed simulations for the subgrid–scale processes of concern, especially with enhancements of computing power. Such an abundance of information tends to obscure the basic idea of statistical description. The same issue can also be identified from a perspective of the assumed–PDF (DDF) approach: we can take as many number of moments as required in principle. The only issue is a computational cost and accuracy benefit.

In other words, in principle, the number of available inputs is less limited in atmospheric problems. Thus, if this available information is simply adopted as constraints for defining the most–likely distribution under the maximum–entropy principle, the number of distribution parameters can arbitrary be increased to get as accurate a distribution as desired. This considera-tion also strengthens the argument of the last subsection. In the context of the assumed–PDF (DDF) approach, a number of "constraints" (*e.g.*, moments) must be decided in such manner that required outputs can be evaluated in sufficiently accurate manner. In other words, the problem must be constrained by required outputs rather than available inputs: that is the essence of the output–constrained distribution principle.

## 4.4 Validation: Diffusion Problem

Over the last two subsections (Secs. 4.2–4.3), arguments have been developed for re-interpreting the maximum–entropy prin-ciple in such a manner that the actual variables required as outputs are to be adopted as "constraints" for determining a distribution. Recall that the output–constrained distribution principle is proposed merely as a guiding principle for choosing an assumed form for distributions, thus we do not expect that those chosen distribution forms can make any perfect predictions. Nevertheless, it would be helpful to quantify the degrees of the accuracy of predictions: that is the purpose of this subsection.

Thus, this subsection tests the proposed output–constrained distribution principle by taking, as an example, a one-dimensional diffusion equation:

$$\left( \frac{\partial}{\partial t} - \frac{\partial^2}{\partial x^2} \right) p = 0 \tag{4.1}$$

with a diffusion coefficient set to unity for simplicity. When a system evolves purely under white noise forcing, as the case of the Brownian motion, the prognostic equation for the distribution, $p$, reduces to this form. Also note that Eq. (4.1) is a special case of the Fokker–Planck equation.

Let us assume that we are interested in predicting only a mean and a variance for the position, $x$, of the distribution, $p$. In other words, the required outputs for our problem are only a mean, $\langle x \rangle$, and a variance, $\langle (x - \langle x \rangle)^2 \rangle$. In this case, the output–constrained distribution principle suggests that it suffices to take a Gaussian distribution, say:

$$p(x,t) = p_0(t) e^{-\lambda_2(t)(x - \langle x \rangle(t))^2}. \tag{4.2}$$

Note that in this particular case, the adopted distribution form also corresponds to an exact solution of the system (4.1).

The time dependence of the parameters, $\lambda_2(t)$ and $\langle x \rangle(t)$, introduced in the above solution (4.2) can be derived by directly substituting (4.2) into Eq. (4.1): *cf.*, Sec. 5.6.1. When the initial conditions are given by

$$\langle x \rangle|_{t=0} = \langle x \rangle_0, \tag{4.3a}$$

$$\langle (x - \langle x \rangle)^2 \rangle|_{t=0} = \frac{1}{2\lambda^*}, \tag{4.3b}$$

the time evolution of the distribution (4.2) is given by

$$p(x,t) = \frac{1}{(1 + 4\lambda^* t)^{1/2}} \left( \frac{\lambda^*}{\pi} \right)^{1/2} \exp \left[ -\frac{\lambda^* (x - \langle x \rangle_0)^2}{1 + 4\lambda^* t} \right]. \tag{4.4}$$

Here, the distribution, $p$, is also normalized so that its integral over the whole domain becomes unity, and the time evolution of the mean and the variance is given by

$$\langle x \rangle = \langle x \rangle_0, \tag{4.5a}$$

$$\langle (x - \langle x \rangle)^2 \rangle = \frac{1 + 4\lambda^* t}{2\lambda^*}$$

$$= \frac{1}{2\lambda^*} + 2t. \tag{4.5b}$$

Note that the mean is a constant with time in the diffusion problem if the initial condition is Gaussian, whereas the variance increases linearly with time.

The basic idea behind the output–constrained distribution principle is to define a PDF in such manner that the required output variables (mean and variance, here) can be evaluated most effectively with the minimum possible parameters. Fitting an actual distribution under an assumed PDF form is not a goal. To test the working of this principle, we consider the two examples, in which we *set* both the initial mean and variance of the output–constrained distribution (4.4) equal to those of an actual initial distribution, and compare the evolutions of means and variances of both distributions.

### 4.4.1 Example 1: Double Gaussian Distribution

As a first example, we take an initial distribution consisting of two Gaussian distributions:

$$p(x, t = 0) = \alpha_1 \left( \frac{\mu_1}{\pi} \right)^{1/2} e^{-\mu_1 (x - x_1)^2} + \alpha_2 \left( \frac{\mu_2}{\pi} \right)^{1/2} e^{-\mu_2 (x - x_2)^2}. \tag{4.6}$$

Here, two Gaussians are centered as $x = x_1$ and $x_2$, respectively. By a normalization condition, we may set

$$\alpha_1 + \alpha_2 = 1. \tag{4.7a}$$

We may further set

$$\alpha_1 x_1 + \alpha_2 x_2 = 0. \tag{4.7b}$$

so that the initial mean is $\langle x \rangle = 0$.

It is immediately seen that the time evolution of this system is given by

$$545 \quad p(x,t) = \frac{\alpha_1}{(1+4\mu_1 t)^{1/2}} \left(\frac{\mu_1}{\pi}\right)^{1/2} e^{-\mu_1(x-x_1)^2/(1+4\mu_1 t)} + \frac{\alpha_2}{(1+4\mu_2 t)^{1/2}} \left(\frac{\mu_2}{\pi}\right)^{1/2} e^{-\mu_2(x-x_2)^2/(1+4\mu_2 t)}. \tag{4.8}$$

Here, the mean remains a constant with time, thus $\langle x \rangle = 0$, and evolution of the variance is given by

$$\begin{aligned}
\langle x^2 \rangle &= \frac{\alpha_1}{2}\frac{1+4\mu_1 t}{\mu_1} + \frac{\alpha_2}{2}\frac{1+4\mu_2 t}{\mu_2} \\
&= \frac{1}{2}\left(\frac{\alpha_1}{\mu_1} + \frac{\alpha_2}{\mu_2}\right) + 2t.
\end{aligned} \tag{4.9}$$

We can evaluate the statistical evolution of this system by an output–constrained distribution (*i.e.*, a single Gaussian distribution) by setting the initial mean and variance identical. Thus, we obtain $\langle x \rangle_0 = 0$ and

$$\lambda^* = \left(\frac{\alpha_1}{\mu_1} + \frac{\alpha_2}{\mu_2}\right)^{-1} \tag{4.10}$$

in Eq. (4.4). Substitution of (4.10) into Eq. (4.5b) shows that this single Gaussian model can predict the time evolution of both the mean (rather trivially) and the variance of a two-Gaussian system perfectly.

Note that in this case, a single Gaussian hardly fits a double Gaussian distribution in any good approximation, especially when two Gaussians are well separated from each other. However, if our interest is merely to predict a variance, then in this example a single Gaussian approximation perfectly serves the purpose, being consistent with a proposed re-interpretation of the maximum–entropy principle. This example may appear to be rather too special and artificial. Nevertheless, it makes the case well that for predicting a limited number of statistical quantities satisfactory, accurately predicting the evolution of the whole distribution is not necessarily a requirement.

### 4.4.2 Example 2 : A Skewed Gaussian Distribution

The second example is a skewed initial distribution given by

$$p(x, t=0) = \left(\frac{\lambda}{\pi}\right)^{1/2} (1+\alpha x) e^{-\lambda x^2}. \tag{4.11}$$

Here, a constant parameter, $\alpha$, controls the skewness of this distribution. This example examines how well an assumed Gaussian distribution predicts the statistics when an actual distribution is not Gaussian.

The time evolution of this system is solved by, for example, a Fourier transform method, as summarized in the Appendix A. The final answer is:

$$p(x,t) = \frac{1}{(1+4\lambda t)^{1/2}}\left[1 + \frac{\alpha x}{(1+4\lambda t]^2}\right]\left(\frac{\lambda}{\pi}\right)^{1/2} \exp\left[-\frac{\lambda x^2}{1+4\lambda t}\right]. \tag{4.12}$$

From this solution, the time evolution of mean and variance is readily evaluated as:

$$\langle x \rangle = \frac{\alpha}{2\lambda(1+4\lambda t)}, \tag{4.13a}$$

$$\langle (x - \langle x \rangle)^2 \rangle = \frac{1+4\lambda t}{2\lambda}\left[1 - \frac{\alpha^2}{2\lambda(1+4\lambda t)^3}\right]. \tag{4.13b}$$

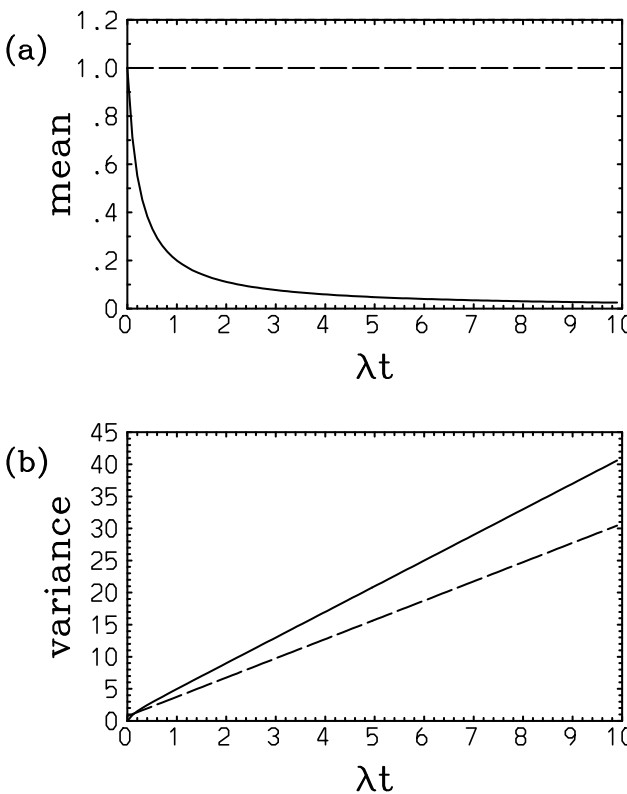

**Figure 1.** Plots of (a) $2\lambda\langle x\rangle/\alpha$ and (b) $2\lambda\langle(x - \langle x\rangle)^2\rangle$ with an initial skewed Gaussian distribution (4.11) with $\alpha/\lambda^{1/2} = 0.5$: exact (solid) and with a single Gaussian approximation (long dash). Note that in (a), although the exact mean (solid) decreases to zero with time, the assumed Gaussian solution (long dash) totally fails to predict this tendency. Nevertheless, the prediction of the variance (b) with the assumed Gaussian (long dash) is still qualitatively correct, even though a tendency of underestimation against the exact value (solid) may be noted.

"Fit" to this problem under the assumed single Gaussian distribution is given by

$$\langle x\rangle_0 = \frac{\alpha}{2\lambda}, \tag{4.14a}$$

$$\lambda^* = \lambda\left(1 - \frac{\alpha^2}{2\lambda}\right)^{-1}. \tag{4.14b}$$

In this case, the assumed PDF fails to predict a gradual shift of the mean to the origin from an initial position (*cf.*, Eq. 4.13a and Eq. 4.14a: Fig. 1(a)). This discrepancy is hardly a surprise, because an assumed–PDF evolution is *not* expected to predict an evolution of an actual distribution in any perfect manner. However, the growth of deviation of the variance with the assumed single–Gaussian (Eq. 4.14b: long dash) from the real value (Eq. 4.13b: solid) is relatively slow (Fig. 1(b)), and an underestimate is only 25% even at $\lambda t = 10$ with a relatively large nondimensional skewness parameter (*i.e.*, $\alpha/\lambda^{1/2} = 0.5$) assumed.

## 5 Liouville Equation Constrained by an Assumed PDF

As emphasized in the last section, in principle, a full physical system, as given by Eq. (2.2) in the present study, is known in atmospheric problems, thus it also prompts us to exploit the Liouville equation (3.24) in predicting the time evolution of a distribution. The next question is how to exploit it efficiently. The output–constrained distribution principle introduced in the last section provides a half of the answer to this question: adopt a distribution form that is defined from the maximum-entropy principle, but taking the required outputs for the host model as the "constraints", $\langle \sigma_l \rangle$ ($l = 1, \ldots, n$). As a result a

distribution with a finite number of parameters, $\{\lambda_i\}$, is obtained. Thus, the problem of defining the time evolution of a continuous distribution function reduces to that of describing the time evolution of a finite number of parameters $\{\lambda_i(t)\}$. The remaining half of the question is how to actually calculate the time evolution of $\{\lambda_i\}$ directly by a set of ordinary differential equations. The present section addresses this remaining half. Importantly, the formulation presented in this section does not reply on the output–constrained distribution (maximum entropy) principle. Rather, it can be applied to any form of assumed

PDFs.

The first key step required for this procedure is, as just suggested, to replace the time derivative, $\partial p / \partial t$, of the distribution by those, $\dot{\lambda}_l$, of the PDF parameters. This is simply accomplished by taking a chain rule to the time derivative, noting the time dependence in the distribution (3.2a) arises solely from the parameters, $\{\lambda_i\}$:

$$\frac{\partial p}{\partial t} = \sum_{i=0}^{N} \frac{\partial p}{\partial \lambda_i} \dot{\lambda}_i. \tag{5.1}$$

A specific example of this procedure is given by Eq. (5.13) below, which can also be performed either more directly or using the formula (5.1) explicitly.

### 5.1 General Formulation

This subsection derives a prognostic set of equations for the PDF parameters $\{\lambda_l\}$ in a general manner. The derivation is repeated in Sec. 5.5 by taking the exponential distribution as an example so that the basic idea can be seen in a more concrete

manner.

We assume a distribution in a general form (3.2a) assuming that a distribution is defined by $N + 1$ free parameters $\lambda_i$ ($i = 0, \cdots, N$). Here, we assume that $\lambda_0$ is a constant factor that is required for normalizing the distribution, as already remarked in introducing the assumed distribution in a general form by Eq. (3.2a), thus

$$p \propto \lambda_0, \tag{5.2a}$$

or

$$\frac{\partial p}{\partial \lambda_0} = \frac{p}{\lambda_0}. \tag{5.2b}$$

Substitution of Eq. (5.2b) into Eq. (5.1) leads to:

$$\frac{\partial p}{\partial t} = p \frac{\dot{\lambda}_0}{\lambda_0} + \sum_{i=1}^{N} \frac{\partial p}{\partial \lambda_i} \dot{\lambda}_i. \tag{5.3}$$

The replacement of the left–hand side of Eq. (5.3) by that of the right–hand side is the key step of reducing the problem of the evolution of a distribution, $p$, as a whole into that of the fixed number of parameters, $\lambda_i$ $(i = 1, \ldots, N)$.

The time evolution of this distribution is constrained in two ways: first, by a normalization condition (3.11) and second, by the Liouville equation (3.24). The normalization condition (3.11) can also be cast into a prognostic form by taking the time derivative:

$$\frac{\partial}{\partial t} \int p d\phi = 0 \tag{5.4}$$

with the normalization (3.11) introduced as an initial condition. Note that in case with PSDs, the right–hand side of Eq. (3.11) must be replaced by $\dot{C}_0$ with $C_0$ stands for the total particle number density. Thus, the following reduction must also be modified accordingly.

When the integral range is fixed with time, the time derivative can be moved inside the integral, operating only on the distribution, $p$, in Eq. (5.4). Further substituting the formula for the time derivative (5.3), we obtain:

$$\frac{\dot{\lambda}_0}{\lambda_0} = -\sum_{i=1}^{N} \left[ \int \frac{\partial p}{\partial \lambda_i} d\phi \right] \dot{\lambda}_i. \tag{5.5}$$

The Liouville equation (3.24) also reduces from Eq. (5.3) to:

$$p \frac{\dot{\lambda}_0}{\lambda_0} + \sum_{i=1}^{N} \frac{\partial p}{\partial \lambda_i} \dot{\lambda}_i + \frac{\partial}{\partial \phi}(pF) = 0, \tag{5.6}$$

and substituting Eq. (5.5) furthermore to the above,

$$\sum_{i=1}^{N} \left[ \frac{\partial p}{\partial \lambda_i} - p \int \frac{\partial p}{\partial \lambda_i} d\phi \right] \dot{\lambda}_i + \frac{\partial}{\partial \phi}(pF) = 0. \tag{5.7a}$$

Eq. (5.7a) is the key result of the present study, because it constitutes a prognostic equation for evaluating the time evolution of $\{\lambda_i\}$. Note that when the same procedure is applied to the diffusion equation (4.1), by replacing $\partial pF / \partial \phi$ by $-\partial^2 p / \partial \phi^2$, it reduces into a set of ordinary differential equations for $\lambda_2$ and $\langle x \rangle$, as shown in Sec. 5.4.1. As in the case with the diffusion equation, more generally, when the assumed PDF form constitutes an exact solution of a given system, Eq. (5.7a) is separated out into the $\phi$– and $\lambda_i$–dependencies, and the latter dependencies can be solved separately, independent of the distribution variable, $\phi$. This point can be understood directly from the fact that Eq. (5.7a) is equivalent to the original Liouville equation (3.24) under the given assumed PDF form.

However, because the assumed PDF generally constitutes merely an approximation of a true distribution, $\phi$–dependence in Eq. (5.7a) cannot be separated out in general case, thus this equation cannot be solved in any consistent manner merely in terms of the assumed–PDF parameters, $\{\lambda_i\}$. The consequence of the approximate nature of the assumed PDF in Eq. (5.7a) is more explicitly seen by substituting the form (3.2c) into Eq. (3.24):

$$\sum_{i=1}^{N} \left[ \frac{\partial p}{\partial \lambda_i} - p \int \frac{\partial p}{\partial \lambda_i} d\phi \right] \dot{\lambda}_i + \frac{\partial}{\partial \phi}(pF) = \mathcal{E}. \tag{5.7b}$$

with $\mathcal{E}$ suggesting a possible error. We should well keep in mind that this very last fact does not change regardless of whatever manner we attempt to predict the evolution of a distribution by an assumed PDF. In other words, Eq. (5.7a) itself is not defected, but the difficulty here is a simple consequence of the assumed PDF approach, which attempts to solve the evolutions of distributions by assuming the forms that are not actual solutions.

Thus, the next goal is to derive a closed set of equations, not depending on $\phi$, from Eq. (5.7a) in order to solve a set of distribution parameters, $\{\lambda_i\}$, in a consistent manner. For this purpose, we need to remove the $\phi$–dependence from Eq. (5.7a). An only option that we can see is to simply integrate it over $\phi$. Here, a goal is to obtain $N$ differential equations for $\lambda_i$ ($i = 1, \ldots, N$) by removing $\phi$–dependence. For this purpose, we apply a set of weights, $\sigma_l$ ($l = 1, \cdots, N$), on Eq. (5.7a) and integrate them over $\phi$. Here, the weight, $\sigma_l$, is an unspecified function of $\phi$, but independent of $\{\lambda_i\}$. After integration in $\phi$, we obtain

$$\sum_{i=1}^{N} \left[ \int \sigma_l \frac{\partial p}{\partial \lambda_i} d\phi - \int \sigma_l p d\phi \int \frac{\partial p}{\partial \lambda_i} d\phi \right] \dot{\lambda}_i = - \int \sigma_l \frac{\partial}{\partial \phi} (pF) d\phi \tag{5.8a}$$

for $l = 1, \cdots, N$. As a result, we obtain $N$ ordinary differential equations for $N$ unknowns. The set of equations (5.8a) is linear in terms of $\dot{\lambda}_i$, thus it can be inverted in principle, and the tendencies, $\dot{\lambda}_i$, can explicitly be evaluated. Here, keep in mind that Eq. (5.8a) is valid only approximately, thus it may be more emphatically stated as:

$$\sum_{i=1}^{N} \left[ \int \sigma_l \frac{\partial p}{\partial \lambda_i} d\phi - \int \sigma_l p d\phi \int \frac{\partial p}{\partial \lambda_i} d\phi \right] \dot{\lambda}_i = - \int \sigma_l \frac{\partial}{\partial \phi} (pF) d\phi + \mathcal{E}_l \tag{5.8b}$$

with $\mathcal{E}_l$ suggesting an associated error. Note further that the right hand side of Eq. (5.8a) can be re–written as

$$\int \sigma_l \frac{\partial}{\partial \phi} (pF) d\phi = - \int pF \frac{\partial \sigma_l}{\partial \phi} d\phi \tag{5.9}$$

by an integration by parts, assuming that $pF$ vanishes at the edges of the integral range. Realize that the key step introduced in the formulation here is to prognose the PDF parameters, $\{\langle \lambda_l \rangle(t)\}$, by Eq. (5.8a). In this manner, we circumvent the principal difficulty of the current assumed–PDF approaches of inverting the relations (3.4) into the form (3.3). Now, the major remaining open question with this procedure is the choice of the weights, $\{\sigma_l\}$, that is the issue to be addressed next.

## 5.2 Choice of the Weights, $\{\sigma_l\}$

Here, the most appropriate choice of the weights, $\{\sigma_l\}$, becomes immediately clear by noting that the left–hand side of Eq. (5.8a) corresponds to a temporal tendency, $d\langle \sigma_l \rangle/dt$, of the "averaged" weight:

$$\frac{d}{dt} \langle \sigma_l \rangle = \int \sigma_l \frac{\partial p}{\partial t} d\phi = \sum_{i=0}^{N} \left[ \int \sigma_l \frac{\partial p}{\partial \lambda_i} d\phi \right] \dot{\lambda}_i, \tag{5.10a}$$

where the last expression reduces to the left-hand side of Eq. (5.8a) with the help of Eq. (5.5). Thus, symbolically, Eq. (5.8a) is equivalent to:

$$\frac{d}{dt} \langle \sigma_l \rangle = \langle F_{\sigma_l} \rangle \tag{5.10b}$$

with $F_{\sigma_l}$ the source term that defines the tendency of $\sigma_l$. By the deduction from Eq. (5.10a), we can conclude that Eq. (5.8a) predicts the time evolution of $\langle \sigma_l \rangle$, as given by Eq. (5.10b), where

$$\langle \sigma_l \rangle = \int p \sigma_l d\phi. \tag{5.10c}$$

It also follows that if $\{\langle \sigma_l \rangle\}$ are chosen as the outputs to be used in the host model, by following the output–constrained distribution principle (*cf.*, Sec. 4.2), Eq. (5.8a) predicts those required outpus consistently under a given assumed PDF (DDF), being equivalent for solving Eq. (5.10b). Thus, we choose $\{\sigma_l\}$ the same as in the constraints (3.10) with $G_l = p\sigma_l$ and $L = N$.

A standard choice following the assumed–PDF (DDF) approach is to set $\sigma_l = \phi^l$. This procedure is equivalent to time integrating the moments for predicting $\{\lambda_i\}$. Eq. (5.8a) or (5.10b) further reduces to a diagnostic method based on moments typically adopted in the subgrid–scale assumed PDF formulations, when $\{\langle \sigma_l \rangle\}$ are taken as moments, and also a diagnostic limit is taken. As already emphasized in introducing the governing equation (2.2) of the system in Sec. 2, the source term, $F$, includes all the physics associated with a variable, $\phi$. A multi–variable extension is also straightforward. Thus, in principle, this formulation can be applied to any assumed–PDF approaches, including those in cloud microphysics and data assimilation.

In subsequent subsections, more specific versions of Eq. (5.8a) for various assumed PDF forms are presented, as a demonstration that this general formulation can actually be used. These results can readily be used as receipts for applying the formulation to any physical problems under given assumed PDF forms, once the source term, $F$, is specified. However, as a detour, we first discuss the generalization of the introduced formulation to the PDE system in the next subsection, and discuss its link to the existing assumed–PDF approaches and the bulk microphysics in Sec. 5.4.

## 5.3 Generalization to the PDE system (2.1)

The discussion of the last subsection suggests that the derivation of the prognostic equations for the assumed–PDF parameters in the one–dimensional dynamical system (2.2) can generalized into partial differential equation (PDE) systems, described by Eq. (2.1), in a relatively straightforward manner.

First note that in a PDE system, the time derivative on the right–hand side of Eq. (5.10a) is replaced by a partial time derivative:

$$\frac{\partial}{\partial t} \langle \sigma_l \rangle = \int \sigma_l \frac{\partial p}{\partial t} d\phi = \sum_{i=0}^{N} \left[ \int \sigma_l \frac{\partial p}{\partial \lambda_i} d\phi \right] \dot{\lambda}_i. \tag{5.11a}$$

Note next that a time–evolution equation for $\sigma_l$ can be derived from the basic governing equation (2.1) by taking a chain rule:

$$\frac{\partial \sigma_l}{\partial t} = \frac{\partial \sigma_l}{\partial \varphi} \frac{\partial \varphi}{\partial t} = \frac{\partial \sigma_l}{\partial \varphi} (-\mathbf{v} \cdot \nabla \phi + F).$$

Thus,

$$F_{\sigma_l} = \frac{\partial \sigma_l}{\partial \varphi} (-\mathbf{v} \cdot \nabla \phi + F), \tag{5.11b}$$

and with the help of Eq. (5.11b), the PDE version of Eq. (5.10b) becomes:

$$\frac{\partial}{\partial t} \langle \sigma_l \rangle = \langle F_{\sigma_l} \rangle. \tag{5.11c}$$

By combining Eqs. (5.11a) and (5.11c), the prognostic set of equations for $\dot{\lambda}_i$ is given by:

$$\frac{\partial}{\partial t}\langle\sigma_l\rangle = \sum_{i=0}^{N}\left[\int \sigma_l \frac{\partial p}{\partial \lambda_i}d\phi\right]\dot{\lambda}_i = \langle F_{\sigma_l}\rangle. \tag{5.11d}$$

Keep in mind that $\dot{\lambda}_i$ in the above designates the partial time derivative. The generality of the final result (5.11d) would not be necessary to emphasize.

## 5.4 Link to the Existing Assumed–PDF Approaches and the Bulk Microphysics

The formulation presented in the last three subsections constitutes a generalization of the existing assumed–PDF approaches in the following manners. Recall that with the help of Eq. (5.10a), the general equation (5.8a) can more symbolically be written as Eq. (5.10b). Here, keep in mind that a spatial dependence of variables with Eq. (2.1) can also be taken into account by simply replacing the time derivative in the left–hand side by a partial derivative (*cf.*, Eq. 5.11c). Also keep in mind the possibilities of generalizations into the systems with multiple variables (*cf.*, Yano 2024), though they remain implicit here. In this manner, Eq. (5.10b) constitutes a general form of governing equations considered in the existing assumed–PDF approaches, as more specifically presented by *e.g.*, Eqs. (3.1)–(3.10) in Larson (2022).

The first generalization to be noticed is the fact that $\langle\sigma_l\rangle$ can be of any output variables as required by a host model so long as $\sigma_l$ are properly defined as functions of dependent variables. Also note that the integral range in definition (Eq. 3.10c) can be taken in any manner, although such a full generalization itself is left for future studies (*cf.*, Yano 2024). Thus, for example, a cloud fraction can also be introduced as one of the output variables of the form, $\langle\sigma_l\rangle$, under this definition. On the other hand, the existing assumed–PDF approaches, rather arbitrarily, restrict these statistical variables, $\langle\sigma_l\rangle$, to be moments (*e.g.*, Golaz *et al.* 2002). The formulation introduced here demonstrates that this restriction is not necessary, but a very wide range of choice can be taken for $\langle\sigma_l\rangle$.

Furthermore, the existing assumed–PDF approaches perform time integrals of the statistical variables, $\langle\sigma_l\rangle$, by Eq. (5.10b). Note that with some variables, Eq. (5.10b) is solved diagnostically by setting the left–hand side to be zero (*cf.*, Larson 2017). After updating $\langle\sigma_l\rangle$, the PDF parameters, $\{\lambda_l\}$, are diagnosed from the given set of $\{\langle\sigma_l\rangle\}$ from the relations:

$$\langle\sigma_j\rangle = \langle\sigma_j\rangle(\{\lambda_l\}) \tag{5.12}$$

for $j = 1,\ldots N$. As already discussed in Sec. 3.2.1, the inversion of $\{\langle\sigma_l\rangle\}$ to $\{\lambda_l\}$ is not always easy.

The introduced general formulation, in turn, shows that the left–hand side of Eq. (5.10b) or (5.11c) can be replaced by the left–hand side of Eq. (5.8a) or (5.11a), thus the problem directly reduces to the prognoses of the PDF parameters of a given distribution. As a result, there is no longer a need for performing a cumbersome inversion. This modification much facilitates the computational procedure. Note especially that when the same set of moments are taken for $\{\langle\sigma_l\rangle\}$, with identical assumed PDF forms, the present formulation is perfectly equivalent to an existing assumed–PDF model apart from the fact that the PDF parameters, $\{\lambda_l\}$, are directly predicted instead of the moment set, $\{\langle\sigma_l\rangle\}$.

Another strength of the present formulation is in more explicitly showing that the right–hand side of Eq. (5.10b) or (5.11c) can also be totally expressed in terms of the PDF parameters. As a result, there is no longer a need to introduce further closures

to the assumed–PDF formulation, as also noted by Golaz *et al.* (2002). However, this last rather obvious point is not always recognized, and some assumed–PDF approaches often introduce additional closures to close their formulations (*e.g.*, Fitch 2019, Naumann *et al.* 2013).

The standard formulations in the bulk microphysics (*cf.*, Milbrandt and Yau 2015 and the references therein) are to adopt the mixing ratio, $q$, the total number density, $N_T$, and the radar reflectivity, $Z$, as the prognostic variables, with the order to be adopted with decreasing truncations. Under the present formulation, neglecting multiplication factors, $q$ and $Z$ correspond to setting $\sigma_1 = r^3$ and $\sigma_2 = r^6$, respectively, whereas $N_T$ is predicted in a standalone manner by separating out the number density, $n$, into the two components by setting, $n = N_T p$. More or less the same remarks follow for them with further flexibilities

in the formulation by re-writing it in terms of a general form of Eq. (5.10b) or (5.11c). Probably, most importantly, the choice of the radar reflectivity, $Z$, as a "constraint" can be questioned from a point of view of the output–constrained distribution principle (*cf.*, Sec. 4.2): though the reflectivity, $Z$, may be a useful variable to compare with the observation, it is not directly required in any microphysical tendencies within a model.[3]

## 5.5   Application (1): Exponential Distribution

In this subsection, we repeat the general derivation presented in Sec. 5.1 by taking the exponential distribution (Eq. 3.22 with $\lambda_0 = p_0$) as an example. Here, we immediately obtain $\partial p / \partial \lambda_0 = p / \lambda_0$, $\partial p / \partial \lambda_1 = -p\phi$, and

$$\frac{\partial p}{\partial t} = p \frac{\dot{\lambda}_0}{\lambda_0} - p\phi \dot{\lambda}_1. \tag{5.13}$$

The normalization condition (5.4) is obtained by integrating the above equation with respect to $\phi$:

$$\frac{\dot{\lambda}_0}{\lambda_0} \int p \, d\phi - \dot{\lambda}_1 \int p\phi \, d\phi = 0. \tag{5.14a}$$

Noting that $\int p \, d\phi = 1$ and $\int p\phi \, d\phi \equiv \langle \phi \rangle = 1/\lambda_1$, it reduces to:

$$\frac{\dot{\lambda}_0}{\lambda_0} - \frac{\dot{\lambda}_1}{\lambda_1} = 0, \tag{5.14b}$$

which can be immediately integrated into:

$$\frac{\lambda_0}{\lambda_1} = \text{const.} \tag{5.14c}$$

---

[3]A standard argument for adopting the 6th moment, $\langle r^6 \rangle$, as a prognostic variable for predicting the PSD distribution is that because it can describe the spread of PSD well. In distribution problems, on the other hand, a standard choice for measuring the spread is the variance, thus a given distribution should rather be constrained by the 2nd moment, $\langle r^2 \rangle$, rather than by the 6th moment, $\langle r^6 \rangle$. The choice of $\sigma = r^6$ would only be justified when a mass distribution rather than a size distribution is considered, because in that case the variance is defined by $\sigma = m^2 \propto r^6$, where $m$ is the particle mass.

On the other hand, from a point of view of the output–constrained distribution principle, what is important is to predict a set of output variables that are required for the host model, rather than simply trying to predict a spread in an accurate manner. In this respect, as already pointed out, the reflectivity, $\langle r^6 \rangle$, is usually not a variable that is directly required within a cloud model. Probably, the most important process to be predicted is the coalescence, which is very crudely speaking, controlled by $n_c^2$, thus a weight to adopt would be $\sigma = n_c$, noting there is already a factor, $n_c$, in the definition of the integral with $\sigma$. For the precipitating particles, the same would apply to the sedimentation rate, which is proportional to a certain power, say, $a$, of the particle size, $r$, then $\sigma = r^a$ would be the choice.

That is the constraint under the normalization condition. Alternatively, the normalization condition can be obtained directly by performing an integral of the distribution analytically:

$$\lambda_0 = \lambda_1. \tag{5.14d}$$

Substitution of Eq. (5.14b) into Eq. (5.13), in the same manner as that of Eq. (5.5) into Eq. (5.6), makes the right–hand side depending only on $\lambda_1$. Substituting this final expression into the Liouville equation (3.24), we obtain:

$$p\left(\frac{1}{\lambda_1} - \phi\right)\dot{\lambda}_1 + \frac{\partial}{\partial\phi}(pF) = 0. \tag{5.15a}$$

Here, this equation contains $\phi$-dependence, thus it cannot be directly used to predict $\lambda_1$.

We remove the $\phi$–dependence from Eq. (5.15a) by multiplying a weight $\sigma_1$ that depends only on $\phi$ and integrating it by $\phi$ over $[0, +\infty]$. We choose the weight $\sigma_1 = \phi$, because the exponential distribution is to be used for predicting the mean value, based on an argument in Sec. 4.2.

Thus,

$$\left[\frac{1}{\lambda_1}\langle\phi\rangle - \langle\phi^2\rangle\right]\dot{\lambda}_1 - \langle F\rangle = 0, \tag{5.15b}$$

or by further noting $\langle\phi\rangle = 1/\lambda_1$ and $\langle\phi^2\rangle = 2/\lambda_1^2$,

$$\dot{\lambda}_1 = -\lambda_1^2\langle F\rangle. \tag{5.16a}$$

This equation states that when there is a positive mean source, $\langle F\rangle > 0$, the slope of the distribution becomes gentler by transporting it to larger values, whereas a mean sink (negative source) steepens the distribution. The above equation can readily be solved analytically, and we obtain

$$\lambda_1(t) = \left[\frac{1}{\lambda_1(0)} + \int_0^t \langle F\rangle dt\right]^{-1}. \tag{5.16b}$$

The significance of the above result may be best interpreted by re–writing it for the mean value:

$$\langle\phi\rangle = \lambda_1(t)^{-1} = \langle\phi\rangle\Big|_{t=0} + \int_0^t \langle F\rangle dt. \tag{5.16c}$$

This is the consistent evolution of the mean state under the assumed exponential distribution.

Here, the weight, $\sigma_1 = \phi$, has been chosen above in a manner consistent with the fact that the exponential distribution has been derived from the maximum–entropy principle taking the mean as the constraint. Yet, the general formulation presented in Sec. 5.1 can be used to predict any constraint defined by the weight, $\sigma_1$, consistently with time under a given assumed distribution. Thus, a natural question to ask is: how the evolution of the PDF parameter, $\lambda_1$, is sensitive to the choice of the weight, $\sigma_1$ for the constraint? To address this question, we now set the weight to be $\sigma_1 = \phi^n$ more generally with an unspecified integer, $n$. In this case, we evaluate the evolution of the assumed distribution (Eq. 3.22) in such a manner that $\langle\phi^n\rangle$ evolves

consistently. As a result, the prediction of the evolution of the parameter, $\lambda_1$, is modified to achieve the best prediction of $\langle \phi^n \rangle$ for a specified particular $n$ with a consequence of deteriorating the prediction of the other moments. Especially, when we set $n \neq 1$, the prediction of the mean value is no longer optimized by the constraint with $\sigma_1 = \phi^n$.

Consequently, instead of Eqs. (5.16b, c), we obtain:

$$\lambda_1(t) = \left[ \frac{1}{\lambda_1^n(0)} + \frac{n}{n!} \int_0^t \langle F \phi^{n-1} \rangle dt \right]^{-1/n}.\tag{5.17a}$$

$$\langle \phi^n \rangle = \langle \phi^n \rangle \Big|_{t=0} + n \int_0^t \langle F \phi^{n-1} \rangle dt,\tag{5.17b}$$

noting that $\langle \phi^n \rangle = n!/\lambda_1^n$. In this case, Eq. (5.17b) presents a consistent evolution of $\langle \phi^n \rangle$.

Keep in mind that the solution (5.17a) is implicit when $F$ itself also depends on $\phi$. A more explicit solution can be derived by setting it more specifically as, say, $F = \phi^m$. Solving an equivalent equation to Eq. (5.15b), we find:

$$\lambda_1(t) = (\lambda_1^{m-1}(0) - \gamma_m t)^{1/(m-1)},\tag{5.17c}$$

where

$$\gamma_m = (m-1)(n+m-1)!/n!,\tag{5.17d}$$

when $m \neq 1$.

It is clear that the parameter, $\lambda_1$, qualitatively changes with a different "rate", $\gamma_m$, of evolution with the varying $n$, thus the evolution of the assumed PDF is sensitive to the choice of the weight, $\sigma_1$. It also follows that a proper choice of $\sigma_1$ is crucial to ensure that an output, $\langle \sigma_1 \rangle$, of a particular interest is consistently predicted. A simple manner to achieve this consistency is to solve a prognostic equation for $\langle \sigma_1 \rangle$ in terms of $\lambda_1$. The general formulation presented in Sec. 5.1 is constructed in this manner.

Here, however, there is a serious problem with the above solution (5.17c): the exponential distribution continuously flattens with time, and the distribution becomes totally homogeneous at $t = \lambda_1^{m-1}(0)/\gamma_m$, then $\lambda_1 = 0$, and the solution breaks down beyond this point with $\lambda_1$ becomes a complex number. Such a collapse of the distribution is a dramatic example of suggesting an inherent limitation of the assumed–PDF approach. Fig. 2 plots the obtained time series of $\lambda_1$ with $m = 0, 2, 3, 4$ in (a)–(d) with the weight exponents, $n = 1$ (solid), $n = 2$ (long dash), and $n = 3$ (short dash). It is seen that the discrepancy of the solution with different weights, $\sigma_1$, exacerbates rapidly as a higher–order dependence of $F$ on $\phi$. Here, we set the initial condition as $\lambda_1(0) = 1$. This is equivalent to normalize the PDF parameter and the time into $\lambda_1/\lambda_1(0)$ and $\lambda_1^{1-m}(0)t$, respectively. The values of $\lambda_1$ evaluated from $\langle \phi^n \rangle$ with the exact solutions in Appendix B are further overlaid by greens with further discussions concerning the exact solutions to be referred therein.

The case with $m = 1$ must be considered separately, and in this case we find:

$$\lambda_1(t) = \lambda_1(0)e^{-t}.\tag{5.17e}$$

As it turns out, this is an exact solution of the evolution of the system, as shown in the Appendix B.

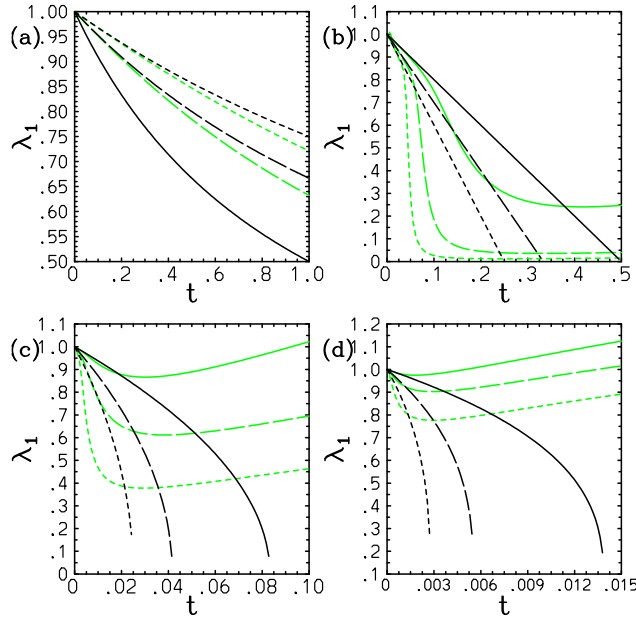

**Figure 2.** Plot of the obtained time series of $\lambda_1$ with $m = 0$ (a), 2 (b), 3 (c), and 4 (d) with the weight exponents, $n = 1$ (solid), $n = 2$ (long dash), and $n = 3$ (short dash). The values of $\lambda_1$ evaluated from $\langle \phi^n \rangle$ obtained by using the exact solutions are further overlaid by greens.

## 5.6 Application (2) : Gaussian Distribution

The second application is the Gaussian distribution, especially because this is a standard distribution assumed in data assimilation. The Gaussian distribution is also often assumed in subgrid-scale distribution problems. Here, the simplest case with a single variable is considered, as given by Eq. (3.23). Here, $p_0 = \lambda_0$, and we take the mean value, $\langle \phi \rangle = -2\lambda_2/\lambda_1$ to be a PDF parameter in place of $\lambda_1$. From a normalization condition

$$p_0 = \left( \frac{\lambda_2}{\pi} \right)^{1/2}. \tag{5.18}$$

This diagnostic relation can be used to update the distribution constant $p_0$ in place of updating it by integrating Eq. (5.5).

We further note:

$$\frac{\partial p}{\partial \langle \phi \rangle} = 2\lambda_2(\phi - \langle \phi \rangle)p, \tag{5.19a}$$

$$\frac{\partial p}{\partial \lambda_2} = -(\phi - \langle \phi \rangle)^2 p. \tag{5.19b}$$

By substituting these expressions into Eq. (5.8a), we obtain:

$$2\lambda_2\langle \dot{\phi} \rangle[\langle \sigma_l(\phi - \langle \phi \rangle) \rangle - \langle \sigma_l \rangle \langle \phi - \langle \phi \rangle \rangle] + \dot{\lambda}_2[-\langle \sigma_l(\phi - \langle \phi \rangle)^2 \rangle + \langle \sigma_l \rangle \langle (\phi - \langle \phi \rangle)^2 \rangle] = \left\langle F \frac{\partial \sigma_l}{\partial \phi} \right\rangle. \tag{5.20}$$

Note that an integration by parts is applied to the last term in Eq. (5.8a) to obtain the right-hand side above. Note also that $\langle \phi - \langle \phi \rangle \rangle = 0$, and thus

 $$2\lambda_2 \langle \dot{\phi} \rangle \langle \sigma_l (\phi - \langle \phi \rangle) \rangle + \dot{\lambda}_2 [-\langle \sigma_l (\phi - \langle \phi \rangle)^2 \rangle + \langle \sigma_l \rangle \langle (\phi - \langle \phi \rangle)^2 \rangle] = \left\langle F \frac{\partial \sigma_l}{\partial \phi} \right\rangle. \tag{5.21}$$

Recall that the Gaussian distribution is obtained from the maximum–entropy principle when a system is constrained by the mean and the variance; thus we set $\sigma_1 = \phi - \langle \phi \rangle$ and $\sigma_2 = (\phi - \langle \phi \rangle)^2$ in the above. With $l = 1$, we obtain

$$2\lambda_2 \langle (\phi - \langle \phi \rangle)^2 \rangle \langle \dot{\phi} \rangle - \langle (\phi - \langle \phi \rangle)^3 \rangle \dot{\lambda}_2 = \langle F \rangle \tag{5.22a}$$

and with $l = 2$,

 $$2\lambda_2 \langle (\phi - \langle \phi \rangle)^3 \rangle \langle \dot{\phi} \rangle + [-\langle (\phi - \langle \phi \rangle)^4 \rangle + \langle (\phi - \langle \phi \rangle)^2 \rangle^2] \dot{\lambda}_2 = 2\langle (\phi - \langle \phi \rangle) F \rangle. \tag{5.22b}$$

Note that the Gaussian distribution is not skewed, thus $\langle (\phi - \langle \phi \rangle)^3 \rangle = 0$. Also note the relations

$$\langle (\phi - \langle \phi \rangle)^2 \rangle = \frac{1}{2\lambda_2}, \tag{5.23a}$$

$$\langle (\phi - \langle \phi \rangle)^4 \rangle = 3\langle (\phi - \langle \phi \rangle)^2 \rangle^2. \tag{5.23b}$$

By substituting these relations into Eqs. (5.22a, b), we obtain the final results:

 $$\langle \dot{\phi} \rangle = \langle F \rangle, \tag{5.24a}$$

$$\frac{d}{dt} \left( \frac{1}{\lambda_2} \right) = 4\langle (\phi - \langle \phi \rangle) F \rangle. \tag{5.24b}$$

Eq. (5.24a) simply means that the mean value evolves by following a tendency defined by the mean source, whereas Eq. (5.24b) suggests that the distribution is more dispersed when a more positive source is found away from the mean value.

### 5.6.1 Diffusion Problem

 The diffusion equation (4.1) considered in Sec. 4.4 is a particular problem that can be solved exactly by the Gaussian distribution (3.23). We note that the diffusion equation (4.1), re–setting $x = \phi$ here, is obtained from the Liouville equation (3.24) by setting the forcing to be an operator, $F = -\partial/\partial \phi$. As a result, the general formulation presented so far can directly be applied to the diffusion equation: by substituting the derivative relations (5.19a, b) into Eq. (5.7a), and also noting that

$$\frac{\partial^2 p}{\partial \phi^2} = [-2\lambda_2 + 4\lambda_2^2 (\phi - \langle \phi \rangle)^2] p \tag{5.25}$$

 we obtain:

$$2\lambda_2 (\phi - \langle \phi \rangle) \langle \dot{\phi} \rangle - [(\phi - \langle \phi \rangle)^2 - \frac{1}{2\lambda_2}][\dot{\lambda}_2 + 4\lambda_2^2] = 0. \tag{5.26}$$

Note that the same can also be obtained by directly substituting (3.23) into Eq. (4.1). The above equation can be solved for the two parameters, $\langle\phi\rangle$ and $\lambda_2$, independent of $\phi$ by setting:

$$\langle\dot{\phi}\rangle = 0, \tag{5.27a}$$

$$\dot{\lambda}_2 + 4\lambda_2^2 = 0 \tag{5.27b}$$

The same pair as the above is more directly obtained by substituting $\langle F\rangle = 0$ and $\langle(\phi - \langle\phi\rangle)F\rangle = -1$ into Eqs. (5.24a, b). Solving for Eqs. (5.27a, b), we arrive at the solution (4.4).

## 5.7 Application (3): Gamma Distribution

The third example to consider is the gamma distribution (*cf.*, Sec. 3.4.3):

$$p = p_0\phi^\mu e^{-\lambda\phi}, \tag{5.28a}$$

where

$$p_0 = \frac{\lambda^{\mu+1}}{\Gamma(\mu+1)} \tag{5.28b}$$

from the normalization condition. Recall that the gamma function, $\Gamma(x)$, is defined by

$$\Gamma(x) = \int\limits_0^{+\infty} \xi^{x-1}e^{-\xi}d\xi. \tag{5.28c}$$

We note the expressions for the derivatives by the distributions parameters:

$$\frac{\partial p}{\partial\mu} = p\log\phi, \tag{5.29a}$$

$$\frac{\partial p}{\partial\lambda} = -\phi p. \tag{5.29b}$$

By substituting these two expressions into Eq. (5.8a), we obtain

$$\dot{\mu}[\langle\sigma_l\log\phi\rangle - \langle\sigma_l\rangle\langle\log\phi\rangle] + \dot{\lambda}[-\langle\sigma_l\phi\rangle + \langle\sigma_l\rangle\langle\phi\rangle] = \left\langle F\frac{\partial\sigma_l}{\partial\phi}\right\rangle \tag{5.30}$$

for $l = 1, 2$. As before, we set $\sigma_1 = \phi$ and $\sigma_2 = \phi^2$. We note especially

$$\langle\phi\log\phi\rangle = \frac{1}{\lambda}[(\mu+1)\langle\log\phi\rangle + 1], \tag{5.31a}$$

$$\langle\phi^2\log\phi\rangle = \frac{1}{\lambda^2}[(\mu+1)(\mu+2)\langle\log\phi\rangle + 2\mu + 3]. \tag{5.31b}$$

Thus, with $l = 1$ and 2, respectively, we obtain

$$\frac{\dot{\mu}}{\lambda} - (\mu+1)\frac{\dot{\lambda}}{\lambda^2} = \langle F\rangle, \tag{5.32a}$$

$$\frac{2\mu+3}{\lambda^2}\dot{\mu} - \frac{2(\mu+2)(\mu+1)}{\lambda^3}\dot{\lambda} = 2\langle\phi F\rangle. \tag{5.32b}$$

By combining Eqs. (5.32a, b), we obtain the equations for the tendencies of the two PDF parameters as:

$$\dot{\mu} = 2\lambda[(\mu+2)\langle F \rangle - \lambda\langle \phi F \rangle], \tag{5.33a}$$

$$\dot{\lambda} = \frac{\lambda^2}{\mu+1}[(2\mu+3)\langle F \rangle - 2\lambda\langle \phi F \rangle]. \tag{5.33b}$$

## 6 Demonstration: Condensation Growth of Cloud Droplets

The general formulation for directly predicting the evolution of the assumed–PDF(DDF) parameters has been presented in the last section. The purpose of this section is to demonstrate the steps of this formulation by taking the condensation growth of cloud droplets as an example. The output-constrained distribution introduced in Sec. 4.2 is also invoked in choosing a distribution form. It is known that the size, $r$, of cloud droplet grows with a rate proportional to $1/r$ under a fixed state of super–saturation (*cf.*, Ch. 6, Rogers and Yau 1989). Thus, by setting a proportionality constant to be unity, the governing

equation (2.2) of this system becomes

$$\dot{r} = \frac{1}{r} \tag{6.1a}$$

with $\phi = r$ and $F = 1/r$. In this case, $p(r)$ becomes a number density of drops with a radius $r$, or PSD. Here, we normalize the PSD with the condition (3.11).

The system (6.1a) can analytically be solved, and the general solution is:

$$r(t) = (r_0 + 2t)^{1/2} \tag{6.1b}$$

with $r_0$ the initial condition. Consequently, when an initial distribution of droplets is given by

$$p(r, t=0) = p_0(r), \tag{6.2}$$

its subsequent evolution is defined by

$$p(r,t)dr = p_0(r_0)dr_0 \tag{6.3a}$$

by following the chain rule of Eq. (3.18). Here, the initial condition, $r_0 = r(0|r,t)$ is related to the droplet size, $r$, at the time $t$ by:

$$r_0 = r(0|r,t) = (r^2 - 2t)^{1/2}. \tag{6.3b}$$

From Eq. (6.3b), we find $dr_0/dr = r/r_0$, and by substituting this final result into Eq. (6.2), we obtain

$$p = (r/r_0)p_0, \tag{6.3c}$$

but $p(r,t) = 0$ for $r^2(t) < 2t$. In this manner, the problem of time evolution of the droplet distribution under the condensation growth is solved analytically.

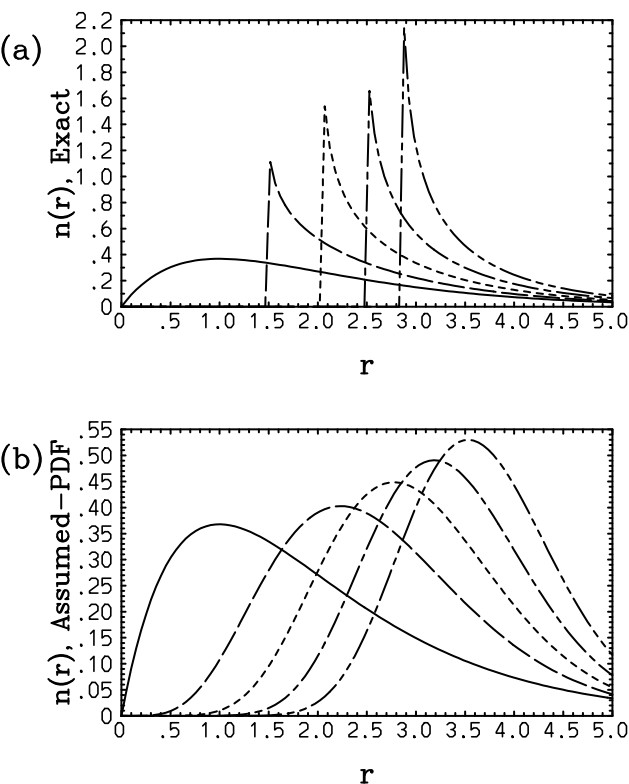

**Figure 3.** Time evolution of cloud-droplet size distribution under condensation growth with the curves corresponding to $t = 0$ (solid) to 4 (double–chain dash) with an interval of $\Delta t = 1$: (a) exact result, and (b) when a gamma distribution is assumed. Note that the whole distribution moves to a larger droplet size with time.

As a specific initial distribution, we set

$$p(r, t = 0) = \frac{\lambda^{\mu+1}}{\Gamma(\mu+1)} r^{\mu} e^{-\lambda r} \tag{6.4}$$

with $\mu = 1$ and $\lambda = 1$. The exact time evolution following this initialization, which obeys Eq. (6.3a) or (6.3c), is shown in Fig. 3(a) with varying curves for $t = 0$–4 with an interval of $\Delta t = 1$. The whole distribution moves with time to larger sizes, $r$, as seen in Fig. 3(a), because of condensational growth, as expected, but also associated with the squeezing tendency by the factor, $r/r_0$, with time, as predicted from the analytical solution (6.3c). The rate of squeezing is larger towards the smaller $r_0$, and forms a shock–wave front at the peak at the minimum droplet size,

$$r = (2t)^{1/2}. \tag{6.5}$$

The peak also sharpens with time, as it moves to larger values with time. The pronounced narrowing tendency of the distribution with time is well known in the literature. Remarkably, it leads to a decrease of the standard deviation with time as seen below.

In the following, the two assumed PSDs are considered for demonstrations of the general formulation of the last section.

## 6.1 Gamma distribution

Considering the fact that our example is a microphysical problem, the most natural choice to consider is the gamma distribution (5.28a), as a commonly adopted distribution in microphysics (*cf.*, Sec. 3.4.3). A general formulation for this case is presented in Sec. 5.7. The only additional information required is to note $\langle F \rangle = \lambda/\mu$ and $\langle rF \rangle = 1$ with the system (6.1a), recalling the definition (5.10c) for the angle bracket, and also $F = 1/r$. Substituting them, Eqs. (5.33a, b) are solved by the 4th–order Runge–Kutta method with a time step of $10^{-2}$ and the initial condition given by $\mu = 1$ and $\lambda = 1$. The result is shown in Fig. 3(b) with the same curves as in (a). Obviously, it is not possible to re–produce the shock–wave structure at the minimum size (Eq. 6.5) by a gamma distribution. However, it is still remarkable that an overall evolution of the size distribution is reasonably reproduced.

The result here is especially remarkable, because the computation of the evolution of the distribution involves only time integrations of two parameters, $\mu$ and $\gamma$, then the full distribution is automatically determined from the assumed gamma distribution form. In a standard approach of directly evaluating the evolution of a distribution by integrating Eq. (3.24) with time, we need to introduce a number of values for $r$, that we wish to evaluate. For example, in Figs. 2(a) (b), 100 points are used for plotting the distribution curves for each time. Furthermore, for numerically integrating the Liouville equation (3.24), we need to take $r$ large enough so that we can set $p = 0$ as the boundary condition at the largest $r$: this requirement further adds number of points required for the computation, say, by the factor of 10. In standard bin microphysics, 30 points are considered minimum (*cf.*, Khain *et al.* 2015). The assumed–PDF(DDF) approach adopted here enables us to compute the same only with the two parameters, instead of many bins for $r$.

To verify the prediction of statistics, Fig. 4 plots the time evolutions of (a) the average size, $\langle r \rangle$, and (b) the standard deviation, $\langle (r - \langle r \rangle)^2 \rangle^{1/2}$ with $\langle r \rangle = (\mu + 1)/\lambda$ and $\langle (r - \langle r \rangle)^2 \rangle = (\mu + 1)/\lambda^2$. Here, the exact evolutions are shown in solid, whereas the approximate predictions with the assumed gamma distribution are shown by long–dash curves. Keep in mind that from a point of view of maximum–entropy principle, the gamma distribution is designed to predict only a mean value properly. An additional algebraic correction factor is not a result of any "constraint" from the point of view of the large deviation principle (*cf.*, Sec. 3.3.2). These predictions are accurate only under limits of an assumed distribution. In fact, the prediction of the mean (Fig. 4(a)) is almost perfect, although that of the standard deviation deviates noticeably from the actual evolution with time (Fig. 4(b)).

## 6.2 Exponential distribution

We further simplify the assumed distribution into an exponential so that there is only a single parameter, $\lambda_1$, in the distribution (*cf.*, Eq. 5.15a). In this case, the assumed distribution no longer fits to the actual evolution initiated with a gamma distribution (6.4) as $r \to 0$.

Here, we face a technical problem for adopting the exponential distribution for this system, because the integral for $\langle F \rangle$ diverges due to a singularity of the source, $F$, at $r = 0$. To avoid this problem, we set $\lambda_1$ of the exponential distribution in such manner that it gives the same mean size, $\langle r \rangle$, as the gamma distribution given by Eq. (6.4) [*i.e.*, $\lambda_1 = \lambda/(1 + \mu)$], and then we

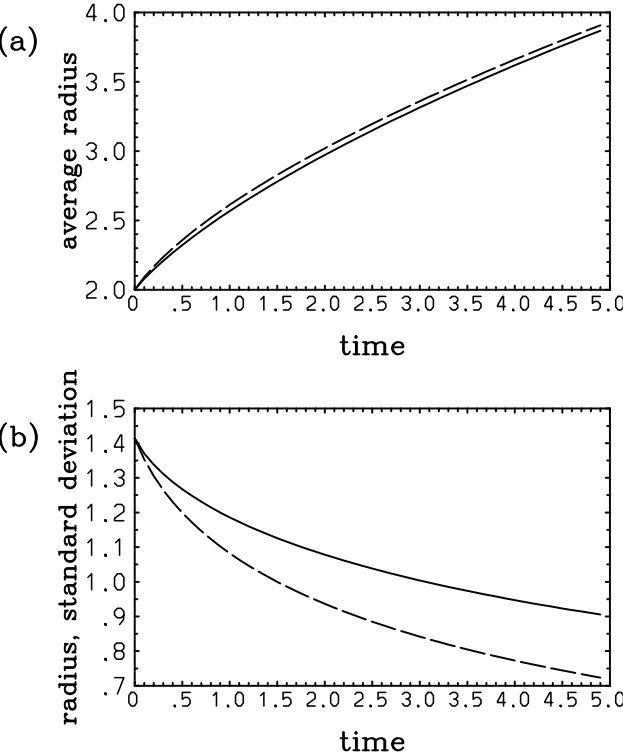

**Figure 4.** Time evolutions of (a) the average size, $\langle r \rangle$, and (b) the standard deviation, $\langle (r - \langle r \rangle)^2 \rangle^{1/2}$ of droplets under the condensation growth. Here, exact evolutions are shown in solid, whereas the predictions with the assumed gamma distribution are shown by long–dashed curves.

adopt a formula, $\langle F \rangle = \lambda / \mu$, directly from the gamma distribution. Thus, $\langle F \rangle = (1 + \mu) \lambda_1 / \mu$. Here, $\mu$ is kept the same as the initial condition of the distribution (6.4). Under this assumption, the time integral of Eq. (5.16a) can be performed, and the evolution of the mean size can be evaluated by Eq. (5.16c). Evolution of the standard deviation is also evaluated as $1 / \lambda_1$.

These results are shown in Fig. 5 in the same format as in Fig. 4: evolution of the standard deviation (long-dashed curve in (b)) now becomes totally opposite to the actual tendency (solid curve). However, the evolution of the mean size is still overall predicted in consistent manner in (a). The results are simply consistent with the assertion in Sec. 4.2 that if a sole purpose of using a distribution is to predict evolution of a mean, the exponential distribution is sufficient for the purpose. Here, this assertion is supported to a good extent, even though an actual distribution does not fit to the exponential distribution at all.

The Appendix C presents further demonstrative mathematical examples.

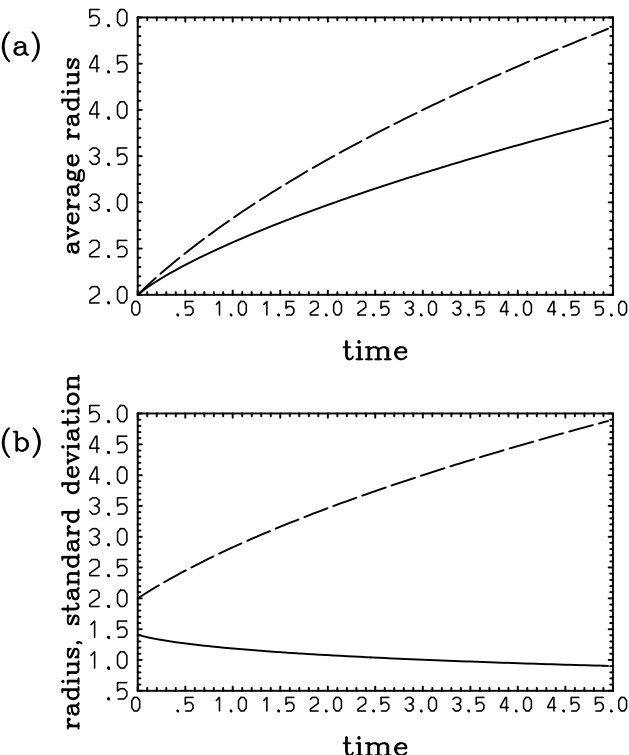

**Figure 5.** The same as in Fig. 3, but long-dashed curves are when an exponential distribution is assumed for prediction.

**7 Conclusions and Discussions**

Distribution problems are identified in various contexts of atmospheric sciences: subgrid–scale distribution, size distribution of hydrometeors, probabilities in data assimilation. Considering dispersed literature on those three distribution problems, the present paper has tried to setup a general perspective for all the distribution problems. As noted in the introduction, the majority of approaches in the distribution problems is based on a framework, which can be termed "assumed PDF", in which

a distribution to be simulated is approximated by a distribution form characterized only by a few free parameters (*cf.*, Golaz *et al.* 2002).

The present work has attempted to answer the basic questions of how such a simple assumed PDF can be best chosen and how the evolution of those PDF parameters can be evaluated consistently. To address these basic questions, it is important to realize that not every statistical aspect of any distribution can be predicted by the assumed–PDF approaches either accurately nor

consistently. More specifically, the number of statistical variables (but not limited to simple moments) that can be consistently predicted by an assumed–PDF approach cannot exceed the number of assumed PDF parameters introduced by the given assumed PDF. Thus, the best that we can accomplish under an assumed–PDF approach is to predict the same number of

statistical variables as that of the assumed–PDF free parameters. The proposed formulation in the present study is designated exactly to accomplish this best.

The present study next notes (Sec. 4.2) that in many atmospheric applications, a given host model does not require to use the full form of a distribution as an output from a distribution model, but instead, requires only a limited number of statistical quantities that correspond to the physical variables of interest in the host model (*e.g.*, domain–averaged total condensed water, cloud fraction) as outputs. Thus, the most desirable formulation would be to predict the evolution of those statistical outputs required in a given host model consistently. The final general formulation presented in Sec. 5.1 is constructed exactly by following this recipe: the time evolution of the assumed–PDF parameters, $\{\lambda_l\}$, is evaluated by Eq. (5.8a), which is equivalent to solving the prediction equation (5.10b) for the required model outputs, $\{\langle \sigma_l \rangle\}$, in terms of $\{\lambda_l\}$.

Probably, the most novel aspect of the present study is in deriving equations for the assumed–PDF parameters and prognosing them directly. This proposed alternative approach can overcome difficulties of a current standard method that requires a difficult mapping between the moments of the PDF (mean, variance, skewness, etc. of overall PDF) and the PDF parameters (*e.g.*, mean of each Gaussian component in a double-Gaussian PDF, *cf.*, Lewellen and Yoh 1993, Machulskaya 2015, Milbrandt and Yau 2015). Furthermore, by introducing a re–interpretation of the maximum–entropy principle (*cf.*, Sec. 3.3) in Sec. 4.2, we propose to adopt those required statistical outputs as "constraints" to define the assumed–PDF form for a given problem to resolve the common problem of how to choose it, which is usually done rather arbitrarily. The paper has also emphasized that approaches of trying to choose a distribution form based on analyses of observational or simulation data found in the literature lack objectivity (*cf.*, Sec. 3.2.2).

In the present study, the possible model outputs and constraints have been limited to much simpler statistical quantities of the form, $\{\langle \sigma_l \rangle\}$. However, generalizations to those physically more significant variables, such as the domain–averaged total condensed water and the cloud fraction, are conceptually straightforward. For example, the cloud fraction can formally be defined in terms of the DDF, $p(q_t)$, of the total water, $q_t$ as:

$$\int\limits_{q^*}^{+\infty} p(q_t)dq_t, \tag{7.1}$$

where $q^*$ is the saturated moisture at a given height. By a similar line of argument as presented in Sec. 5, a prognostic equation for the cloud fraction, as defined above, can be derived by integrating Eq. (5,7a) over the range of $[q^*, +\infty]$, although a full derivation is left for a future paper. Alternatively, a prognostic equation for the consistent DDF parameters can more directly be derived from the prognostic equation for the cloud fraction (7.1), to be written down in a similar manner as for Eq. (5.10b). A consistent assumed–distribution form can also be derived by introducing the integral (7.1) as a constraint for the maximum–entropy principle.

An obvious consequence from this formulation is that the evolution of the PDF (DDF) would be predicted differently by choosing the weights, $\{\sigma_l\}$, differently. However, this consequence should not be considered an inherent shortcoming of the present formulation. Rather, one should realize that this is the fundamental limitation of the assumed–PDF(DDF) approach: it cannot predict every statistic of a distribution accurately as already emphasized. Simply, a different choice of the set, $\{\sigma_l\}$,

predicts a different set of statistical outputs, $\{\langle\sigma_l\rangle\}$, consistently. This point has been explicitly demonstrated in Sec. 5.5. Thus, our best advice is to choose $\{\langle\sigma_l\rangle\}$ to be those actually required by the host model so that they are actually predicted consistently under a given assumed PDF form.

The present work is built upon a solid basis of the two well–known mathematical and physical principles: 1) the maximum–entropy principle, which guides the determination of the most likely distribution of a problem, and 2) the Liouville equation, which predicts the time evolution of distributions. The stochastic–collection equation for hydrometeor size–distribution can also be treated in analogous manner as the Liouville equation. However, adopting these two basic principles in practical atmospheric problems is not quite straightforward, but we face two major difficulties: First, application of the maximum–entropy principle to the atmospheric problems is not straightforward, because it is often not obvious how to identify the physical constraints required for determining a distribution (*cf.*, Yano *et al.* 2016). Second, though the Liouville equation permits us to perform a prediction of distributions in a formal manner, both for DDF and PDF, its direct use would entail enormous numerical cost. We have addressed these two difficulties by adopting a simple distribution form containing only a small number of parameters so that its evolution with time can be predicted only in terms of those parameters. In principle, the formulations presented in this study are applicable to any atmospheric model in a straightforward manner, albeit with required coding and testing, because the term, $F$, for the physical tendency in the Liouville equation (3.24) is left unspecified in presenting the general formulations. This straightforwardness includes no need for introducing any extra closure, say, based on a turbulence model, for example, as partially the case with CLUBB (Cloud Layers Modified By Binormals: Golaz *et al.* 2002, Larson and Golaz 2005, Larson *et al.* 2019), so that higher–order moments can be expressed in terms of the moments considered under a given truncation of the system.

As a demonstrative example with a specified $F$, the paper has considered the condensational growth of cloud droplets. A rather unusual feature of this demonstration is in comparing its performance with an exact numerical result: the flexibility of the proposed formulation permits us to do this very easily. Additional mathematical demonstrative examples are found in the Appendix C. Further examples with further elaborations in methodology are also found in the study by Yano (2024). We also anticipate that fuller applications of the developed formulation are still to follow, especially because the only way to evaluate the accuracy of the method is to directly compare the results with more accurate sophisticated evaluations, as presented in Secs. 5.5, 6, and Appendix C herein as well in Yano (2024). Yet, it is hoped that this simple example provides a concrete idea about how to use this general formulation, and that readers can already apply their formulation to their own problems.

Two relatively distinct steps are involved in the present formulation: 1) determination of the PDF form based on the output-constrained distribution principle, as a re–interpretation of the maximum entropy principle; 2) prognostic equations for the PDF parameters derived from the Liouville equation. Thus, whenever new types of the output "constraints" are introduced to a problem, new types of general PDF forms must first be derived based on step 1). Next, new sets of prognostic equations must be derived based on step 2). Alternatively, two steps can be adopted separately. For example, evolution of a PDF form defined under the output-constrained distribution principle can be evaluated by a more traditional assumed-PDF approach based on moments. Conversely, currently existing assumed-PDF schemes can be re-written based on step 2), but without changing the assumed PDF forms.

The general formulation for the distribution problem presented here, first of all, constitutes a natural extension of the existing assumed–PDF(DDF) approaches for the subgrid–scale distributions as discussed in Sec. 5.3. The work further suggests that the same general formulation is also applicable to other distribution problems, including cloud microphysics and data assimilation. Extensive further general possibilities of the proposed formulation are yet to be fully explored. For example, constraints introduced by Eq. (3.10) have assumed a fixed integral range. On the other hand, for example, the cloud–fraction problem needs to take as an output an integral in respect to the total water above the saturation value (*cf.*, Eq. 7.1), which changes with time. It requires a further generalization of the formulation (*e.g.*, with Eq. 5.5).

Furthermore, there is a key issue left unaddressed: performance of integrals designated by $\langle \cdots \rangle$ throughout the paper. In the present study, all the integrals of the problem have been performed analytically, apart from those including an unspecified physical–tendency term, $F$. Performing the latter integrals in general, especially with a complex physical–tendency term, is however, not trivial. The most flexible approach currently available is Monte-Carlo integration (e.g., Gentle 2003, Larson and Schanen 2013). However, this approach is numerically rather expensive, and hardly considered an ultimate answer. A key to a success of the prognostic assumed–PDF formulation presented herein, especially our key result, Eq. (5.8a), is to perform those integrals in efficient manner, albeit it may be with a numerical aide. The integral problems in general can much be facilitated by adopting Laplace's method in asymptotic expansion, and more specifically by invoking Watson's Lemma (*cf.*, Ch. 6, Bender and Orszag 1978), considering a fact that more or less all the types of distributions of interest here take the form of an exponential decay away from the maximum. Especially, all these integrals reduce to a form of an asymptotic expansion in term of gamma functions, if a Taylor expansion of a subdominant contribution is possible. The stronger a distribution is peaked, the fewer terms are required.

Another aspect, that has not been explicitly taken into account, is stochasticity: its potential importance in atmospheric modelling is hardly overemphasized (*cf.*, Berner *et al.* 2017). Formulations already exist for taking stochasticity into account as a generalization of the Liouville equation (*cf.*, Risken 1984).

## Appendix A: Mathematical Details of Sec. 4.4.2

We apply a Fourier integral

$$p(x,t) = \int_{-\infty}^{+\infty} \tilde{p}(k,t)e^{ikx}dk \tag{A.1}$$

to the diffusion problem (4.1). The Fourier transform for (A.1) is given by

$$\tilde{p}(k,t) = \frac{1}{2\pi} \int_{-\infty}^{+\infty} p(x,t)e^{-ikx}dx. \tag{A.2}$$

Thus, the initial condition is transformed into:

$$\tilde{p}(k,t=0) = \frac{1}{2\pi} \int_{-\infty}^{+\infty} (1+\alpha x)e^{-\lambda x^2 - ikx}dx. \tag{A.3}$$

The transform of the first term is straightforward and

$$\int_{-\infty}^{+\infty} e^{-\lambda x^2 - ikx} dx = e^{-k^2/4\lambda} \int_{-\infty}^{+\infty} e^{-\lambda(x+ik/2\lambda)^2} dx = \left(\frac{\pi}{\lambda}\right) e^{-k^2/4\lambda}. \tag{A.4}$$

For performing the second term in the transform (A.3), we note a relation:

$$xe^{-\lambda(x+ik/2\lambda)^2} = -\frac{1}{2\lambda}\frac{\partial}{\partial x}e^{-\lambda(x+ik/2\lambda)^2} + \frac{ik}{4\lambda^2}e^{-\lambda(x+ik/2\lambda)^2}. \tag{A.5}$$

Finally, in performing the inverse Fourier transform (A.1), we re-factorize the exponent by noting:

$$k^2(t + \frac{1}{4\lambda}) - ikx = (t + \frac{1}{4\lambda})[k - \frac{ix}{2}(t + \frac{1}{4\lambda})^{-1}] + \frac{x^2}{4}(t + \frac{1}{4\lambda})^{-1}. \tag{A.6}$$

## Appendix B: Exact Solutions of the Problem in Sec. 5.5

The problem of the system with $F = \phi^m$ in Sec. 5.5 can be solved analytically, and the evolution of the PDF can also be solved in a closed form by following the methodology of Sec. 6: the resulting solution of the PDF is:

$$p(\phi, t) = p(\phi_0(\phi, t), t = 0)\frac{d\phi_0}{d\phi}, \tag{B.1}$$

where $\phi_0 = \phi_0(\phi, t)$ is an initial condition that leads to a state, $\phi$, at time, $t$.

Once an explicit PDF form is given, numerical integrals further provide the required moments, $\langle\phi^n\rangle$. As a minor technicality, the actual integral is performed in respect to $\phi_0$ after transforming $\phi$ into $\phi_0$ , because the distribution tends to stretch to larger values increasingly with time, and we need to increase the upper limit of the integral in respect to $\phi$ accordingly. The variable transformation allows us to stretch the integral range in respect to $\phi$ automatically by simply preforming the integral over the same range in respect to $\phi_0$. Once, a moment, $\langle\phi^n\rangle$, for a given $n$ is obtained, a consistent exponent, $\lambda_1$, can be evaluated by the formula:

$$\lambda_1 = \left[\frac{n!}{\langle\phi^n\rangle}\right]^{1/n} \tag{B.1a}$$

by invoking the fact that the moments are given by

$$\langle\phi^n\rangle = \frac{n!}{\lambda_1^n} \tag{B.1b}$$

under an assumption of the exponential distribution. The parameter, $\lambda_1$, estimated from Eq. (B.1a) is added as green curves in Fig. 2.

The following subsections discuss the exact solutions for specific choice of the power, $m$, that is required to accomplish those steps.

### B.1 When $m = 0$

The system simply reduces to $\dot\phi = 1$, which is solved into $\phi = \phi_0 + t$. It follows that $\phi_0 = \phi - t$ and $d\phi_0/d\phi = 1$. By substituting those relations into Eq. (B.1), we find:

$$p(\phi, t) = \begin{cases} \lambda_1 e^{-\lambda_1(x-t)} & x \geq t \\ 0 & x < t \end{cases} \tag{B.2}$$

From this distribution, the moments can be readily evaluated analytically, and the first three moments are given by:

$$\langle \phi \rangle = \frac{1}{\lambda_1(0)} + t, \tag{B.3a}$$

$$\langle \phi^2 \rangle = 2 \left( \frac{1}{\lambda_1^2(0)} + \frac{t}{\lambda_1(0)} + \frac{t^2}{2} \right), \tag{B.3b}$$

$$\langle \phi^3 \rangle = 6 \left( \frac{1}{\lambda_1^3(0)} + \frac{t}{\lambda_1^2(0)} + \frac{t^2}{3\lambda_1(0)} + \frac{t^3}{27} \right). \tag{B.3c}$$

The values of $\lambda_1$ obtained from the above results can directly be compared with those obtained by setting $m = 0$ in Eq. (5.17c, d):

$$\lambda_1 = \frac{1}{\lambda_1(0)} + \frac{t}{n} \tag{B.4}$$

with the varying $n$. The assumed formulation can predict the consistent value of $\lambda_1$ with the constraint of $\sigma_1 = \phi^n$ up to $O(t)$, in general, and the agreement is perfect especially with $n = 1$.

### B.2 When $m = 1$

In this case, we find

$$\phi = \phi_0 e^t, \quad \phi_0 = \phi e^{-t}, \quad \frac{d\phi_0}{d\phi} = e^{-t} \tag{B.5a, b, c}$$

From these relations, we find that this system evolves exactly as prescribed by Eq. (5.17e), maintaining the exponential distribution.

### B.3 When $m > 1$

In this case, the exact solution of the system is:

$$\phi = [\phi_0^{-(m-1)} - (m-1)t]^{-1/(m-1)} \tag{B.6}$$

and it follows that

$$\phi_0 = [\phi^{-(m-1)} + (m-1)t]^{-1/(m-1)}, \quad \frac{d\phi_0}{d\phi} = \frac{1}{\phi^m}[\phi^{-(m-1)} + (m-1)t]^{-m/(m-1)} \tag{B.7a, b}$$

By substituting Eqs. (B.7a, b) into Eq. (B.1), an explicitly form of the PDF evolution is obtained.

Note that the solution (B.6) becomes singular at $t = \phi^{-(m-1)}/(m-1)$ with the tendency to $\phi \to +\infty$. Due to this tendency, the distribution increasingly presents a longer tail with time with an increasingly significant deviation from an exponential distribution. This tendency also aggravates with the increasing $n$, thus the estimate of $\lambda_1$ based on the constraint, $\sigma_1 = \phi^n$, deteriorate also faster with time with the increasing $n$ (*cf.*, Eqs. 5.17c, d), as seen in Fig. 2.

## Appendix C: Further Demonstrative Examples

This appendix further compares between the exact and assumed–PDF solutions by taking three simple dynamical systems, and their basic behaviors can be inferred from the forcing forms. These comparisons follow up a demonstration of the assume–PDF approach taking a simple physical system in Sec. 6, and further expose basic characteristics of the assume–PDF solutions.

As simple dynamical systems, we consider the following three forms of forcing:

(i) $F = -\phi$: the only stable fixed point of this system is $\phi = 0$, and the system approaches to this fixed point regardless of the initial condition, $\phi_0$, with $\phi = \phi_0 e^{-t}$.

(ii) $F = \phi(\phi - 1)$: the system consists of a stable fixed point at $\phi = 0$ and an unstable fixed point at $\phi = 1$. Every initial point below $\phi = 1$ exponentially approaches to $\phi = 0$, and every initial point above $\phi = 1$ exponentially approaches to infinity with time. An explicit solution is:

$$\phi = \frac{\phi_0}{\phi_0 - (\phi_0 - 1)e^t} \tag{C.1}$$

with $\phi_0$ the initial condition.

(iii) $F = -\phi(\phi - 1)(\phi - 2)$: the system consists of two stable fixed points at $\phi = 0$ and 2, and an unstable fixed point at $\phi = 1$. Every initial point below and above $\phi = 1$, respectively, exponentially approaches to $\phi = 0$ and 2. An explicit solution is:

$$\phi = 1 \pm (1 - Ae^{-2t})^{-1/2} \tag{C.2a}$$

where

$$A = \frac{\phi_0(\phi_0 - 2)}{(\phi_0 - 1)^2} \tag{C.2b}$$

and the sign in the solution is chosen by the sign of $\phi_0 - 1$.

We examine the evolution of these systems by initializing the distribution as Gaussian of the form (3.23) with $\langle \phi \rangle|_{t=0} = 1$ and $\lambda_1|_{t=0} = 10$. Here, the initial peak of the distribution is taken to be sharp enough so that we can focus on the evolution of the system initialized at the vicinity of $x = 1$. The assumed peak point is not a stable fixed point with any of those systems, and it is more precisely an unstable fixed point with the latter two systems. Thus, the evolution is expected to be away from the given initial peak in all these three cases.

The exact evolutions of those three systems with this initial distribution can be derived by using the relations:

(i) $$\phi_0 = \phi e^t, \quad d\phi_0/d\phi = e^t. \tag{C.3}$$

(ii)
$$\phi_0 = \frac{\phi}{\phi - (\phi - 1)e^{-t}}, \tag{C.4a}$$

$$\frac{d\phi_0}{d\phi} = \frac{e^{-t}}{(\phi - (\phi - 1)e^{-t})^2}. \tag{C.4b}$$

(iii)
$$\phi_0 = 1 \pm (1 - Be^{2t})^{-1/2}, \tag{C.5a}$$

$$\frac{d\phi_0}{d\phi} = 1 \pm (1 - Be^{2t})^{-3/2}e^{2t} \tag{C.5b}$$

with

$$B = \frac{\phi(\phi - 2)}{(\phi - 1)^2}. \tag{C.5c}$$

The exact evolutions of the distributions are obtained by substituting these relations into Eq. (B.1).

Evolution of a distribution under the assumed PDF (*i.e.*, Gaussian) is evaluated in terms of the evolutions of the two PDF parameters, $\langle \phi \rangle$ and $\lambda_1$, as given by Eqs. (5,24a, b). For this purpose, the following relations for the three systems are invoked:

(i)
$$\langle F \rangle = -\langle \phi \rangle, \quad \langle (\phi - \langle \phi \rangle)F \rangle = -\frac{1}{2\lambda}. \tag{C.6a, b}$$

(ii)
$$\langle F \rangle = \frac{1}{2\lambda} + F(\langle \phi \rangle), \quad \langle (\phi - \langle \phi \rangle)F \rangle = \frac{1}{2\lambda}(3\langle \phi \rangle - 1). \tag{C.7a, b}$$

(iii)
$$\langle F \rangle = -\frac{3}{2\lambda}(\langle \phi \rangle - 1) + F(\langle \phi \rangle), \quad \langle (\phi - \langle \phi \rangle)F \rangle = -[\frac{3}{2\lambda^2} + \frac{1}{\lambda}(3\langle \phi \rangle^2 - \frac{9}{2}\langle \phi \rangle + 1)]. \tag{C.8a, b}$$

Figs. 6–8, respectively, show the obtained evolutions of the distributions for those three systems. In each figure, the upper and lower frames, (a) and (b), respectively, show the results for the exact computations and with the assumed PDFs (Gaussian). Here, the interval for the plots are set varying, respectively, as $\Delta t = 0.6$, 0.2, and 0.3 for those three cases. Different intervals are chosen for the ease of following the sequence of the evolution of the distributions for all three cases visually.

With the first dynamical system (i), the distribution peak moves towards $\phi = 0$, a stable fixed point of the system, with both exact and assumed–PDF based calculations, as seen in Fig. 6. The exact evolution (Fig. 6(a)) presents a strong tendency of the sharpening of the peak as well. This sharpening tendency is substantially weaker with the assumed–PDF calculation (Fig. 6(b)).

With the second system (ii), the distribution tends to spread with time in both directions, towards the two stable fixed points at $\phi = 0$ and $\phi \to +\infty$. Both calculations present an expected spreading tendency of the distribution with time, but in different manners. The exact calculation (Fig. 7(a)) shows that the distribution peak moves to a smaller $\phi$ with time, but at the same time, with a tendency of stretching a long tail toward a larger $\phi$. However, the calculation with the assumed Gaussian distribution

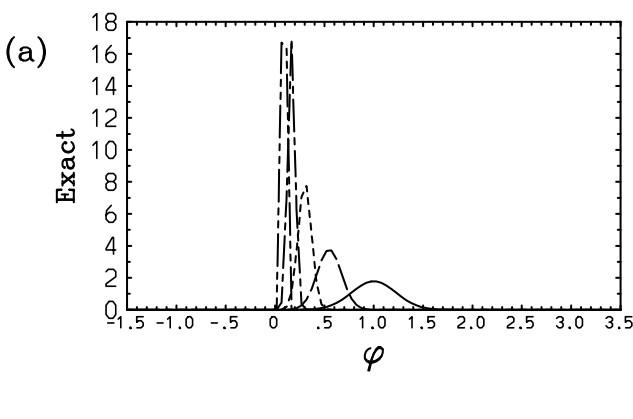

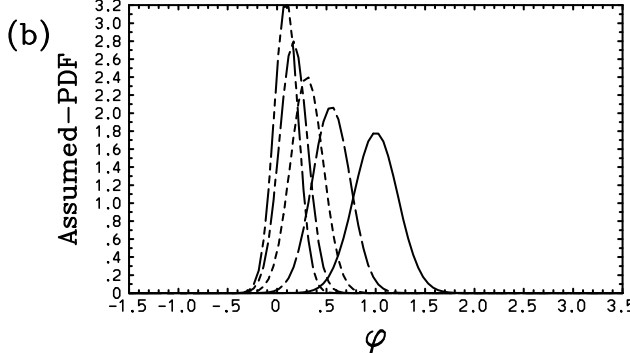

**Figure 6.** Evolution of the distribution for the model (i) with the interval of $\Delta t = 0.6$ with the initial distribution given by the solid line for (a) the exact solution and with (b) the assumed Gaussian distribution, shown with the varying types of the curves. In both cases, the distribution peak moves towards smaller values with time.

cannot reproduce the tendency of the distribution to be skewed with time by design. The assumed–PDF calculation (Fig. 7(b)), instead, shows that the distribution spreads by moving the peak toward a larger $\phi$ with time. The tendency of spread is less dramatic, too.

With the third dynamical system (iii), the system initialized in vicinity to the unstable fixed point, $\phi = 1$, tends to evolve towards either of the two stable fixed points at $\phi = 0$ and 2. As a result, as the exact calculation shows (Fig. 8(a)), an initial distribution peaked at $\phi = 1$ splits into the two peaks centered close to those two stable fixed points with time. Of course, it is not possible to reproduce such a tendency by assuming a Gaussian distribution as an assumed form. Thus, under the assumed–PDF calculation (Fig. 8(b)), we only see a tendency that the distribution gradually spread with time. One may also get an
impression that the rate of spread of the distribution is not sufficient enough to reproduce a drastic tendency towards the two peaks in exact calculation.

From the overview so far, it is clear that the assumed PDF approach can reproduce the actual evolution of the distribution in a manner less satisfactory even qualitatively. However, as it has been emphasized in the main text, the purpose of the assumed

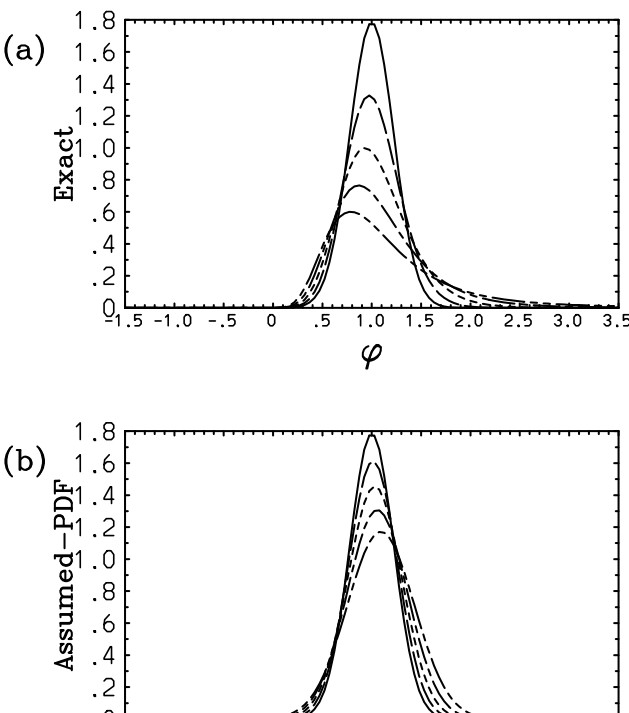

**Figure 7.** The same with Fig. 6, but for the model (ii) with the interval of $\Delta t = 0.2$. The peak moves towards smaller and larger values, respectively, with the exact and assumed distributions with this model.

PDF is not to predict the evolution of the whole distribution. Its purpose is solely limited to reproduce limited statistics that are specified under the output–constrained distribution principle. More specifically, under the given specific assumed–PDF formulation with the Gaussian, the sole goal is to predict the mean and the variance consistently. Thus, the main question to be addressed is how those statistics have been predicted.

To address this question, Fig. 9 (a) and (b), respectively, plot the time series of the man and the standard deviation with the assumed–PDF (black) and exact (green) calculations. The curves for the models (i), (ii), and (iii) are, respectively, shown by the solid, long, and short dashes. Here, the assumed PDF predicts the means of the models (i) and (iii) perfectly. The predictions of the standard deviations of these two models are overall consistent with the exact results, although the predicted tendencies are less pronounced than the exact results, especially with the model (i). On the other hand, the agreement of the evolution of the standard deviation with the model (iii: short dash) is still rather remarkable, considering the fact that the assumed PDF does not reproduce an overall evolution of the distribution even qualitatively. This provides another example, in addition to the case of Sec. 4.4.1, demonstrating that it is not indispensable to fit the distribution quantitatively well to obtain a realistic prediction of a required statistics.

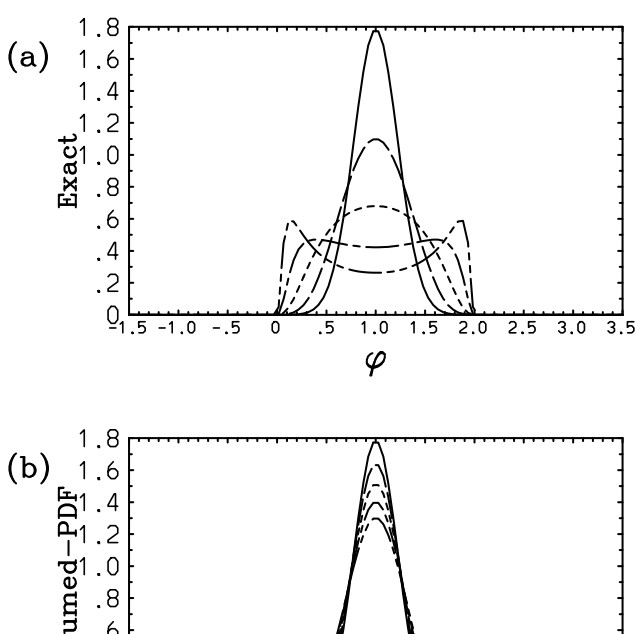

**Figure 8.** The same with Fig. 6, but for the model (iii) with the interval of $\Delta t = 0.4$. The distribution peak monotonously decreases with time in both cases.

In the case with the model (ii), the calculations with the exact distribution gets its own problem: the numerical integral of the distribution is extremely sensitive to the integral range and the total number of points used. In Fig. 9, the integral is performed over the range of -49 to 51 with $10^4$ points in respect to $\phi_0$. Examination of those sensitivities shows that the overall behavior of the curve (long–dash in green) is still correct, although the random–looking oscillatory behavior sensitively changes with the integral range and the number of points adopted.

Comparing the curves (long–dash) for the model (ii) with the exact (green) and assumed–PDF (black) calculations, we conclude that the assumed–PDF approach can predict the tendencies of the statistics (mean and variance) only up to $t \simeq 0.5$, but beyond that point, the actual evolution begins to dramatically deviate from the monotonously increasing tendencies predicted by the assumed–PDF method. The former shows that, due to the shift of the peak towards the smaller values with time, both the average and the standard deviation begin to decrease after $t \simeq 0.5$. This later transition is simply failed to be captured by the assumed–PDF solution.

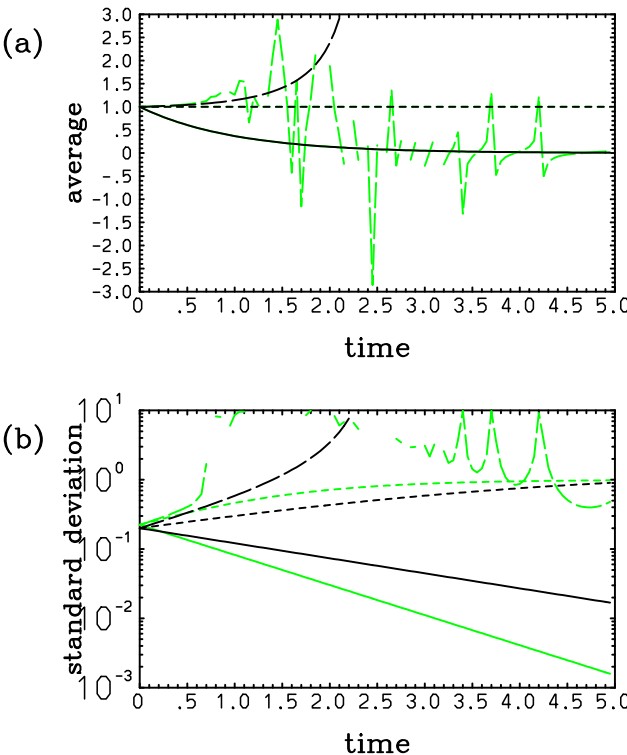

**Figure 9.** Statistics of the dynamical systems under considerations: (a) the average and (b) standard deviation for the models (i: solid), (ii, long dash), and (iii: short dash). Black curves are the results with the assumed PDF, and greens the results from direct integrals of the exact solutions. Here, the results of the means for the models (i) and (ii) with the assumed PDF agree perfect with the exact results.

*Author contributions.* The three authors contributed equally for the conceptualization, formal analysis, investigation, methodology, and writing.

*Competing interests.* The authors declare that they have no conflict of interest.

*Acknowledgements.* JIY thanks Wim Verkley for carefully reading an earlier version of this manuscript. Thanks are also due to Craig Bishop for his active inputs.

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
