# Peer review of "Technical Note: # General Formulation For the Distribution Problem: # Prognostic Assumed PDF Approach Based on"

_EGUsphere, 2023_

## Referee Comment (RC1)

**Review of "Technical Note: General Formulation For the Distribution Problem: Prognostic Assumed PDF Approach Based on The Maximum–Entropy Principle and The Liouville Equation" by Yano et al. (egusphere-2023-2278)**

This study presents the application and required mathematical background of an approach to predict a small number of relevant parameters that characterize the evolution of a distribution function rather than the full distribution's evolution, using the maximum entropy principle and Liouville equation. The manuscript shows that this approach is relevant for various aspects of the atmospheric sciences (subgrid-scale processes, hydrometeors, and assimilation) and highlights its applicability for the evolution of droplet size distributions under condensational growth. Overall, the paper is well written, interesting, and balances mathematical rigor with an educational introduction to the topic, i.e., an excellent *Technical Note* that deserves prompt publication in *Atmospheric Chemistry and Physics*.

All line numbers refer to the document that underwent the access review.

**Minor Comments**

Ll. 51 – 52: Lognomal or gamma distributions are frequently used to describe the droplet size distribution, especially for the condensation mode.

Ll. 56 – 52: I recommend referencing the work by Seifert and Beheng (2001; 2006) on cloud microphysics.

Ll. 195 – 196: Observational evidence is often used to determine distributions (e.g., Marshall and Palmer 1948).

Eqns. 3.2a, 3.17, 3.18: A definition of distributions as $p = dP/d\phi$ could be useful to the reader to understand subsequent derivations.

Ll. 665 – 668: While I get the argument, the example is not appropriate. While Z is indeed proportional to the radar reflectivity, it primarily represents the variance of droplet masses and is thus an important parameter to constrain the width of the droplet size distribution.

**Technical Comments**

Figures: Many figures miss units.

**References**

Seifert, A., & Beheng, K. D. (2001). A double-moment parameterization for simulating autoconversion, accretion and selfcollection. *Atmospheric Research*, *59*, 265-281.

Seifert, A., & Beheng, K. D. (2006). A two-moment cloud microphysics parameterization for mixed-phase clouds. Part 1: Model description. *Meteorology and Atmospheric Physics*, *92*(1-2), 45-66.

Marshall, J. S., & Palmer, W. M. K. (1948). The distribution of raindrops with size. *Journal of Atmospheric Sciences*, *5*(4), 165-166.

---

## Referee Comment (RC2)

**Review of "General Formulation For the Distribution Problem: Prognostic Assumed PDF Approach Based on The Maximum–Entropy Principle and The Liouville Equation" by Yano et al.**

This manuscript could be considered for publication after a major revision.

This study provides a unified approach to the distribution problem. By following the authors' procedure, we can systematically derive the time evolution equation closed in PDF parameters. However, I am curious how accurate and reliable the derived model is. I also think the theory needs some more clarification and sophistication. Here are some more general comments related to this point:

- To select the assumed PDF form, the authors advocate the use of the output-constrained maximum-entropy principle. This is a unique and interesting approach. But, I could not figure out how the host model determines the necessary variables (outputs). Please also see my comment (6) below.
- It is not clear how accurate the derived model can be. More systematic evaluation of the error is desired. If the assumed PDF is an exact solution of the original equation, and if the initial PDF follows the assumed form, there is no error. We may further expect that the error remains small even if the initial PDF does not follow the assumed form. I believe these points should be stressed more explicitly. Then, what if the assumed PDF is not an exact solution? Intuitively, this should cause a deviation from the true solution. To assess the reliability of the derived model, it is critical to understand how fast the error grows in time. It would be difficult, but please provide a more careful discussion on this point.
- The advantage of using the maximum-entropy principle is not fully clear to me. I agree it is a convenient way to estimate the PDF from. At the same time, the theory presented in Sec.5.1 can be applied to any PDF form. I suppose the authors are implicitly assuming that the error is smaller if we choose the PDF form based on the maximum-entropy principle, but this is not at all trivial. It would be very interesting if the authors could prove or demonstrate this.
- The examples presented in the manuscript are rather simple. As long as the authors declare that Eq.(2.1) is their ultimate target of the theory, it is desirable to present some examples based on Eq.(2.1).
- The authors repeatedly stress that the theory can also be applied to subgrid–scale modeling, and data assimilation. To provide better insight and perspective, the ideas the authors have in mind should be formulated more explicitly.

The manuscript is well organized, but sometimes lacks clarity and, in my opinion, there is some inconsistency in the notation and logical flow.

I believe the quality of the study will be significantly enhanced if these points are addressed.

**Major Comments**

1) **[request] P.4 ll.109–115 "Typically, as argued by Yano et al. (2005) …"**

Here, the governing equation system of this study is introduced, but the description is ambiguous and confusing.

- It is not clear what "Typically" at the beginning indicates. I feel it is confusing because "typically any" does not make sense to me. Please consider removing it.
- Please clarify that $\phi$ in Eq.(2.1) is defined on the real space $(x, y, z)$.
- Consider, e.g., vapor mixing ratio $\phi = q_v$, then, there is a diffusion term and the governing equation does not fall into the form of Eq.(2.1). Or, does the source term $F$ also take care of the diffusion term? Please clarify this point.

2) **[request] Eq.(2.2) and p.5 ll.123–124 "As a specific example, …"**

This part is also confusing. If we consider bulk cloud microphysics models, the time evolution equation of cloud water mixing ratio $q_c$ falls into Eq.(2.2) if there is no wind and the cloud droplet sedimentation is ignored. Still, $q_c$ is a field variable defined on the real space $(x, y, z)$, i.e., $q_c = q_c(x, y, z, t)$. However, if we consider the condensational growth of a cloud droplet, $dr/dt = 1/r$, the droplet radius $r$ is not a field variable but just a function of $t$, i.e., $r = r(t)$. In other words, for $q_c$, we can consider Eq.(2.2) is an approximated form of Eq.(2.1), but for droplet radius $r$, there is no equation corresponding to Eq.(2.1). To avoid confusion, when introducing Eq.(2.2), the authors should explain that $\phi$ may not be a field variable anymore, and that Eq.(2.1) does not exist for such variables.

3) **[comment] P.6 l.153 "…, there is no closed analytical formula for reconstructing the original distribution from a given series of moments: …"**

It seems to me that we can derive an approximation of the moment generating function from the series of moments, then we can estimate the true PDF by using, e.g., the saddlepoint approximation (Daniels (1954) and Butler (2007)). I am also curious how efficiently the maximum entropy principle can estimate the true PDF from a given series of moments compared to other methods such as the above.

4) **[request] P.6 ll.176–177 "The prognostic equations for these moments, or diagnostic approximations of these equations, are, in turn, known from the turbulence theories; …"**

This only applies to subgrid-scale turbulence problems. Further, in general, we cannot derive the prognostic equations for moments closed in moments only from Eq.(2.2). Please rephrase the sentence appropriately.

5) **[request] P.12 ll.347–349 "More general formulations for the partial-differential equations (PDE) …"**

Because the authors are thinking that Eq.(2.1) is the ultimate application of the present theory, and also because it is not trivial, the authors should show the Liouville equation of Eq.(2.1).

6) **[question] Sec.4.2 "Output–Constrained Distribution Principle"**

Interesting idea, but I do not fully agree. How can we specify the outputs necessary for the host model? For cloud microphysics, in one-moment bulk schemes, the typical prognostic variables are $q_c$ and $q_r$. For two-moment bulk schemes, $n_c$ and $n_r$ are added. From a cloud microphysical consideration, this makes some sense. But, how can we tell the optimal outputs necessary for the host model without the knowledge in cloud microphysics?

7) **[request] Eq.(4.1)**

Please mention the $F$ in Eq.(2.2) corresponding to Eq.(4.1) is the Brownian motion. Please also explain how we can apply the Liouville equation (3.24) if $F$ is noisy. (I think it is more common to call it the Fokker-Planck equation.)

8) **[request] P.16, l.465 "..., it suffices to take a Gaussian distribution, ..."**

The authors should mention that Gaussian is the exact solution of the diffusion equation.

9) **[request] P.21, ll.573–576 "As in the case with the Gaussian equation, more generally, when the assumed PDF form constitutes an exact solution of a given system, ..."**

I think this is a very important and plausible remark. Could you provide a mathematical proof of this?

10) **[question] P.22, ll.618–619 "Eq. (5.8a) or (5.10b) further simply reduces to a diagnostic method based on moments, ..."**

What do you mean by "diagnostic method based on moments"? Please elaborate.

11) **[comment] Sec.5.1 "General Formulation"**

The formulation in this section does not rely on the PDF form (3.15) derived from the maximum-entropy principle. From eq. (3.15), we can derive more specific relations such as $\partial p/\partial \lambda_i = -\sigma_i p$. It would be beneficial to simplify the formulae in Sec.5.1 further by applying such relations.

12) **[question] P.24 ll.666–667 "..., it is not directly required in any microphysical tendencies within a model."**

In standard warm phase cloud bulk schemes, not water mixing ratio $q$, but cloud water mixing ratio $q_c$ and rain water mixing ratio $q_r$ are the prognostic variables. How can we justify this from the output-constrained distribution principle? Or, do the authors think just $q$ is sufficient?

13) **[request] P.25 l.699 "..., setting the weight as $\sigma_1 = \phi^n$ ..."**

This is confusing. It seems the authors are still assuming $p = \lambda_1 \exp(-\lambda_1 \phi)$ (not $\lambda_0 \exp(-\lambda_1 \phi^n)$), but use $\sigma_1 = \phi^n$ when deriving the time evolution equation of $\lambda_1$. However, as derived in Eq.(3.15) by the authors, exponential distribution is obtained from the maximum–entropy principle when the system is constrained by the mean, i.e., $\sigma_1 = \phi$. Please clarify the reasoning why $\sigma_1$ other than $\phi$ is being considered here.

**Minor Comments**

**14) [request] P.2, ll.30–31 "Here, it is hard to overemphasize the clear difference between these two distributions."**

Please define DDF explicitly. The readers can eventually understand that DDF is not normalized but PDF is normalized, but it should be clarified when introduced for the first time.

**15) [request] P.9 Eq.(3.15) $p_0$**

Please relate $p_0$ to $\lambda_o$. $p_0 = \exp(-\lambda_0)$?

**16) [comment] P.13 Eq.(3.5)**

I think the correct equation is

$$\frac{\partial n}{\partial t} + \nabla_h \cdot (\boldsymbol{u}n) + \frac{\partial}{\partial z}[(w - w_t)n] = S.$$

**17) [comment] P.13 Eq.(3.6)**

$r - r'$ has to be $\left(r^3 - r'^3\right)^{1/3}$. If the authors are talking about collision-coalescence, (1/2) is needed for the first term on the r.h.s.

**18) [request] P.20 Eq.(5.2a)**

Please clarify that the definition of $\lambda_0$ has been altered from that provided in Eq.(3.14).

**19) [question] P.25 Eqs.(5.16a) and (5.16b)**

I think something is wrong with these equations. We can derive

$$\frac{d}{dt}\langle \phi^n \rangle = \int \phi^n \frac{\partial p}{\partial t} d\phi = n \int p\phi^{n-1} F d\phi = n\langle \phi^{n-1} F \rangle.$$

But, obviously, this is not consistent with Eq.(5.16b).

**Typo**

**20) P.3 l.77 "operation numerical forecasts" -> "operational numerical forecasts"**

**21) P.7 l.197 "form" -> "from"**

**22) P.10 ll.348–349 "..., then there results a gamma distribution." -> "..., then the result is a gamma distribution."??**

**23) P.19 l.534 "with by" -> "with"?**

**24) P.20 l.561 "the right–hand side of Eq. (3.11)" -> "the right–hand side of Eq. (5.4)"**

**25) P.21 l.525 "of of"**

**26) P.24 l.665 "form"**

**27) P.25 Eq.(5.15b) and (5.16a) "dt>"**

**References**

H. E. Daniels. "Saddlepoint Approximations in Statistics." Ann. Math. Statist. 25 (4) 631 - 650, December, 1954. https://doi.org/10.1214/aoms/1177728652

Butler RW. Saddlepoint Approximations with Applications. Cambridge University Press; 2007. https://doi.org/10.1017/CBO9780511619083

---

## Community Comment (CC1)

**Response to the Reviewer RC1**

We much appreciate a positive review by the present Reviewer concluidng that "Overall, the paper is well written, interesting, and balances mathematical rigor with an educational introduction to the topic, i.e., an excellent Technical Note that deserves prompt publication in *Atmospheric Chemistry and Physics*."

We respond to the more specific comments as follows:

**Minor Comments**

● Ll.51–51: We will add in revision that the gamma distributions is also often adopted in bulk microphysics. On the other hand, we are not aware that the lognormal distribution is also used in bulk microphysics.

● Ll.56: The references to Seifert and Beheng (2001, 2006) will be added in revision.

● Ll.195–196: In revision, the allusion to Marshall and Palmer (1948) will definitely be made.

However, we notice that the main problem with this short section 3.2.2 as a whole: it begins by discussing about the issues of identifying appropriate assumed PDF forms in the context of the subgrid–scale distribution problem. However, it fails to specify this context in the beginning. Then, it suddenly turns the topic to the PSD in microphysics. In revision, Sec. 3.2.2 will be divided into the two paragraphs, with the first paragraph focusing on the subgrid–scale distribution problem and the second paragraph focusing on the PSD.

More specifically, Marshall and Palmer (1948) will be referred in the beginning of the second paragraph.

● Eqns. 3.2a, 3.17, 3.18: By following the comment, the cumulative probability, $P(\varphi' < \varphi)$, will be introduced in revision, and its relation to the probability density, $p = dP/d\varphi$, will be explicitly listed just following Eq. (3.2a). As the Reviewer also suggests, this relation will be recalled in presenting Eqs. (3.17) and (3.18) in revision as well.

● Ll.665–668: From the point of view of the output–constrained distribution principle, the Reviewer's argument for using the 6th moment in the particle size is consistent, only if a mass distribution is considered for the problem. In that case, the 6th moment in the particle size corresponds to the second moment in mass distribution. Thus, this second moment must be used according to the output–constrained distribution principle. when one wishes to constrain the spread of this distribution. However, when a size distribution is considered, the spread of the distribution would be considered in terms of a variance in size distribution, which is the second moment of the size.

Yet, from a point of view of the output–constrained distribution principle, a more important factor is to choose an actual output that is required within a given model. For the cloud particles, probably, the most important process to be predicted is the coalescence, which is very crudely speaking, controlled by $n_c^2$, thus a weight to adopt would be $\sigma = n_c$, noting there is already a factor, $n_c$, in the definition of the integral with sigma. For the precipitating particles, the same would apply to the sedimentation rate, which is proportional to a certain power, say, $a$, of the particle size, $r$, then $\sigma = r^a$ would be the choice.

These elaborations will be included in the revision.

**Technical Comments**

● Figures: Please note that all the variables in the present study are nondimensional (*i.e.*, without units). This basic point will be remarked to the end of the revised introduction.

---

## Author Comment (AC1)

**Response to the Reviewer RC2 (Referee 1)**

*Structure of the Response*

Please note that in the following response, the Reviewer comments are quoted by »...«.

*General Remarks:*

We much appreciate a very thorough examination of our manuscript by the present Reviewer. We also acknowledge that the present Reviewer has revealed various critical issues, that would not have been noticed otherwise.

Most importantly, we are glad with the present Reviewer's conclusion that »This manuscript could be considered for publication after a major revision.«

After summarizing our work by the first two sentences of the second paragraph, the present Reviewer yet lists several questions to be clarified. We first respond to those listed items:

- *Output-constrained maximum-entropy principle*:

In application of the output-constrained maximum-entropy principle, the Reviewer questions »how the host model determines the necessary variables (outputs)«. Here, we identify the two separate issues behind: the first is the fact that this principle literally works only when a distribution–based approach is adopted for a subgrid–scale modeling problem. In the context of the cloud microphysics and data assimilation, this notion of the »necessary variables (outputs)« for the host model must be generalized from its literal meaning. Second, more fundamentally, the notion of the »necessary variables (outputs)« for the host model is not well explained even in the context of the subgrid–scale modeling problem. These two issues will be further elaborated in revision.

Here, recall that the purpose of the subgrid–scale modeling/parameterization is to provide certain specific grid–averaged quantities to the host model. In the convection parameterization problem, those are called the apparent sources, $Q_1$ and $Q_2$, *i.e.*, tendencies of the temperature and moisture due to the subgrid–scale processes. All the other details are only for a purpose of a consistent calculation of the subgrid–scale processes.

In case of the clouds microphysics with explicit cloud modeling (thus the cloud processes themselves are not "parameterized"), certain variables must be passed over to different components of the model, that plays a role of "host model" in this context. For example, the mixing ratios, $q_c$ and $q_r$, of clouds and rain must be counted for an accurate definition of the buoyancy in the momentum equation. Some radiation schemes require inputs of mean radius, $r_c$ and $r_p$, of the cloud and rain droplets, although those are typically *not* prognostic variables of the cloud microphysics. Those variables are considered to be "the necessary variables (outputs) for the host model".

The case of data assimilation is more subtle, because there is neither a host model nor another model components to which information must be passed around. Yet, for the operational purposes, we are not interested to know a

full shape of a probability distribution of a variable in order to quantify the uncertainty. In traditional assimilation formulations, we merely asks for the standard–deviation errors/uncertainties of variables: those are considered the "necessary outputs" for the data assimilation.

- *Accuracy the derived model*:

As the Reviewer correctly points out »It is not clear how accurate the derived model can be. More systematic evaluation of the error is desired.« In the present study, only a preliminary evaluation of the method is presented for the simplest case in Sec. 6. Further evaluations are performed in Yano (2024), which appeared online only after the submission of the present manuscript. The reference to Yano (2024) will be added in revision.

By following the Reviewers' suggestion, in the revised text, the following more basic points will be added: »If the assumed PDF is an exact solution of the original equation, and if the initial PDF follows the assumed form, there is no error. We may further expect that the error remains small even if the initial PDF does not follow the assumed form.«

In general, unfortunately, there is no obvious methodology for predicting the potential errors of the methodology. It appears to us that the only feasible approach is to run a model explicitly for an evaluation. For this reason, it is not possible for us to »provide a more careful discussion on this point« at this point. This remark will also be added in revision.

- *Advantage of using the maximum-entropy principle*:

Although here the Reviewer states that »The advantage of using the maximum-entropy principle is not fully clear to me«, the issue to be fully clarified is not quite well stated in the comments. Instead, the Reviewer appears to be rather supportive to this principle: »I agree it is a convenient way to estimate the PDF from. At the same time, the theory presented in Sec.5.1 can be applied to any PDF form.« We do not assume that »the error is smaller if we choose the PDF form based on the maximum-entropy principle« even implicitly. Exactly as the Reviewer remarks, because »this is not at all trivial«. Please refer to the discussions over L205–220 and the references therein for more.

- *Examples presented in the manuscript*:

As the Reviewer states, »The examples presented in the manuscript are rather simple.« We believe that these are legitimate choices considering the main goal of the present manuscript presenting the principles, rather than proving them. Also as the Reviewer correctly points out, »Eq.(2.1) is their ultimate target of the theory«. Yet, such a full development is still far from the present state of the development, as can also be perceived by the sequel paper (Yano 2024). Thus, presenting any »examples based on on Eq.(2.1)« is also just too premature at this stage.

- *Applicability to both subgrid–scale modeling and data assimilation*:

Our argument for the applicability of the proposed formulation to both »subgrid–scale modeling and data assimilation« merely remains a formal level (See L27–37): we propose a general formulation for solving the distribution problem. This point must be clear for the all. Since the problem of both »subgrid–scale modeling and data assimilation« reduce to that of the distribution in space and of the probability, respectively, it is also natural to claim that the present formulation is applicable to both of those problems. Issues of formulating the subgrid–scale modelings as a distribution problem is already extensively discussed in Yano (2016): though this paper is already cited, in revision, this very point will more be explicitly stated. A full formulation based of the data assimilation under the present framework is still to be fully developed, yet a preliminary note is already available: https://drive.google.com/file/d/1i1NowEip69t5LdUOdZlBrdxDtm6wZyXI/view?usp=drive_link. However, this material is not yet at an appropriate state to be quoted in a more formal manner.

Readers are advised to refer to the references cited in the paragraph over L55–59. The lead sentence of this paragraph will be modified in revision to make is clear where they can find necessary references to understand how the distributions are applied to those three problems.

The present Reviewer concludes the general remarks by stating that »The manuscript is well organized«. Yet, the Reviewer also points out that it »sometimes lacks clarity;« and »some inconsistency in the notation and logical flow«: these issues will be addressed in revision by fully considering the further comments by the present Reviewer in the following.

**Major Comments**

The Reviewer suggests that »the quality of the study will be significantly enhanced« by addressing the following Major Issues. To those we respond as follow.

*1) [request] P.4 ll.109–115 "Typically, as argued by Yano et al. (2005) . . . ":*

As the Reviewer correctly points out, the current sentence (L109–110) leading to Eq. (2.1) is »ambiguous and confusing«. Also as the Reviewer correctly points out, not all the physical variables considered in atmospheric science take the form of Eq. (2.1). A good example is the radius, $r$, of the water droplets as considered in Sec. 6, as the Reviewer points out. It would even not be fair to argue that "typically" the governing equations for the physical variables take the form of Eq. (2.1), and others are "exceptions".

A more precise statement would be that many dependent variables in atmosphere depend on both space and time such as the temperature, moisture (water–vapor mixing ratio), etc: those variables are advected by the wind (including the wind itself), as represented by Eq. (2.1): the lead sentence in concern will be modified accordingly in revision. Yano *et al.* (2005) and Yano (2016) show that the basic formulations for the subgrid–scale modelings (parameterizations) can be reproduced by simply examining this general form (2.1). More specifically, Yano (2014) shows that all the essential, basic standard formulas for the mass–flux convection parameterizaiton can be reproduced by only considering Eq. (2.1). These elaborations will also be added in revision.

Finally, the source term, $F$, simply includes all the physical tendencies of a variable, $\phi$, apart from the advection tendency. This remark will be added in revision, too. Concerning the possibility of $F$ containing spatial derivatives, please refer to L125—126.

*2) [request] Eq.(2.2) and p.5 ll.123–124 "As a specific example, ...":*

The Reviewer argues that the introduction of Eq. (2.2) is also »confusing«: however, we are rather puzzle with this. The present Reviewer strangely attaches some physical significance to Eq. (2.2), when there is no such suggestion is made in the text. This equation in concern is introduced by merely "for ease of the deductions" (L116–117). Probably, this lead sentence was too terse to avoid any misunderstanding, thus will be further elaborated in revision.

Please also note that the discussion concerning the condensation growth in the last half of the same paragraph (L120–125) is not directly linked to Eq. (2.2), but more about the generality of the source term, $F$, without specifying it in the present study. To avoid this confusion, this part will be made a standalone paragraph in revision.

*3) [comment] P.6 l.153 "..., there is no closed analytical formula for reconstructing the original distribution from a given series of moments: ...":*

We still believe that this statement is correct. Of course, there are many formulas that link between the moments and the corresponding distribution. A particular, general category is called the "generators", because this function, defined from a given distribution, can generate the corresponding moments in a sequential manner. Here, what the present Reviewer points out is such an example that can be obtained under the saddlepoint approximation. However, please note that as the case with any other generators, this version of generators can generate the moments from a given distribution, but not other way round.

Please also note that the maximum entropy principle is based on a completely different principle: it *does not* estimate nor approximate a given distribution, although the moments may be used for this purpose. This principle simply derives the "most likely", with its meaning carefully discussed in Sec. 3.3 of the manuscript, for a given particular system when the constraints to this system is known. However, there is no guarantee that the system follow this distribution (L209–210): it merely suggests to be "most likely".

*4) [request] P.6 ll.176–177 "The prognostic equations for these moments, or diagnostic approximations of these equations, are, in turn, known from the turbulence theories; ...":*

Here, this is another example that the discussions of the manuscript tends to be biased towards the subgrid–scale distribution problem due to the lead author's interest. In revision, the clause of "in the context of the subgie–scale distribution problem" will be added for the clarity. We are afraid that no equivalent procedure is known for both the cloud microphysics and data assimilation.

*5) [request] P.12 ll.347–349 "More general formulations for the partial-differential equations (PDE) ...":*

We disagree with the request by the present Reviewer to explicitly present the prediction equation for the distribution of variables governed by Eq. (2.1) for the three reasons: 1) although Eq. (2.1) is the ultimate application, it is not at all considered in the present manuscript; 2) the probability equation for the system (2.1) is fairly complicated, and nothing would be effectively understood just by looking at this equation (see *e.g.*, Eq. 15 of Larson 2004); 3) under the assumed prognostic PDF approaches, as discussed in Sec. 5.1, immediately after Eq. (5.10b), prognostic equations for the PDF parameters can be derived directly from the governing equation (2.1). Thus, there is no need to consider the probability equation for the system (2.1) in the end, as pointed out in Sec. 5.1. This point is already remarked in Sec. 2: see L132–133.

To make this last point better stands out, in revision, the corresponding discussion in Sec. 5.1 will be expanded into a standalone short subsection.

At the same time, for the satisfaction of curiosity of the present Reviewer as well as some readers, we will directly refer to Eq. (15) of Larson (2004) for its explicit form in revision: this reference must be readily accessible for most of the ACP readers.

6) *[question] Sec.4.2 "Output–Constrained Distribution Principle"*:

As stated in response to the item "Output-constrained maximum-entropy principle" in general remarks, we identify the two separate issues behind: the first is the fact that this principle literally works only when a distribution–based approach is adopted for a subgrid–scale modeling problem. In the context of the cloud microphysics and data assimilation, this notion of the »necessary variables (outputs)« for the host model must be generalized than its literal meaning. Second, more fundamentally, the notion of the "necessary outputs for the host model" is not well explained even in the context of the subgrid–scale modeling problem. These two issues will be further elaborated in revision.

More specifically, in the context of the cloud modeling, the "necessary output variables for the host model" are *not identical* to the prognostic variables used in a cloud model. The question here is what variables are required as output from the cloud model for the whole system. Clearly the mixing ratios, $q_c$ and $q_r$, for the cloud and rain are important for defining the buoyancy that drives the momentum equation, for example. On the other hand, it is less obvious where the system would require the number densities, $n_c$ and $n_r$, of the cloud and rain: a certain radiation scheme may required this, but not always. In other words, although $n_c$ and $n_r$ are the prognostic variables of the cloud model, these may not be output variables required in the host model.

7) *[request] Eq.(4.1)*:

In revision, it has been mentioned that »$F$ in Eq.(2.2) corresponding to Eq.(4.1) is the Brownian motion«, and also that Eq. (4.1) is a special case of the Fokker–Planck equation.

In the present study, $F$ is assumed to be deterministic, as already suggested in Sec. 3.5 (L344), and also to be stated earlier in Sec. 2 in revision. Generalization

with stochasticity is already remarked at L347 as well as L960–962.

8) *[request] P.16, l.465 "..., it suffices to take a Gaussian distribution, . . . "*:
By following the request of the Reviewer, in revision, the following remark will be added immediately following Eq. (4.1): Note that in this particular case, the adopted distribution form also corresponds to an exact solution of the system (4.1).

9) *[request] P.21, ll.573–576 "As in the case with the Gaussian equation, more generally, when the assumed PDF form constitutes an exact solution of a given system, . . . "*:
This point can be understood directly from the fact that Eq. (5.7a) is equivalent to the original Liouville equation (3.24) under the given assumed PDF form. This remark will be added in revision, by following the request of the present Reviewer.

10) *[question] P.22, ll.618–619 "Eq. (5.8a) or (5.10b) further simply reduces to a diagnostic method based on moments, . . . "*:
The phrase "diagnostic method based on moments" will be modified in revision as "diagnostic method based on moments typically adopted in the subgrid–scale assumed PDF formulations".

11) *[comment] Sec.5.1 "General Formulation"*:
As stated in the introduction, the present study addresses the two major open questions associated with the assumed PDF approaches: 1) how the assumed PDF form can be determined? and 2) how the parameters for the assumed PDF can be predicted consistently? As emphasized in the concluding section (L933–940), these two questions are addressed separately in the present study. Thus, the prognostic assumed–PDF formulation presented in Sec. 5 does not necessarily follow from the assumed PDF form defined by the output–constrained maximum entropy introduced in Sec. 4, as the Reviewer correctly points out here: See L89–97. This point will also be made more explicit in the introduction as well as in the beginning of Sec. 5 in revision.

It also follows that the relation, $\partial p/\partial \lambda_i = -\sigma_i p$, expected from the maximum entropy principle (3.15), is only a special case of the general formulation considered in Sec. 5. Also considering the fact that this reduction does not much simplify the formulation (it still takes about the same space in the equations), we will not introduce this simplification in revision, although the Reviewer suggests to do so.

12) *[question] P.24 ll.666–667 "..., it is not directly required in any microphysical tendencies within a model."*:
We are afraid that the Reviewer is bit confused with this sentence: "it" here refers to the reflectivity, $Z$, rather than the mixing ratios, $q_c$ and $q_r$, as the Reviewer somehow assumes. To avoids this confusion, "it" will be replaced by "the reflectivity, $Z$" in revision. Please refer to our response to the item 6), if

more background issues must be addressed.

13) *[request] P.25 l.699 "..., setting the weight as $\sigma_1 = \phi^n$ ...":*
See our response to the item 11) above for the general matters.

There is no confusion here, once one understands that the determination of the assumed PDF form and the prediction of the assumed–PDF parameters are mutually independent procedures. Note especially that the output–constrained maximum entropy principle is merely a guiding principle, but *not* a physical principle that the system must satisfy: *cf.*, the discussion over L205–214. Thus, we can choose a constraint, $\sigma_1$, for a given distribution, that is *not dictated* by the output–constrained maximum entropy principle, and without contradicting with any physics.

Putting it differently, although the exponential distribution is derived by assuming the mean as a sole output variable (*i.e.*, constraint); the general formulation in Sec. 5 can be applied to any $\sigma_1$, which may not necessarily correspond the assumed output variable (*or* constraint), that is used for deriving the given distribution. The question that we pose here is that how sensitive the evolution of the distribution by trying to predict a different statistical quantity, $\langle \sigma_1 \rangle$, consistently, based on Eq. (5.10a). This minor exercise is very worthwhile to show, because the result is quite sensitive to the choice of the weight, $\sigma_1$, as shown in Fig. 2. [Please note that due the error pointed out in the item 19) by the Reviewer, Fig. 2 must be corrected.]

**Minor Comments**
*14) [request] P.2, ll.30–31 "Here, it is hard to overemphasize the clear difference between these two distributions.":*
The point here is very simple: distribution and probability are the two distinctively different concepts, and this simple fact must be well respected. Note that neither a (frequency) distribution of a subgrid–scale variable nor a size distribution of hydrometeor particles is a probability. Conversely, the probability reduces neither to any *simple* distribution, nothing to do with a probability. The text will be elaborated in revision.

*15) [request] P.9 Eq.(3.15) $p_0$:*
Yes, it is $p_0 = \exp(-\lambda_0)$, as will be remarked in revision.

*16) [comment] P.13 Eq.(3.25):*
Eq. (3.25) will be modified from an advection form to a flux form as suggested in revision.

*17) [comment] P.13 Eq.(3.26b):*
We much appreciate the present Reviewer for pointing us out the mistakes concerning the stochastic collection equation (3.26b). As pointed out, in revision, the argument of the equation will be modified from the size, $r$, to the mass, $m$. Also a missing factor $1/2$ in front of fhe first term on the right–hand side will be added.

*18) [request] P.20 Eq.(5.2a)*:

We note certain difficulties for using the two notations, $\lambda_0$ and $p_0$, for the normalization factor of a distribution, as pointed out by the Reviewer here. Yet, we are inclined to stick to this "double standard" considering the advantages of both notions: the choice of $\lambda_0$ as a normalization factor in discussing the assumed–PDF in general manner, as in Sec. 3.2, has an unbeatable advantage to treat all the parameters of a distribution with a single notation, $\lambda_i$. On the other hand, in more specific situations, such as in Eq. (3.15), it is more intuitive to adopt the notion of $p_0$ for a normalization constant. Note that in those situations, other PDF parameters also often take different notations than $\lambda_i$ for the same reason.

*19) [question] P.25 Eqs.(5.16a) and (5.16b)*:

This is a very sharp observation by the Reviewer!: of course, the original Eq. (5.16b) was wrong, and it must become

$$\frac{d}{dt}\langle \phi^n \rangle = \int \phi^n \frac{\partial p}{\partial t} d\phi = n \int p\phi^{n-1} F d\phi$$

for consistency, as the Reviewer points out. The reduction of this part was carefully re–examined, and the errors were identified: after those corrections, Eq. (5.16b) indeed reduces to the above. As a consequence, Eq. (5.16a) must also be modified into:

$$\lambda_1(t) = \left[ \frac{1}{\lambda_1^n(0)} + \frac{n}{n!} \int_0^t \langle F\phi^{n-1} \rangle dt \right]^{-1/n} .$$

Those modifications will be applied in revision.

**Typo**

20)–27) We much appreciate the various typos pointed out by the present Reviewer. All those typos will be corrected in preparing the final manuscript.

**References**

Yano, J.-I.: Prognostic Assumed-PDF (DDF) Approach: Further Generalization and Demonstrations, EGUsphere [preprint], https://doi.org/10.5194/egusphere-2024-287, 2024.

---

## Author Response (AR1)

**Final Response to the Reviewers**

**General Remarks**

The following response is essentially identical to what we responded to each
Reviewer in the interactive discussion phase, apart from some wording changes
to make it a final as well as additional minor edit.

Please note that in the following response, the Review texts are quoted by »...«.

**Major Additional Modification**

Along with the modifications in response to the two reviewers, there is another
major modification in preparing this final manuscript: In this revision process, it
has been realized that in Sec. 6, it is possible to derive the exact solution for the
evolution of the droplet–size distribution of the condensation–growth problem in
a closed form without additional numerical integrals. This modification has also
revealed an error in an original analysis: the exact evolution of the distribution,
now, takes a form of a propagation of a shock wave, as seen in the revised Fig. 3.
Yet, in spite of this qualitative change of the result, the overall conclusions in this
section do not change, except for another notable point that the assumed gamma
distribution predicts the mean radius much better than previously assessed (*cf.*,
Fig. 4).

**Response to the Reviewer RC1**

We much appreciate a positive review by the present Reviewer concluidng that
»Overall, the paper is well written, interesting, and balances mathematical rigor
with an educational introduction to the topic, i.e., an excellent Technical Note
that deserves prompt publication in *Atmospheric Chemistry and Physics*.«

We respond to the more specific comments as follows:

**Minor Comments**

• Ll.51–51 (Ll.50–51 in revision): We will add in revision that the gamma
distributions is also often adopted in bulk microphysics. On the other hand, we
are not aware that the lognormal distribution is also used in bulk microphysics.

• Ll.56 (Ll.56 in revision): The references to Seifert and Beheng (2001, 2006)
have been added in revision.

• Ll.195–196 (Ll.224–225 in revision): In revision, the allusion to Marshall and
Palmer (1948) has been included in revision.

However, we notice that the main problem with this short section 3.2.2 as a
whole: it begins by discussing about the issues of identifying appropriate as-
sumed PDF forms in the context of the subgrid–scale distribution problem.
However, it fails to specify this context in the beginning. Then, it suddenly turns
the topic to the PSD in microphysics. In revision, Sec. 3.2.2 has been divided
into the two paragraphs, with the first paragraph focusing on the subgrid–scale
distribution problem and the second paragraph focusing on the PSD.

More specifically, Marshall and Palmer (1948) have been referred in the beginning of the second paragraph.

● Eqns. 3.2a, 3.17, 3.18: By following the comment, the cumulative probability, $P(\varphi' < \varphi)$, has been introduced in revision, and its relation to the probability density, $p = dP/d\varphi$, has been explicitly listed just following Eq. (3.2a). As the Reviewer also suggests, this relation has been recalled in presenting Eqs. (3.17) and (3.18) in revision as well (L322).

● Ll.665–668: From the point of view of the output–constrained distribution principle, the Reviewer's argument for using the 6th moment in the particle size is consistent, only if a mass distribution is considered for the problem. In that case, the 6th moment in the particle size corresponds to the second moment in mass distribution. Thus, this second moment must be used according to the output–constrained distribution principle, when one wishes to constrain the spread of this distribution. However, when a size distribution is considered, the spread of the distribution would be considered in terms of a variance in size distribution, which is the second moment of the size.

Yet, from a point of view of the output–constrained distribution principle, a more important factor is to choose an actual output that is required within a given model. For the cloud particles, probably, the most important process to be predicted is the coalescence, which is very crudely speaking, controlled by $n_c^2$, thus a weight to adopt would be $\sigma = n_c$, noting there is already a factor, $n_c$, in the definition of the integral with sigma. For the precipitating particles, the same would apply to the sedimentation rate, which is proportional to a certain power, say, $a$, of the particle size, $r$, then $\sigma = r^a$ would be the choice.

These elaborations have been included in the revision as a footnote to the end of Sec. 5.2. This choice is to avoid this subsection to be overwhelmed by these microphysical elaborations, although those caveats are crucial to be mentioned, because many microphycsists would pose the same questions as the present Reviewer poses here.

**Technical Comments**
● Figures: Please note that all the variables in the present study are nondimensional (*i.e.*, without units). This basic point has been remarked to the end of the revised introduction (L112–114).

**Response to the Reviewer RC2 (Referee 1)**

*General Remarks:*
We much appreciate a very thorough examination of our manuscript by the present Reviewer. We also acknowledge that the present Reviewer has revealed various critical issues, that would not have been noticed otherwise.

Most importantly, we are glad with the present Reviewer's conclusion that »This manuscript could be considered for publication after a major revision.«

After summarizing our work by the first two sentences of the second paragraph,

the present Reviewer yet lists several questions to be clarified. We first respond to those listed items:

- *Output-constrained maximum-entropy principle*:

In application of the output-constrained maximum-entropy principle, the Reviewer questions »how the host model determines the necessary variables (outputs)«. Here, we identify the two separate issues behind: the first is the fact that this principle literally works only when a distribution–based approach is adopted for a subgrid–scale modeling problem. In the context of the cloud microphysics and data assimilation, this notion of the »necessary variables (outputs)« for the host model must be generalized than its literal meaning. Second, more fundamentally, the notion of the »necessary variables (outputs)« for the host model is not well explained even in the context of the subgrid–scale modeling problem. These two issues have been further elaborated in revision: L462–477.

Here, recall that the purpose of the subgrid–scale modeling/parameterization is to provide certain specific grid–averaged quantities to the host model. In the convection parameterization problem, those are called the apparent sources, $Q_1$ and $Q_2$, *i.e.*, tendencies of the temperature and moisture due to the subgrid–scale processes. All the other details are only for a purpose of a consistent calculation of the subgrid–scale processes.

In case of the clouds microphysics with explicit cloud modeling (thus the cloud processes themselves are not "parameterized"), certain variables must be passed over to different components of the model, that plays a role of "host model" in this context. For example, the mixing ratios, $q_c$ and $q_r$, of clouds and rain must be counted for an accurate definition of the buoyancy in the momentum equation. Some radiation schemes require inputs of mean radius, $r_c$ and $r_p$, of the cloud and rain droplets, although those are typically *not* prognostic variables of the cloud microphysics. Those variables are considered to be "the necessary variables (outputs) for the host model".

The case of data assimilation is more subtle, because there is neither a host model nor another model components to which information must be passed around. Yet, for the operational purposes, we are not interested to know a full shape of a probability distribution of a variable in order to quantify the uncertainty. In traditional assimilation formulations, we merely asks for the standard–deviation errors/uncertainties of variables: those are considered the "necessary outputs" for the data assimilation.

- *Accuracy the derived model*:

As the Reviewer correctly points out »It is not clear how accurate the derived model can be. More systematic evaluation of the error is desired.« In the present study, only a preliminary evaluation of the method is presented for the simplest case in Sec. 6. Further evaluations are performed in Yano (2024), which appeared online only after the submission of the present manuscript. The reference to Yano (2024) has been added in revision (*e.g.*, L1000–1001).

By following the Reviewers' suggestion, in the revised text, the following more basic points has been added: »If the assumed PDF is an exact solution of the

original equation, and if the initial PDF follows the assumed form, there is no error. We may further expect that the error remains small even if the initial PDF does not follow the assumed form« (L193–196).

In general, unfortunately, there is no obvious methodology for predicting the potential errors of the methodology. It appears to us that the only feasible approach is to run a model explicitly for an evaluation. For this reason, it is not possible for us to »provide a more careful discussion on this point« at this point. This remark has also been added in revision (L1001–1004).

- *Advantage of using the maximum–entropy principle*:

Although here the Reviewer states that »The advantage of using the maximum-entropy principle is not fully clear to me«, the issue to be fully clarified is not quite well stated in the comments. Instead, the Reviewer appears to be rather supportive to this principle: »I agree it is a convenient way to estimate the PDF from. At the same time, the theory presented in Sec.5.1 can be applied to any PDF form.« We do not assume that »the error is smaller if we choose the PDF form based on the maximum-entropy principle« even implicitly. Exactly as the Reviewer remarks, because »this is not at all trivial«. Please refer to the discussions over L205–220 (L231–240 in revision) and the references therein for more.

- *Examples presented in the manuscript*:

As the Reviewer states, »The examples presented in the manuscript are rather simple.« We believe that these are legitimate choices considering the main goal of the present manuscript presenting the principles, rather than proving them. Also as the Reviewer correctly points out, »Eq.(2.1) is their ultimate target of the theory«. Yet, such a full development is still far from the present state of the development, as can also be perceived by the sequel paper (Yano 2024). Thus, presenting any »examples based on on Eq.(2.1)« is also just too premature at this stage.

- *Applicability to both subgrid–scale modeling and data assimilation*:

Our argument for the applicability of the proposed formulation to both »sub-grid–scale modeling and data assimilation« merely remains a formal level (See L27–37: L27–36 in revision): we propose a general formulation for solving the distribution problem, that must be clear for all. Since the problem of both »subgrid–scale modeling and data assimilation« reduce to that of the distri-bution in space and of the probability, respectively, it is also natural to claim that the present formulation is applicable to both of those problems. Issues of formulating the subgrid–scale modelings as a distribution problem is already extensively discussed in Yano (2016): though this paper is already cited, in re-vision, this very point has more been explicitly stated in revision (L78–79). A full formulation based of the data assimilation under the present frame-work is still to be fully developed, yet a preliminary note is already available: https://drive.google.com/file/d/1i1NowEip69t5LdUOdZlBrdxDtm6wZyXI/view?usp=drive_link. However, this material is not yet at an appropriate state to be quoted in a more formal manner.

Readers are advised to refer to the references cited in the paragraph over L55–59

(L55–60 in revision). The lead sentence of this paragraph has been modified in revision to make is clear where they can find necessary references to understand how the distributions are applied to those three problems.

The present Reviewer concludes the general remarks by stating that »The manuscript is well organized«. Yet, the Reviewer also points out that it »sometimes lacks clarity;« and »some inconsistency in the notation and logical flow«: these issues has been addressed in revision by fully considering the further comments by the present Reviewer in the following.

**Major Comments**

The Reviewer suggests that »the quality of the study will be significantly enhanced« by addressing the following Major Issues. To those we respond as follow.

*1) [request] P.4 ll.109–115 "Typically, as argued by Yano et al. (2005) ...":*

As the Reviewer correctly points out, the original sentence (L109–110) leading to Eq. (2.1) is »ambiguous and confusing«. Also, as the Reviewer correctly points out, not all the physical variables considered in atmospheric science take the form of Eq. (2.1). A good example is the radius, $r$, of the water droplets as considered in Sec. 6, as the Reviewer points out: it would even not be fair to argue that "typically" the governing equations for the physical variables take the form of Eq. (2.1), and others are "exceptions".

A more precise statement would be that many dependent variables in atmosphere depend on both space and time such as the temperature, moisture (water–vapor mixing ratio), etc: those variables are advected by the wind (including the wind itself), as represented by Eq. (2.1): the lead sentence in concern has been modified accordingly in revision (L116). Yano *et al.* (2005) and Yano (2016) show that the basic formulations for the subgrid–scale modelings (parameterizations) can be reproduced by simply examining this general form (2.1). More specifically, Yano (2014) shows that all the essential, basic standard formulas for the mass–flux convection parameterizaiton can be reproduced by only considering Eq. (2.1). These elaborations have also been added in revision (L121–124).

Finally, the source term, $F$, simply includes all the physical tendencies of a variable, $\phi$, apart from the advection tendency. This remark has been added in revision, too (L118–119). Concerning the possibility of $F$ containing spatial derivatives, please refer to L125—126 (L138-139 in revision).

*2) [request] Eq.(2.2) and p.5 ll.123–124 "As a specific example, ...":*

The Reviewer argues that the introduction of Eq. (2.2) is also »confusing«: however, we are rather puzzled with this. The present Reviewer strangely attaches some physical significance to Eq. (2.2), when there is no such suggestion is made in the text. This equation in concern is introduced by merely "for ease of the deductions" (L116–117: 125–126 in revision). Probably, this lead sentence was too terse to avoid any misunderstanding, thus has been further elaborated in revision by furhter addinig the phrase "and without arguing for any general

physical relevance" (L128) immediately following Eq. (2.2).

Please also note that the discussion concerning the condensation growth in the last half of the same paragraph (L120–125 in original) is not directly linked to Eq. (2.2), but more about the generality of the source term, $F$, without specifying it in the present study. To avoid this confusion, this part has been made a standalone paragraph in revision (L133–140).

*3) [comment] P.6 l.153 "..., there is no closed analytical formula for reconstructing the original distribution from a given series of moments: ...":*

We still believe that this statement is correct. Of course, there are many formulas that link between the moments and the corresponding distribution. A particular, general category is called the "generators", because this function, defined from a given distribution, can generate the corresponding moments in a sequential manner. Here, what the present Reviewer points out is such an example that can be obtained under the saddlepoint approximation. However, please note that as the case with any other generators, this version of generators can generate the moments from a given distribution, but not other way round.

Please also note that the maximum entropy principle is based on a completely different principle: it *does not* estimate nor approximate a given distribution, although the moments may be used for this purpose. This principle simply derives the "most likely", with its meaning carefully discussed in Sec. 3.3 of the manuscript, for a given particular system when the constraints to this system is known. However, there is no guarantee that the system follow this distribution (L209–210: L235–236 in revision): it merely suggests to be "most likely".

*4) [request] P.6 ll.176–177 "The prognostic equations for these moments, or diagnostic approximations of these equations, are, in turn, known from the turbulence theories; ...":*

Here, this is another example that the discussions of the manuscript tends to be biased towards the subgrid–scale distribution problem due to the lead author's interest. In revision, the clause of "in the context of the subgie–scale distribution problem" has been added for the clarity (L201). We are afraid that no equivalent procedure is known for both the cloud microphysics and data assimilation.

*5) [request] P.12 ll.347–349 "More general formulations for the partial-differential equations (PDE) ...":*

We disagree with the request by the present Reviewer to explicitly present the prediction equation for the distribution of variables governed by Eq. (2.1) for the three reasons: 1) although Eq. (2.1) is the ultimate application, it is not at all considered in the present manuscript; 2) the probability equation for the system (2.1) is fairly complicated, and nothing would be effectively understood just by looking at this equation (see *e.g.*, Eq. 15 of Larson 2004); 3) under the assumed prognostic PDF approaches, as discussed in Sec. 5.1, immediately after Eq. (5.10b), prognostic equations for the PDF parameters can be derived directly from the governing equation (2.1). Thus, there is no need to consider the probability equation for the system (2.1) in the end, as pointed out in Sec. 5.1.

This point is already remarked in Sec. 2: see L132–133 (L145–146 in revision). The same remarks has been repeated earlier in the paragraph with Eq. (2.2) in revision so that it is even harder to miss this point (L130–132).

To make this last point better stands out, in revision, the corresponding discussion in Sec. 5.1 has been expanded into a standalone short subsection (Sec. 5.3).

At the same time, for the satisfaction of curiosity of the present Reviewer as well as some readers, we will directly refer to Eq. (15) of Larson (2004) for its explicit form in revision (L377): this reference must be readily accessible for most of the ACP readers.

6) *[question] Sec.4.2 "Output–Constrained Distribution Principle"*:

As stated in response to the item "Output-constrained maximum-entropy principle" in general remarks, we identify the two separate issues behind: the first is the fact that this principle literally works only when a distribution–based approach is adopted for a subgrid–scale modeling problem. In the context of the cloud microphysics and data assimilation, this notion of the »necessary variables (outputs)« for the host model must be generalized than its literal meaning. Second, more fundamentally, the notion of the "necessary outputs for the host model" is not well explained even in the context of the subgrid–scale modeling problem. These two issues have been further elaborated in revision.

More specifically, in the context of the cloud modeling, the "necessary output variables for the host model" are *not identical* to the prognostic variables used in a cloud model (L471–473). The question here is what variables are required as output from the cloud model for the whole system. Clearly the mixing ratios, $q_c$ and $q_r$, for the cloud and rain are important for defining the buoyancy that drives the momentum equation, for example. On the other hand, it is less obvious where the system would require the number densities, $n_c$ and $n_r$, of the cloud and rain: a certain radiation scheme may required this, but not always. In other words, although $n_c$ and $n_r$ are the prognostic variables of the cloud model, these may not be output variables required in the host model.

7) *[request] Eq.(4.1)*:

In revision, it has been mentioned that »$F$ in Eq.(2.2) corresponding to Eq.(4.1) is the Brownian motion«, and also that Eq. (4.1) is a special case of the Fokker–Planck equation (L507–509).

In the present study, $F$ is assumed to be deterministic, as already suggested in Sec. 3.5 (L344, L373 on revision), and also stated earlier in Sec. 2 in revision (L139–140). Generalization with stochasticity is already remarked at L347 (L369 in revision) as well as L960–962 (L1033–1034 in revision).

8) *[request] P.16, l.465 "..., it suffices to take a Gaussian distribution, . . . "*:

By following the request of the Reviewer, in revision, the following remark has been added immediately following Eq. (4.1): Note that in this particular case, the adopted distribution form also corresponds to an exact solution of the system (4.1: L514).

9) *[request] P.21, ll.573–576 "As in the case with the Gaussian equation, more generally, when the assumed PDF form constitutes an exact solution of a given system, ... ":*

This point can be understood directly from the fact that Eq. (5.7a) is equivalent to the original Liouville equation (3.24) under the given assumed PDF form. This remark has been added in revision, by following the request of the present Reviewer (L626–627).

10) *[question] P.22, ll.618–619 "Eq. (5.8a) or (5.10b) further simply reduces to a diagnostic method based on moments, ... ":*

The phrase "diagnostic method based on moments" has been modified in revision as "diagnostic method based on moments typically adopted in the subgrid–scale assumed PDF formulations" (L666–668).

11) *[comment] Sec.5.1 "General Formulation":*

As stated in the introduction, the present study addresses the two major open questions associated with the assumed PDF approaches: 1) how the assumed PDF form can be determined? and 2) how the parameters for the assumed PDF can be predicted consistently? As emphasized in the concluding section (L933–940: L1005–1012), these two questions are addressed separately in the present study. Thus, the prognostic assumed–PDF formulation presented in Sec. 5 does not necessarily follow from the assumed PDF form defined by the output–constrained maximum entropy introduced in Sec. 4, as the Reviewer correctly points out here: See L89–97 (L94–102 in revision). This point will also be made more explicit in the introduction in revision (L73–75).

It also follows that the relation, $\partial p/\partial \lambda_i = -\sigma_i p$, expected from the maximum entropy principle (3.15), is only a special case of the general formulation considered in Sec. 5. Also considering the fact that this reduction does not much simplify the formulation (it still takes about the same space in the equations), we will not introduce this simplification in revision, although the Reviewer suggests to do so.

12) *[question] P.24 ll.666–667 "..., it is not directly required in any microphysical tendencies within a model.":*

We are afraid that the Reviewer is bit confused with this sentence: "it" here refers to the reflectivity, $Z$, rather than the mixing ratios, $q_c$ and $q_r$, as the Reviewer somehow assumes. To avoids this confusion, "it" has been replaced by "the reflectivity, $Z$" in revision (L732). Please refer to our response to the item 6), if more background issues must be addressed.

13) *[request] P.25 l.699 "..., setting the weight as $\sigma_1 = \phi^n$ ...":*

See our response to the item 11) above for the general matters.

There is no confusion here, once one understands that the determination of the assumed PDF form and the prediction of the assumed–PDF parameters are mutually independent procedures. Note especially that the output–constrained maximum entropy principle is merely a guiding principle, but *not* a physical

principle that the system must satisfy: *cf.*, the discussion over L205–214 (L231–240 in revision). Thus, we can choose a constraint, $\sigma_1$, for a given distribution, that is *not dictated* by the output–constrained maximum entropy principle, and without contradicting with any physics.

Putting it differently, although the exponential distribution is derived by assuming the mean as a sole output variable (*i.e.*, constraint), the general formulation in Sec. 5 can be applied to any $\sigma_1$; it may not necessarily correspond to the assumed output variable (*or* constraint), that is used for deriving the given distribution. The question that we pose here is that how sensitive the evolution of the distribution by trying to predict a different statistical quantity, $\langle \sigma_1 \rangle$, consistently, based on Eq. (5.10a). This minor exercise is very worthwhile to show, because the result is quite sensitive to the choice of the weight, $\sigma_1$, as shown in Fig. 2. [Please note that due the error pointed out in the item 19) by the Reviewer, Fig. 2 has also been corrected.]

See L764–767 in revision.

**Minor Comments**

*14) [request] P.2, ll.30–31 "Here, it is hard to overemphasize the clear difference between these two distributions.":*

The point here is very simple: distribution and probability are the two distinctively different concepts, and this simple fact must be well respected. Note that neither a (frequency) distribution of a subgrid–scale variable nor a size distribution of hydrometeor particles is a probability. Conversely, the probability reduces neither to a subgrid–scale distribution nor any other distributions. The text has been elaborated in revision (L30–31).

*15) [request] P.9 Eq.(3.15) $p_0$:*
Yes, it is $p_0 = \exp(-\lambda_0)$, as beiing remarked in revision (L287).

*16) [comment] P.13 Eq.(3.25):*
Eq. (3.25) has been modified from an advection form to a flux form as suggested in revision.

*17) [comment] P.13 Eq.(3.26b):*
We much appreciate the present Reviewer for pointing us out the mistakes concerning the stochastic collection equation (3.26b). As pointed out, in revision, the argument of the equation has been modified from the size, $r$, to the mass, $m$. Also a missing factor $1/2$ in front of fhe first term on the right–hand side has been added.

*18) [request] P.20 Eq.(5.2a):*
We note certain difficulties for using the two notations, $\lambda_0$ and $p_0$, for the normalization factor of a distribution, as pointed out by the Reviewer here. Yet, we are inclined to stick to this "double standard" considering the advantages of both notions: the choice of $\lambda_0$ as a normalization factor in discussing the assumed–PDF in general manner, as in Sec. 3.2, has an unbeatable advantage

to treat all the parameters of a distribution with a single notation, $\lambda_i$. On the other hand, in more specific situations, such as in Eq. (3.15), it is more intuitive to adopt the notion of $p_0$ for a normalization constant. Note that in those situations, other PDF parameters also often take different notations than $\lambda_i$ for the same reason.

In revision, a remark has been added in introducing Eq. (5.2a): L598–599. Furthermore, more general remarks have been added in L176–179 and L293—294.

*19) [question] P.25 Eqs.(5.16a) and (5.16b)*:
This is a very sharp observation by the Reviewer!: of course, the original Eq. (5.16b: Eq. 5.17b in revision) was wrong, and it must become

$$\frac{d}{dt}\langle \phi^n \rangle = \int \phi^n \frac{\partial p}{\partial t} d\phi = n \int p\phi^{n-1} F d\phi$$

for consistency, as the Reviewer points out. The reduction of this part was carefully re–examined, and the errors were identified: after those corrections, Eq. (5.17b) indeed reduces to the above. As a consequence, Eq. (5.17a) must also be modified into:

$$\lambda_1(t) = \left[ \frac{1}{\lambda_1^n(0)} + \frac{n}{n!} \int_0^t \langle F\phi^{n-1} \rangle dt \right]^{-1/n}.$$

Those modifications have been applied in revision.

**Typo**
20)–27) We much appreciate the various typos pointed out by the present Reviewer. All those typos have been corrected in preparing the final manuscript.

---

## Referee Report (RR1)

**Review of "General Formulation For the Distribution Problem: Prognostic Assumed PDF Approach Based on The Maximum–Entropy Principle and The Liouville Equation" by Yano et al.**

After careful assessment, I have decided to recommend this manuscript for publication after a minor revision.

Some of the points I previously raised have been addressed in the revised manuscript. A guide for applying the theory to more general forms of systems is now included in Sec. 5.3. The meaning of the output-constrained maximum-entropy principle is further elaborated. Additional details for clarification are provided throughout. All of which contributed to improving the manuscript's quality.

What is still missing is a more thorough evaluation of the error in the derived model, as such information is crucial to understand the methodology's reliability. I agree, however, with the authors that deriving and evaluating the error systematically would be complex. To give readers a rough idea of when and how the derived model performs accurately, a more in-depth discussion of the error using simple examples would be valuable. For this purpose, I request the authors to expand the discussion related to Fig.2 (see also Comment (6) below). Additionally, I would suggest exploring other simple but qualitatively different cases, such as those involving fixed points (steady solutions):

- $d\phi/dt = -\phi$. Here, $\phi = 0$ is a globally stable fixed point.
- $d\phi/dt = \phi(\phi - 1)$. Here, $\phi = 0$ is a stable fixed point, and $\phi = 1$ is an unstable fixed point.
- $d\phi/dt = \phi(\phi - 1)(2 - \phi)$. Here, $\phi = 0, 2$ are stable fixed points, and $\phi = 1$ is an unstable fixed point.

Some more comments are provided below.

**Major Comments**

1) **[request] P.4 ll.118–119 "... $F$ designates all …"**
   It is not at all obvious to the readers that $F$ can be space dependent. Please clarify this point here. I would explain $F$ is a functional of $\phi(x, y, z)$.

2) **[comment] P.6 ll.139–140 "…, there is no closed analytical formula for reconstructing the original distribution from a given series of moments: …"**
   For the authors' information, I found an interesting paper that is closely relevant to this problem.
   > Chao Dang and Jun Xu, "Novel algorithm for reconstruction of a distribution by fitting its first-four statistical moments", Applied Mathematical Modelling, Volume 71, 2019, Pages 505-524, https://doi.org/10.1016/j.apm.2019.02.040.

3) **[question] P.23 Eq.(5.10b)**

I think we can solve the derived model in a slightly different way. Let us consider $\{\langle \sigma_l \rangle\}$, not $\{\lambda_l\}$, are the prognostic variables.

1. From $\{\langle \sigma_l \rangle(t)\}$, we can estimate the PDF $p(\phi, t)$ by using the maximum entropy principle.
2. Using the estimated $p(\phi, t)$, we can calculate the $\{\langle F_{\sigma_l} \rangle(t)\}$.
3. Using Eq.(5.10b), we can numerically calculate $\{\langle \sigma_l \rangle(t + \Delta t)\}$.

By repeating this procedure, we can numerically calculate the time evolution of weights $\{\langle \sigma_l \rangle(t)\}$. Isn't this easier than solving Eq. (5.8a)?

4) **[comment] Sec.5.3 "Generalization to the PDE system (2.1)"**

I appreciate that the authors added this section; we can now see that the generalization of the proposed theory is indeed straightforward. From the derivation provided in this section, I also feel that we may not need to bring the Liouville equation.

5) **[comment] P.28, ll.787–790 "... the solution breaks down beyond this point …"**

I think this behavior is reasonable. Please note that the solution of $d\phi/dt = \phi^n, \phi(0) \neq 0$ blows up in finite time if $n > 1$.

6) **[question, request] Fig.2**

Is Fig.2b correct? If I understand Eq. (5.17e) correctly, $\lambda_1(t)$ does not depend on $n$ when $m = 1$.

For each $m = 0, 1, 2, 3$, please plot the true $\lambda_1(t) = 1/\langle \phi \rangle(t)$ and discuss which choice of $n$ is the most accurate.

**Minor Comments**

7) **[request] P.5 ll.139–140 "Furthermore, in the present study, the source term, $F$, is assumed to be deterministic."**

This is not correct. Brownian motion is considered in Sec. 4.4. Please rephrase.

**Typo**

8) **P.8 l.219 "... be be …"**

9) **P.11 l.314 "... to constraint …" -> "... to constrain …"**

10) **P.21 l.582 "... with by …"**

11) **P.22 l.622 "... $\partial^2 \phi/\partial \phi^2$" -> "... $\partial^2 p/\partial \phi^2$"**

12) **P.26 l.730 "... questioned form"**

**References**

Chao Dang and Jun Xu, "Novel algorithm for reconstruction of a distribution by fitting its first-four statistical moments", Applied Mathematical Modelling, Volume 71, 2019, Pages 505-524,https://doi.org/10.1016/j.apm.2019.02.040.

---

## Referee Report (RR2)

**Review of "General Formulation For the Distribution Problem: Prognostic Assumed PDF Approach Based on The Maximum–Entropy Principle and The Liouville Equation" by Yano et al.**

I would like to recommend this manuscript for publication after a minor revision. Following the reviewer's previous comments, the authors have corrected Fig 2 and added more examples, all of which provide valuable insights for the readers to understand when and how the proposed method would work efficiently. Below, I have raised a few additional points to further enhance the quality. Further review of the revised manuscript will not be necessary, but please consider incorporating them into the manuscript.

**Major Comments**

1) **[question] P.46 ll.1182–1186 "In the case with the model (ii), the calculations with the exact distribution gets its own problem: ..."**
The mean and variance cannot be defined for some fat-tailed distributions such as the Cauchy distribution. I speculate that this may be the cause of the slow numerical convergence. What do you think?

2) **[suggestions] Figs. 2 and 9**
If the exact solutions of the ODE do not blow up in finite time (e.g., $m = 0$ and 1 in Sec. 5.5, and (i) and (iii) in Appendix C), the moments predicted by the proposed method do not deviate significantly from the exact solutions. I think this is an interesting property worth highlighting somewhere in the manuscript.

**Minor Comments**

3) **[typo] P.21 l.589 "reply" -> "rely"**

4) **[suggestion] P.28 l.803 "..., and in this case we find:" -> "..., and in this case we find for all n:"**

5) **[typo] P.47 Fig.9 "(i) and (ii)" -> "(i) and (iii)"**

---

## Author Response (AR2)

**Final Response to the Referee 1 (Report #2)**

*General Remarks*

The following response is presented in a self–contained manner, *i.e.*, the response can be read without referring to the original review comments. Yet the original comment texts are also inserted with green. Please note that the Referee comments are quoted by »...« whenever they are quoted in the response text.

*Response*

We much thank to the present Referee, because the manuscript has been substantially augmented by adding the two further appendices in response.

After careful assessment, I have decided to recommend this manuscript for publication after a minor revision.

We are glad to learn that »After careful assessment«, the present Referee has »decided to recommend this manuscript for publication after a minor revision«.

Some of the points I previously raised have been addressed in the revised manuscript. A guide for applying the theory to more general forms of systems is now included in Sec. 5.3. The meaning of the output-constrained maximum-entropy principle is further elaborated. Additional details for clarification are provided throughout. All of which contributed to improving the manuscript's quality.

Also gladly the present Referee acknowledges that »Some of the points ... raised have been addressed in the revised manuscript«. Especially, »A guide for applying the theory to more general forms of systems is now included« as a standalone »Sec. 5.3«. Furthermore, »The meaning of the output-constrained maximum-entropy principle is further elaborated«, and »Additional details for clarification are provided throughout«. The Referee concludes that »All of which contributed to improving the manuscript's quality.«

What is still missing is a more thorough evaluation of the error in the derived model, as such information is crucial to understand the methodology's reliability. I agree, however, with the authors that deriving and evaluating the error systematically would be complex. To give readers a rough idea of when and how the derived model performs accurately, a more in-depth discussion of the error using simple examples would be valuable. For this purpose, I request the authors to expand the discussion related to Fig.2 (see also Comment (6) below). Additionally, I would suggest exploring other simple but qualitatively different cases, such as those involving fixed points (steady solutions):

• $d\phi/dt = -\phi$. Here, $\phi = 0$ is a globally stable fixed point.

• $d\phi/dt = \phi(1-\phi)$. Here, $\phi = 0$ is a stable fixed point, and $\phi = 1$ is an unstable fixed point.

• $d\phi/dt = \phi(1-\phi)(2-\phi)$. Here, $\phi = 0, 2$ are stable fixed points, and is $\phi = 1$ an unstable fixed point.

Yet, the Referee also points out that »a more thorough evaluation of the error in the derived model« is »still missing«. The Referee further emphasizes that »such

information is crucial to understand the methodology's reliability«. At the same time, the Referee also agrees with us, as we responded previously, that »deriving and evaluating the error systematically would be complex«. As a compromise, this time, the Referee alternatively suggests »a more in-depth discussion of the error using simple examples« »To give readers a rough idea of when and how the derived model performs accurately«. As more specific proposals along this line, the Referee requests us more specifically two additional exercises:

1) »to expand the discussion related to Fig.2«: The Referee further elaborates on this exercise in the Major Comments 5) and 6) below. Thus, we also provide our response on this item below, only remarking for now that this exercise has been performed in revision, as explained below.

2) to explore »other simple but qualitatively different cases, such as those involving fixed points (steady solutions)«:

More specifically, the Referee suggests to consider the following simple dynamical systems:

(i) $d\phi/dt = -\phi$. »Here, $\phi = 0$ is a globally stable fixed point.«

(ii) $d\phi/dt = \phi(1 - \phi)$. »Here, $\phi = 0$ is a stable fixed point, and $\phi = 1$ is an unstable fixed point.«

(iii) $d\phi/dt = -\phi(1 - \phi)(\phi - 2)$. »Here, $\phi = 0, 2$ are stable fixed points, and $\phi = 1$ is an unstable fixed point.«

These three dynamical systems have been thoroughly examined in revision in a newly–introduced Appendix C.

Some more comments are provided below.

The Referee further provides »Some more comments«:

**Major Comments**

1) [request] P.4 ll.118–119 "... $f$ designates all ..."

It is not at all obvious to the readers that point here. I would explain can be space dependent. Please clarify this $F$ is a functional of $\phi(x, y, z)$.

By following the suggestion, the following sentence has been inserted in revision (L119–120):

"The source, $F$, generally depends on the variable, $\phi$, and also possibly on time $t$ and space."

2) [comment] P.6 ll.139–140 "..., there is no closed analytical formula for reconstructing the original distribution from a given series of moments: ..."

For the authors' information, I found an interesting paper that is closely relevant to this problem. Chao Dang and Jun Xu, "Novel algorithm for reconstruction of a distribution by fitting its first-four statistical moments", Applied Mathematical Modelling, Volume 71, 2019, Pages 505-524, https://doi.org/10.1016/j.apm.2019.02.040.

We much appreciate the present Referee for pointing us out an existence of a very interesting paper by Dang and Xu (2019) on this subject: the essence of this paper is based on the fact that under an assumed PDF formulation (in our own terminology), it is indeed possible to define the PDF parameters from

a given set of moments, *i.e.*, the operation of inverting the relations (3.4) to (3.3) in the manuscript, once a PDF form is pre–defined, as well known in the assumed–PDF community. Yet, the paper goes even further by asking a question of which PDF forms (or PDF models, in their own terminology) provides the best fit when several options of the PDF forms are considered.

Thus, this work is still short of resolving the issue in concern. Yet, we have decided to cite this paper in revision along with Daniels (1954) and Butler (2007), suggested in the previous round by the present Referee, so that extensive literature on the relations between the moments and the distribution can be suggested to the readers (L170).

3) [question] P.23 Eq.(5.10b):
I think we can solve the derived model in a slightly different way. Let us consider $\{\langle\sigma_l\rangle\}$, not $\{\sigma_l\}$, are the prognostic variables.
1. From $\{\langle\sigma_l\rangle(t)\}$, we can estimate the PDF $p(\phi, t)$ by using the maximum entropy principle.
2. Using the estimated $p(\phi, t)$, we can calculate the $\{\langle F_{\sigma_l}\rangle(t)\}$.
3. Using Eq.(5.10b), we can numerically calculate $\{\langle\sigma_l\rangle(t + \Delta t)\}$.
By repeating this procedure, we can numerically calculate the time evolution of weights $\{\langle\sigma_l\rangle(t)\}$. Isn't this easier than solving Eq. (5.8a)?

Here, the present Referee argues that at every time step, once the values of $\{\langle\sigma_l\rangle(t)\}$ are updated, we can evaluate the updated distribution form simply by by invoking the maximum entropy principle without performing a time integral of the parameters. The proposal, indeed, sounds appealing with a first glance. However, the Referee is unfortunately missing some key issues behind.

First, the maximum entropy principle can merely define a general distribution form, as given by Eq. (3,15) in the manuscript, once the constraints, $\{\langle\sigma_l\rangle(t)\}$, are specified. However, as explicitly pointed out immediately following this formula, the maximum entropy principle itself does not provide specific values for the distribution parameters, $\{\lambda_l(t)\}$. Those are still to be determined by inverting the relations (3.4) into (3.3). This inversion is precisely the main difficulty with the standard assumed–PDF approaches, because the relations (3.4) are often highly nonlinear, and their inversions into the form (3.3) are hardly trivial, as already suggested in the introduction, and also more specifically discussed in Sec. 3.2.1. The proposed prognostic formulation circumvents the difficulty of this inversion problem.

To make this point clearer, in revision, the following short remark is added towards the end of Sec. 5.1 (L654–656):

"Realize that the key step introduced in the formulation here is to prognose the PDF parameters, $\{\langle\lambda_l\rangle(t)\}$, by Eq. (5.8a). In this manner, we circumvent the principal difficulty of the current assumed–PDF approaches of inverting the relations (3.4) into the form (3.3)."

4) [comment] Sec.5.3 "Generalization to the PDE system (2.1)":
I appreciate that the authors added this section; we can now see that the generalization of the proposed theory is indeed straightforward. From the derivation provided in this section, I also feel that we may not need to bring the Liouville equation.

The present Referee appreciates »that the authors added this section«. The Referee further remarks that »I also feel that we may not need to bring the Liouville equation.« Yes, it is indeed true that Eq. (5.11d) can be derived without invoking the Liouville equation. However, we still believe it important to invoke the Liouville equation explicitly so that we can also explicitly demonstrate that we are indeed solving the Liouville equation under this approach, albeit with an extremely truncated form. Furthermore, without directly invoking the Liouville equation, it would be rather difficult to identify the middle expression in Eq. (5.11d), that explicitly links the temporal tendency of $\langle \sigma_l \rangle$ to that of the PDF parameters, $\lambda_l$.

5) [comment] P.28, ll.787–790 "... the solution breaks down beyond this point . . .":

I think this behavior is reasonable. Please note that the solution of $d\phi/dt = \phi^2$, $\phi(0) \neq 0$ blows up in finite time if $n > 1$.

No, this behavior is not reasonable. Curves for the exact solutions have been added in green in revision to Fig. 2, and a full discussion of this system has also been presented in the Appendix B in revision. As the Referee correctly points out, the system with »$d\phi/dt = \phi^2$, $\phi(0) \neq 0$ blows up in finite time if $n > 1$«. However, as explicitly discussed in the newly–added Appendix B, this merely translates into a strong tendency of the distribution to stretch towards the larger values with time, and the distribution itself continues to evolve without singularity.

6) [question, request] Fig.2:

Is Fig.2b correct? If I understand Eq. (5.17e) correctly, $\lambda_1(t)$ does not depend on $n$ when $m = 1$.

For each $m = 0, 1, 2, 3$, please plot the true $\lambda_1(t) = 1/\langle \phi \rangle(t)$ and discuss which choice of $n$ is the most accurate.

We much appreciate a question concerning this figure, because there was a totally stupid error in the code, in which the definitions of the parameters, $m$ and $n$, are swapped round everywhere, that also led to additional errors. The figure has been thoroughly modified in revision: the four panels now show the cases for $m = 0, 2, 3$, and 4, because the case with $m = 1$ is trivial, as the reviewer correctly pointed out. As already remarked in response to 5), the exact PDF solutions for those systems have been evaluated in revision in the Appendix B, and furthermore, the values of $\lambda_1$ diagnosed from $\langle \phi^n \rangle$ obtained by using the exact evolution of the distributions are also added as green curves in Fig. 2.

**Minor Comments**

7) [request] P.5 ll.139–140 (L140–141 in revisoin) "Furthermore, in the present study, the source term, $F$, is assumed to be deterministic":

This is not correct. Brownian motion is considered in Sec. 4.4. Please rephrase.

By following the suggestion of the Referee, the phrase "but except for the case of the Brownian motion considered in Sec. 4.4" has been added in revision.

**Typo**

8) P.8 l.219 "... be be ..."

9) P.11 l.314 "... to constraint ..." → "... to constrain ..."

10) P.21 l.582 "... with by ..."

11) P.22 l.622 "... "... $\partial^2\phi/\partial\phi^2$" → "... $\partial^2 p/\partial\phi^2$"

12) P.26 l.730 "... questioned form"

Various typos have been corrected as pointed out by the Referee. We much appreciate the careful proof reading given. Furthermore, for finalizing our manuscript thorough proof reading has also been applied to remove further typos and grammatical errors.

---

## Author Response (AR3)

**Further Response to the Reviewer**

**General Remarks**

The following response is presented in a self–contained manner, *i.e.*, the response can be read without referring to the original review comments. Yet the original comment texts are also inserted with blue. On the other hand, the Reviewer's comments are quoted by »...« when they are quoted in the response text.

**Response**

I would like to recommend this manuscript for publication after a minor revision. Following the reviewer's previous comments, the authors have corrected Fig 2 and added more examples, all of which provide valuable insights for the readers to understand when and how the proposed method would work efficiently. Below, I have raised a few additional points to further enhance the quality. Further review of the revised manuscript will not be necessary, but please consider incorporating them into the manuscript.

We are very glad to read that the present Reviewers now »would like to recommend this manuscript for publication«. In response, we have performed »a minor revision« by following the Reviewer's »few additional points to further enhance the quality.«

**Major Comments**

1) [question] P.46 ll.1182–1186 "In the case with the model (ii), the calculations with the exact distribution gets its own problem: ...":

The mean and variance cannot be defined for some fat-tailed distributions such as the Cauchy distribution. I speculate that this may be the cause of the slow numerical convergence. What do you think?

Yes, the difficulty with the model (ii) is due to the »fat-tailed distributions«. More precisely, the distribution increasingly presents a long tail to the positive direction with time, that leads to the difficulty. This tendency is reflected to singularities in the solutions (C.4a, b), that lead to a more direct cause of difficulties in the integrals. These points are elaborated in revision (ll.1196–1198).

2) [suggestions] Figs. 2 and 9:

If the exact solutions of the ODE do not blow up in finite time (e.g., m = 0 and 1 in Sec. 5.5, and (i) and (iii) in Appendix C), the moments predicted by the proposed method do not deviate significantly from the exact solutions. I think this is an interesting property worth highlighting somewhere in the manuscript.

The singular behavior of the solutions for the PDF parameters has been augmented in in Sec. 5.5 (ll.794–807), and explicitly noted in the Appendix C (ll.1204–1205) in revision.

As the Reviewer correctly points out, it is an important conclusion from the present study that the proposed prognostic assumed–PDF calculation overall works well, when the solutions for the assumed PDF parameters are not singular. We also note a similar issue in multiple–dimension cases in Yano (2024). These

points have been iterated in a newly–introduced paragraph in the conclusion section (ll.1037–1042) in revision.

**Minor Comments**

3) [typo] P.21 l.589: "reply" → "rely"

"reply" has been modified to "rely" as suggested.

4) [suggestion] P.28 l.803: "..., and in this case we find:" → "..., and in this case we find for all $n$:"

"..., and in this case we find:" has been modified to "..., and in this case we find for all $n$:" as suggested.

5) [typo] P.47 Fig.9: "(i) and (ii)" → "(i) and (iii)"

"(i) and (ii)" has been modified to "(i) and (iii)" as suggested.